# Sub-pangenome analysis reveals structural variants associated with fruit color and bacterial wilt resistance in eggplant

Qian You [1,5] ✉, Ze Peng [2,5], Zhiliang Li[1,5], Yaolan Jiang[1,5], Penglong Wan[2,5], Yahui Zhao[2], Wei Zhao[1], Songyuan Zhang[1,2], Hefen Cheng[1], Chengjie Chen[2], Zhou Heng[1], Ming Hu[2], Yongfeng Zhou [3], Brandon S. Gaut [4], Baojuan Sun [1] ✉, Tao Li [1] ✉ & Yi Liao [2] ✉

Eggplant (*Solanum melongena* L.) is a globally important Solanaceae crop, yet trait-relevant genomic variants remain poorly characterized. Here, we perform population genomic analyses of 226 eggplant accessions sampled mainly from a major domestication center spanning Southeast Asia and South China, and find that genetic relationships closely track geographic origin. We generate chromosome-scale assemblies for 11 representative accessions using long-read sequencing and integrate six published genomes to build a pangenome resource. Using this resource, association scans identify a 12.4 Mb inversion on chromosome 10 segregating at 50.44% frequency that is strongly associated with fruit color, likely through hitchhiking with *SmMYB1*. We also detect variants associated with bacterial wilt resistance, including a premature stop codon in *SmCYP82D47* and copy number variations in *SmEPS1* and *SmRoq1* homologs. Together, our results illuminate the evolution and phenotypic impact of large structural variants and provide genomic resources for eggplant genetics and breeding.

Eggplant (*Solanum melongena* L.), also known as brinjal or aubergine, is a globally important vegetable crop in the Solanaceae. With annual production of 57.4–59.3 million tons worldwide, it ranks as the third most produced solanaceous crop after potato (*S. tuberosum* L.) and tomato (*S. lycopersicum* L.) (FAO, 2020–2022). Domesticated in the Old World and cultivated for centuries, eggplant is now widely grown and consumed, particularly in Asia, the Mediterranean basin, and Southeast Europe. However, its origin and domestication history remain debated. The "out of Africa" hypothesis proposes that the wild progenitor of cultivated eggplant, *S. insanum*, originated in Africa and later spread to Asia, where domestication gave rise to modern *S. melongena*[1]. An alternative model suggests multiple independent domestication events in distinct regions, including Southeast Asia and the Indian subcontinent[2,3]. These hypotheses may not be mutually exclusive, but could reflect different stages of eggplant evolution, domestication, and diffusion. Beyond *S. melongena*, two other cultivated eggplant species, scarlet eggplant (*S. aethiopicum* L.) and gboma eggplant (*S. macrocarpon* L.), are indigenous to Africa and have long been cultivated there, but are now grown more broadly worldwide[4]. *S. aethiopicum* is also used as an ornamental crop, whereas *S.*

[1]Guangdong Key Laboratory for New Technology Research of Vegetables, Vegetable Research Institute, Guangdong Academy of Agricultural Sciences, Guangzhou, Guangdong, China. [2]Key Laboratory of Biology and Genetic Improvement of Horticultural Crops (South China), Ministry of Agriculture and Rural Affairs, College of Horticulture, South China Agricultural University, Guangdong, China. [3]National Key Laboratory of Tropical Crop Breeding, Shenzhen Branch, Guangdong Laboratory of Lingnan Modern Agriculture, Key Laboratory of Synthetic Biology, Ministry of Agriculture and Rural Affairs, Agricultural Genomics Institute at Shenzhen, Chinese Academy of Agricultural Sciences, Shenzhen, China. [4]Department of Ecology and Evolutionary Biology, University of California, Irvine, CA, USA. [5]These authors contributed equally: Qian You, Ze Peng, Zhiliang Li, Yaolan Jiang, Penglong Wan. ✉e-mail: kuaileyouqianmeng@163.com; sunbaojuan@hotmail.com; tianxing84@163.com; yiliao@scau.edu.cn

*macrocarpon* is valued for both leaves and fruits. These species are thought to have been domesticated from distinct wild relatives, *S. anguivi* and *S. dasyphyllum*, respectively[5]. In contrast to several New World solanaceous crops, eggplant did not experience a strong early domestication bottleneck[6]. Nonetheless, recent evidence indicates substantial reductions in genomic diversity during subsequent spread and improvement, for example during expansion from Southeast Asia into China and Japan[7,8]. Defining genome-wide diversity in cultivated eggplant will refine our understanding of domestication and facilitate the discovery and deployment of breeding relevant alleles.

The rapid expansion of eggplant genomic resources has deepened our understanding of its origin, domestication, genome organization, and evolutionary history, and has begun to resolve the genetic architecture of several agronomically important traits. To date, genome assemblies for at least eight eggplant genotypes are available. Two were generated using short read sequencing, including 'Nakate Shinkuro'[9] and '67/3'[10,11], and six were generated using long read sequencing, including 'HQ 1315'[12], 'GUIQIE 1'[13], 'NO211'[14], and three additional accessions (PI 180485, PI 196043, and PI 200854)[15]. Although these assemblies have served as valuable reference resources for genetic and genomic studies, they have not been comprehensively integrated into a pangenome framework. Pangenomes have transformed plant genomics by representing within species diversity more effectively and reducing reference bias. An early eggplant pangenome effort used the '67/3' reference together with resequencing data from 25 accessions, identifying 816 genes absent from the reference and selective sweep regions enriched for candidate genes related to fruit color, prickliness, and fruit shape[11]. Using the same reference, more than 3,400 accessions were genotyped by Single Primer Enrichment Technology (SPET), supporting a model of independent domestication in Southeast Asia and the Indian subcontinent[7]. However, these short read based strategies are generally underpowered for detecting large structural variants, particularly insertions[16]. While our study was in progress, two eggplant pangenome studies were reported[17,18]. One constructed pangenome graphs from 40 chromosome-scale genome assemblies, whereas the other developed an Asia-representative eggplant pangenome using resequencing data from 22 accessions together with four genome assemblies. These independent efforts further underscore the value of pangenome resources for capturing within-species genomic diversity and for identifying trait-associated variants, including structural variants that contribute to key agronomic traits, such as prickliness, and *Fusarium oxysporum* f. sp. *Melongenae* resistance etc.

Genomic resources have also accelerated the dissection of agronomically important traits in eggplant, particularly fruit color and bacterial wilt (BW) resistance[19–21]. However, most mapping studies for these traits have relied on bi-parental populations[22–25], which inherently sample only a limited fraction of the available genetic and phenotypic diversity. Genome-wide association studies (GWAS) using diverse panels offer a complementary strategy and have been applied in eggplant, but have largely focused on single-nucleotide polymorphisms (SNPs) and small insertions and deletions (InDels)[22,24,26]. In contrast, structural variant-informed genetic mapping in eggplant has only begun to emerge, as exemplified by a recent pangenome study published while our manuscript was under review[17]. This is given extensive evidence that SVs contribute to speciation and domestication and can reveal trait loci missed by SNP based GWAS, including presence absence variants[15,27–30].

Here, we present population genomic analyses of a diverse panel of 226 eggplant accessions primarily collected from China and Southeast Asia, a region proposed as a major center of eggplant domestication. Using whole-genome resequencing data, we characterize population structure and genome-wide diversity and generate de novo chromosome-scale assemblies for 11 representative accessions. By integrating these assemblies and six published genomes with the resequencing dataset, we construct eggplant pangenome graphs using both reference-based and reference-free approaches. We show that this subpangenome resource enables the discovery of variants associated with complex traits. Specifically, we identify a large inversion of up to 12.4 Mb that is strongly associated with fruit color, as well as copy-number variation in candidate genes associated with bacterial wilt resistance. This pangenome resource, together with the related genes and molecular markers, accelerates biological studies and supports genomics-assisted breeding in eggplant.

## Results

### Genome sequence, population structure, and genetic diversity

To explore the genetic diversity and population structure of eggplant germplasm and to help guide our sampling for de novo genome sequencing and assembly, we collected resequencing data for 226 eggplant accessions, representing phenotypic and genetic diversity of the cultivated eggplant. The panel consisted of 219 accessions of *S. melongena*, two of its closest wild relatives (*S. insanum* and *S. incanum*), four scarlet eggplant accessions (*S. aethiopicum*), and one wild eggplant species (*S. violaceum*) (Fig. 1a, b and Supplementary Data 1). These accessions were mainly distributed across East Asia and Southeast Asia, which is hypothesized to be a main domestication center and is also the locus of > 60% of world eggplant production[2,7,8]. Among the 226 re-sequenced accessions, 198 were generated in this study and 28 were from previous studies[11–13]. This approach generated 5.50 terabytes of sequencing data, with an average coverage of 20.87× per sample (Supplementary Data 1). To understand population genetics of this diverse set, we mapped these reads to a chromosome-scale assembly (described below) of 'S076', a representative breeding line of South China that has purple fruits and is highly resistant to BW disease. Altogether, we identified 3,698,811 SNPs and 349,227 InDels (1–50 bp) of high quality.

To infer population structure, we performed principal component analysis (PCA) using 153,934 SNPs spanning all 226 accessions and filtered based on allele frequency, missing rates, and linkage disequilibrium (LD). The top three principal components accounted for 55.41% of the total variance (Fig. 1c, d). Given the noticeable slowing of the cross-validation (CV) error decline starting at $K = 4$ in the admixture analysis, the 226 accessions were assigned to four groups (C1, C2, C3, and C4). The assignment of four groups was further supported by PCA, phylogenetic analysis, and the geographical origins of the samples (Fig. 1c–f and Supplementary Figs. 1 and 2). Almost all the samples from Southeast Asia (31/35) were assigned to the C1 group. The C2 group contained accessions mainly from South America (9/12) and Europe (10/18). The majority (80.00%, 52/65) of the C3 group contained accessions from a wide range of 19 Provinces in China. The C4 group primarily consisted (80.77%, 63/78) of members from South China (especially Guangdong Province). These results indicate that the population structure of sampled eggplants is largely correlated with geographic distributions. Additionally, the LD among SNPs rapidly decreased at 105 Kb ($r^2 = 0.5$) (Supplementary Fig. 3). Nucleotide diversity (π) varied among the groups, with the C3 (π = 0.168) and C4 (π = 0.183) groups substantially lower than that of C1 (π = 0.309) and C2 (π = 0.224). These π values were comparable to that from two previous studies utilizing either wild African eggplants (0.03 - 0.10)[31] or a worldwide diversity panel (0.20 - 0.36) in eggplant[7]. The most genetically divergent groups were C2 and C4 with the highest *Fst* value of 0.212, while the closest genetic distance was observed between C3 and C4 groups (*Fst* = 0.084). Overall, the C1 and C2 groups from Southeast Asia, South America, and Europe displayed much higher genetic diversity and differentiation compared to the two groups from China (C3 and C4).

### A sub-pangenome of gene space in eggplant

We used information from the resequencing data (226 accessions) (including 'HQ-1315' and 'GUIQIE-1') together with four published

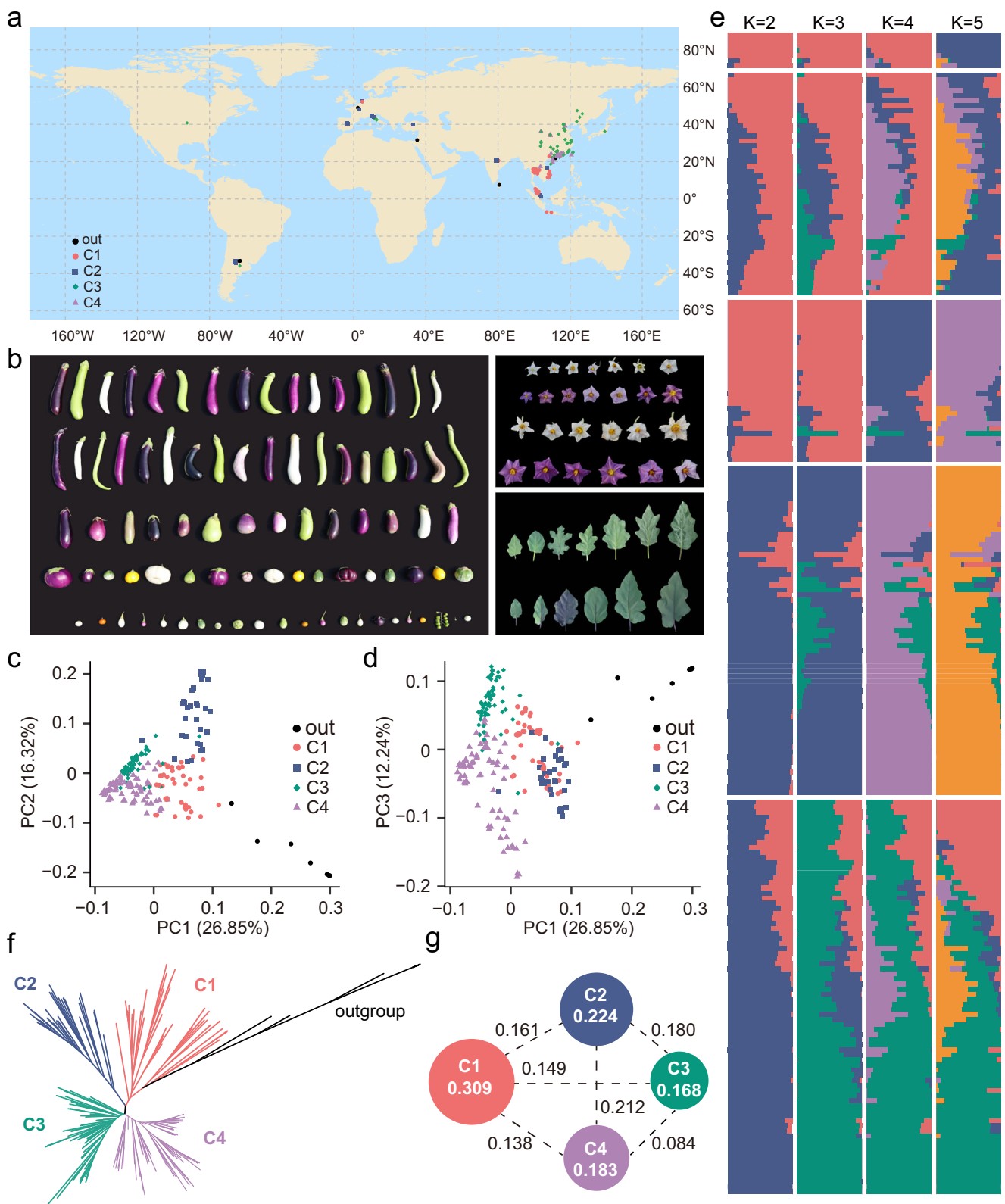

assemblies to identify 11 accessions (plus six published ones) that represent all major clusters of an eggplant phylogenetic tree (Fig. 2a). To further investigate the extent to which these 17 samples represent the genetic diversity of eggplant, a phylogenetic tree comprising 3,673 accessions (Supplementary Fig. 4) was constructed by further including accessions from two published studies[7,11]. The 17 samples generally represent most major clusters, with a focus on capturing

the genetic diversity of eggplant from East and Southeast Asia. To construct a sub-pangenome primarily targeting this regional diversity, we de novo sequenced the 11 accessions with an average of 30.18× PacBio HiFi reads and 103.79× Hi-C sequencing data for genome assembly and anchoring (Supplementary Fig. 5). The resulting 11 genomes had contig N50s ranging between 38.73 ‐ 59.67 Mb and total assembly sizes between 1.14 ‐ 1.20 Gb (Supplementary Data 2).

**Fig. 1 | Geographic distribution and genomic diversity of 226 eggplant accessions. a** Geographic sampling of the 226 eggplant accessions, each of which is represented by a colored shape on the map. Accessions from out (outgroup), C1, C2, C3, and C4 are shown as black dots, red dots, blue squares, green diamonds, and purple triangles, respectively. **b** The diverse characteristics of fruits, flowers, and leaves of eggplant accessions, representing most types of cultivated eggplants worldwide. **c, d** PCA of 226 eggplant accessions. PC1 (26.85%) clearly separates most of the eggplant accessions of China (C3 and C4) from those of Southeast Asia, South America, and Europe (C1 and C2); PC2 (16.32%) separates accessions of Southeast Asia (C1) from those of South America and Europe (C2); and PC3 (12.24%)

divides accessions from China into two subgroups (C3 and C4). **e** Ancestral component analysis of eggplant accessions with ADMIXTURE for $K = 2$-$5$. **f** A maximum likelihood phylogenetic tree of eggplant accessions using *S. violaceum*, *S. insanum*, *S. incanum*, and *S. aethiopicum* as the outgroups. The populations identified by ADMIXTURE are represented by different colors, including 44 C1 (red branches), 32 C2 (blue branches), 65 C3 (green branches), and 78 C4 (purple branches) accessions. **g** π and *Fst* of eggplant populations. Numbers in the circle represent π, and numbers next to the dashed lines show *Fst* values. Source data are provided as a Source Data file.

These chromosome-scale genome assemblies all had a Benchmarking Universal Single-Copy Orthologs (BUSCO) completeness score exceeding 98.70% and relatively more contiguous sequences compared with that of the five previously published assemblies except for accession NO211 (Fig. 2b), which supported the high quality of these assemblies. Further genome annotation predicted between 68.79% ~ 79.85% repetitive sequences and 33,620 ~ 36,174 protein-coding genes in each genome (Supplementary Data 2). A gene-based pangenome analysis was performed utilizing these 17 genomes. The number of pan-gene families increased when more genomes were added and nearly reached a plateau after adding 17 genomes (Fig. 2c). A total of 15,406 (41.15%) core, 3,657 (9.77%) softcore, 17,889 (47.79%) dispensable, and 483 (1.29%) private gene families were obtained (Fig. 2d, e). The core gene families exhibited higher gene expression (Fig. 2f) but lower nonsynonymous to synonymous substitution ($K_a/K_s$) ratios than the other types of genes (Fig. 2g, Supplementary Fig. 6a). Since these genomes were selected from different genetic backgrounds and represent phenotypic diversity for fruit color, fruit size, fruit shape, and disease resistance/susceptibility, it is possible that the dispensable gene families contribute to phenotypic variability. Among the significantly overrepresented gene ontology (GO) terms for dispensable gene families, we specifically identified a series of defense-related terms, such as "defense response", "defense response to virus", and "defense response to Gram-negative bacterium", etc. (Supplementary Data 3), which may provide a potential repertoire for identifying disease resistance/susceptibility genes in eggplant. In addition, the 11 de novo sequenced genomes were also used for a gene-based pangenome analysis. Although the numbers of different gene family classes were slightly different, the main conclusion and trend were highly consistent (Supplementary Figs. 6b and 7 and Supplementary Data 3).

**Population properties, fitness, and functional effects of structural variations in eggplant**
To explore the impact of SVs on genome diversity and function in eggplant, we employed a de novo assembly-based approach[32–34] to identify SVs (≥ 50 bp) from the genome assemblies of 16 diverse eggplant accessions—10 generated in this study and 6 from previous studies[12–15]—together with genome assemblies from two outgroup species, *S. insanum* and *S. violaceum*. By comparing the other eggplant genome assemblies against the reference genome 'S076', we identified four types of SVs: insertion (INS), deletion (DEL), translocation (TRA), and inversion (INV), from the pairwise whole genome alignments (see Methods). A cumulative total of 156,540 SVs were identified, with counts per assembly ranging from 5,219 to 23,771, which were fairly consistent with their respective levels of divergence from the reference (Fig. 3a and Supplementary Data 4). We then merged SVs by different types across genomes, resulting in a total of 76,481 uniquely located calls (Fig. 3a), with nearly 60% being accession-specific (see examples of INS and DEL in Fig. 3b). As anticipated, smaller variants tended to be more common than larger ones, as shown by the size distribution of deletions and insertions (Supplementary Fig. 8). This distribution exhibited a peak around 12 kb, likely reflecting the length of long terminal repeat (LTR)

retrotransposons (Supplementary Fig. 8). We also observed 15 large (near or larger than 5 Mb) inversions segregating in the eggplant population, which we further confirmed with synteny or Hi-C maps (Supplementary Data 5 and Supplementary Figs. 9 and 10). Similar to other plant genomes[35], the most abundant SVs in eggplant were caused by insertions of transposable elements, followed by the contraction or expansion of dinucleotide microsatellites (e.g., 'ATAT'), tandem duplications, and, finally, gene-related and complex variations (Fig. 3c). We examined the frequencies of SVs across our dataset of 16 genomes. Near 81.5% of deletions and insertions were detected in only one or two assemblies, with larger ones being generally less frequent, suggesting SVs segregate at low frequencies within population (Supplementary Fig. 11a). To further explore these frequency patterns, we calculated the minor allele frequency for each SV type, excluding translocations due to their low number. We also combined INSs and DELs into a single category because of their unknown ancestral state. We further classified them into subgroups based on their overlap with different genomic features, such as LTR retrotransposons (LTR-TE), DNA transposons (DNA-TE), 'TATA' motif satellite sequences, or genic regions. The minor allele frequency spectra (AFS) confirmed the low frequency for SVs as well as other variants (i.e., single nucleotide variations/SNVs and small InDels) but it revealed no compelling pattern that suggested SVs are segregating at significantly different frequencies than SNVs and InDels (Supplementary Fig. 11b). We therefore employed an outgroup species, *S. insanum*, to ascertain the ancestral state and deduced the ancestral status for ~74.3% (56,486/76,002) of deletions and insertions (Supplementary Data 6). The derived allele frequency spectrum (dAFS) showed that SVs segregate at significantly lower frequencies than either synonymous or nonsynonymous SNPs, suggesting that SVs are, on average, more deleterious to fitness than SNPs (Fig. 3d). A similar pattern was also observed when using *S. violaceum* as the outgroup, although ancestral states could be inferred for a smaller proportion of sites (28.1%) (Supplementary Fig. 11c). Among the SV classes examined, inversions segregated at the lowest frequencies, consistent with previous reports[36–38].

We also investigated the effects of SVs on gene function. Overall, SVs were significantly depleted in genic regions, and only a small fraction directly overlapped coding sequences (Fig. 3e), where they may disrupt gene structure. Among the various types of SVs, LTR-TE insertions exhibited a farther distance to genes compared to other types of SVs (Fig. 3f), likely owing to insertional biases or a larger effect on gene expression. To investigate the broad effects of SVs on gene expression, we compared the expression profiles across five tissues (i.e., leaf, root, stem, flower, and fruit) between two sets of genes: those with overlapping SVs and those without. We measured expression variation using the Pearson Correlation Coefficient between these two gene sets with RNA-seq data from eight accessions. The set of genes without SVs overlapping had significantly lower expression variance compared to the set of genes with overlapping SVs ($p < 2.2e$-$16$, Wilcoxon test; Fig. 3g and Supplementary Fig. 12a). The set of genes with SVs directly overlapping their exons generally showed higher expression variance than genes with SVs in other genic contexts, with this

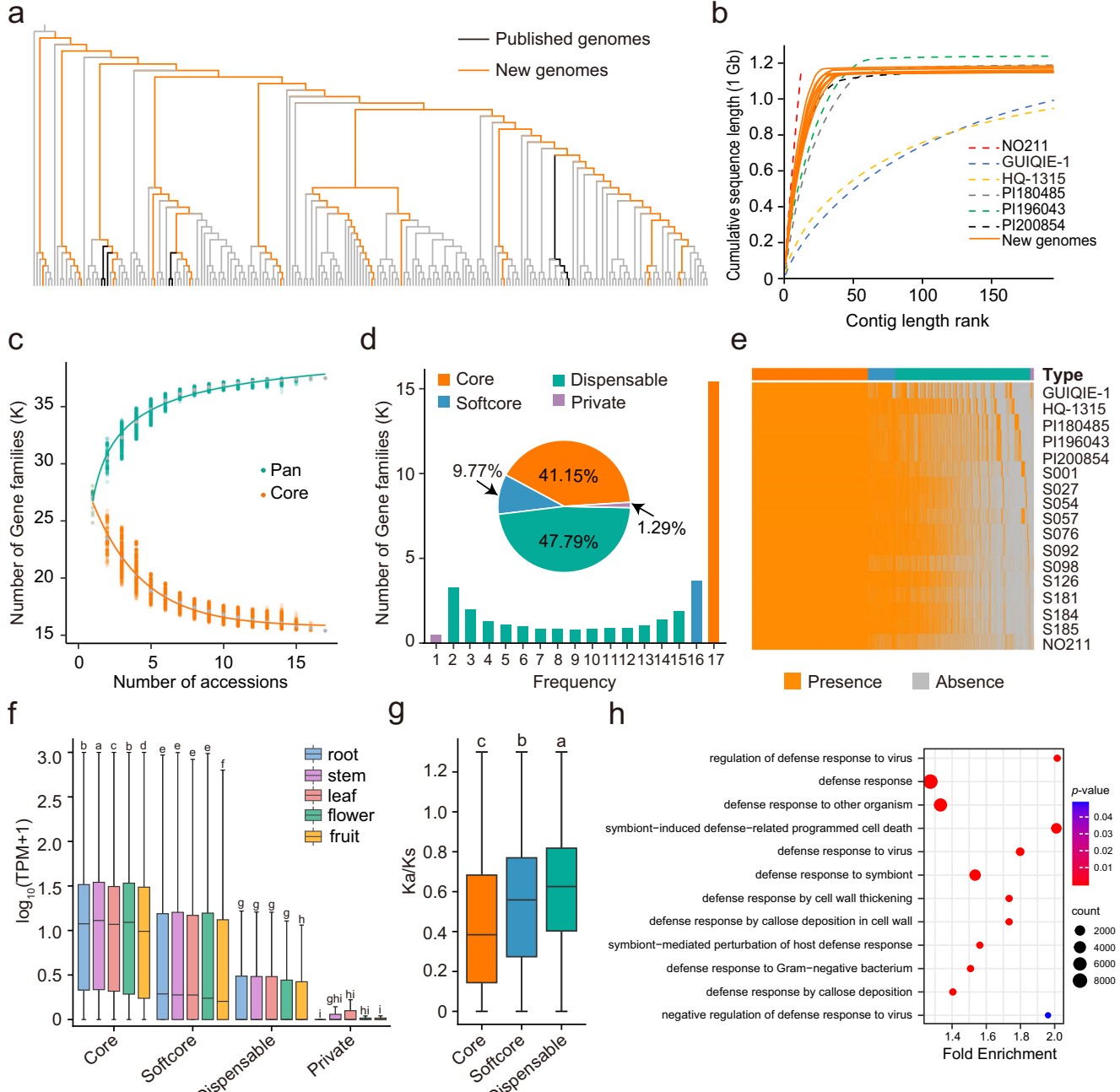

**Fig. 2 | Pangenome analysis of 17 eggplant accessions. a** A phylogenetic tree of 230 accessions based on 4-fold degenerate sites using the neighbor-joining method. Orange, black, and gray branches indicate the 11 newly assembled genomes, six published genomes, and other accessions, respectively. **b** Comparison of sequence contiguity among 17 genomes using the 195 longest contigs (except for NO211). PI196043 only has 195 contigs. **c** Number of pan and core gene families considering 1-17 genomes. **d** Frequencies and proportions of core, softcore, dispensable, and private gene families. **e** Compositions of each of the 17 genomes. Colors are explained in (**d**). Every row represents an accession. **f** Comparison of gene expressions in five organs among core, softcore, dispensable, and private genes. Gene expressions were calculated for the 11 de novo genomes due to availability of transcriptome data. **g** Comparison of $K_a/K_s$ values among core ($n$ = 903,921), softcore ($n$ = 187,240), and dispensable ($n$ = 268,973) genes. Only

ortholog gene pairs were compared. A different version considering both ortholog and paralog gene pairs were provided in Supplementary Fig. 6. In boxplots (**f**, **g**) the lower and upper edges of the box represent the first and third quartiles, respectively, and the central line indicates the median. The whiskers extend to the smallest and largest values within 1.5 × the interquartile range (IQR). Different letters indicate significant differences (Tamhane's T2 test, two sided, $p$ < 0.05). **h** Gene Ontology (GO) terms significantly enriched in dispensable gene families and associated with defense. Circle sizes indicate different GO counts (number of genes associated with a GO in the dispensable genes); fold enrichment score was calculated as the percentage of a GO in dispensable genes relative to that in all genes; circle colors indicate different $p$ values in -$\log_{10}$ scale (hypergeometric test, one sided). Source data are provided as a Source Data file.

difference being significant in most comparisons (Supplementary Fig. 12b).

Through analysis of RNA-seq data from 111 eggplant accessions, we identified 93 SV-affected genes whose fruit expression levels

were significantly altered (Supplementary Data 7). For example, a 418-bp deletion upstream of *evm.model.Chr02.3134* (Fig. 3h), which encodes a WAT1-related protein, and a 10,296-bp deletion upstream of *evm.model.Chr02.4128* (Fig. 3i), which encodes a BTB/POZ- and

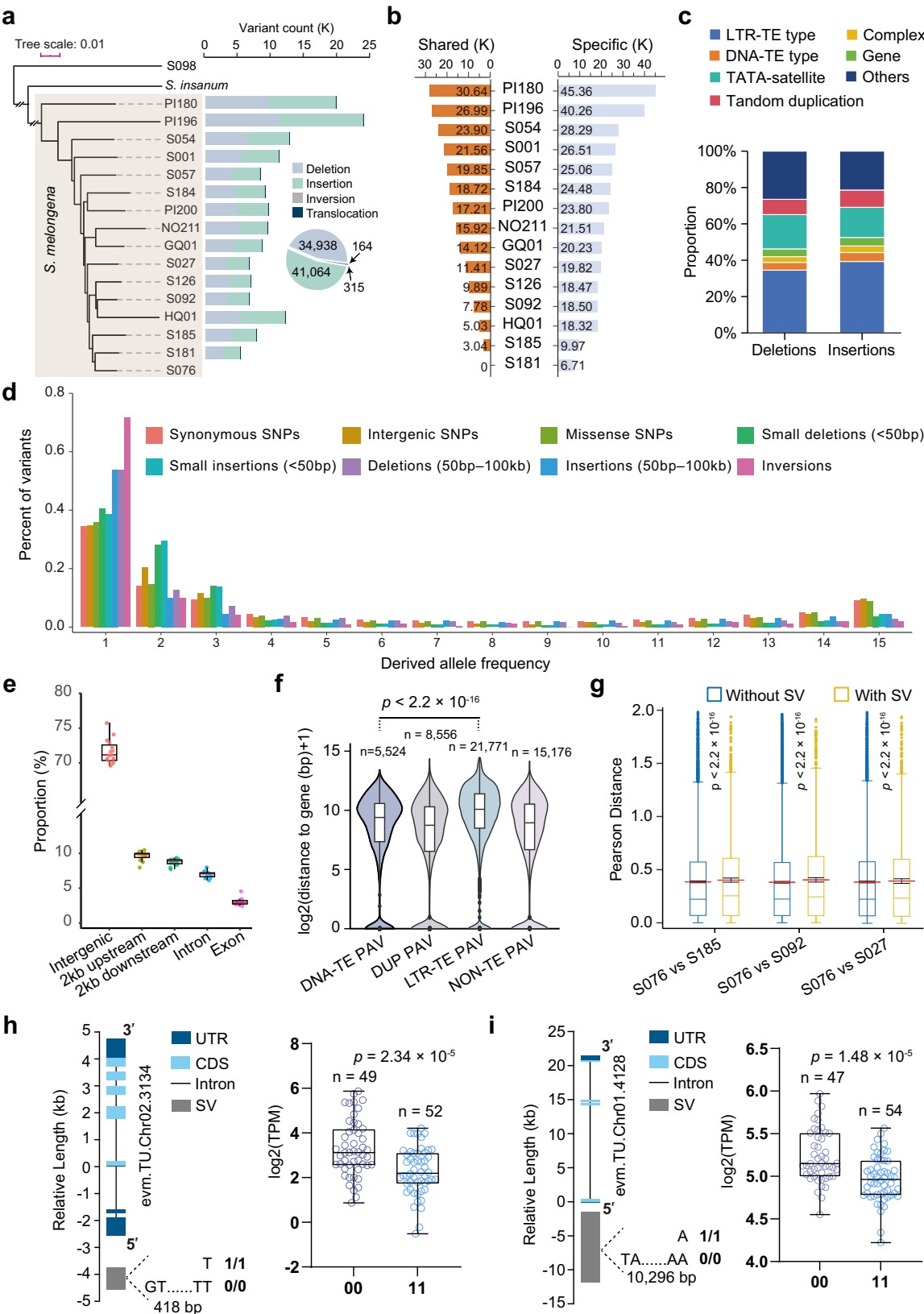

MATH-domain–containing protein, were each associated with significantly reduced expression in accessions carrying the deletion. Collectively, these results establish an integrated SV–transcriptome resource for eggplant, enabling systematic dissection of genomic diversity and the functional and fitness effects of structural variation.

## Graph-based sub-pangenome of eggplant

To further capture SV diversity in cultivated eggplant, we generated an additional callset comprising 128,575 deletions and 11,925 insertions (≥ 50 bp) from resequencing data of 226 eggplant accessions (198 in this study and 28 from previous works), using three complementary short-read-based approaches (see Methods). By integrating this callset with

**Fig. 3 | Structural variations in eggplant. a** Numbers of SVs identified between each query genome and the reference S076. The phylogenetic tree displaying on the left was constructed using 327,235 4-fold degenerate sites. **b** The number of shared and specific (i.e., unique to a single accession) SVs was analyzed as additional samples were included. The order in which samples were added follows the level of divergence. **c** The percentage of different SV types. **d** The derived allele frequency for different kinds of genomic variations. *S. insanum* was used as the outgroup to infer the ancestral state. **e** Proportion of SVs that overlap with different genomic contexts. The boxplots are generated based on SVs detected from 15 genomes relative to the reference S076 (*n* = 15). **f** Distance of different types of presence-absence variants (PAVs) to genes. The number of PAVs in each type is shown in the corresponding boxplot. **g** Expression variation was measured by the Pearson distance between two gene sets, with at least one SV within 2 kb versus without. A total of 1742 (with SV) vs 9504 (without SV) genes, 1631 vs 9792 genes, and 1489 vs 10,041 genes were included for the S076 vs S185, S092, and S027 comparisons, respectively. A 10,000-bootstrap resampling procedure was used to estimate the mean and its 95% confidence interval; the red line indicates the mean. **h** Expression of *evm.model.Chr02.3134* in accessions with (1/1) or without the SV (0/0), based on RNA-seq data. **i** Expression of *evm.model.Chr01.4128* in accessions with (1/1) or without the SV (0/0), based on RNA-seq data. In boxplots (**e**–**i**), the interquartile range is shown as the lower and upper edges of the boxes, respectively, and the central lines stand for the median. The whiskers extend to the largest and smallest values within the 1.5 × IQR in (**e**–**g**), and to the largest and smallest values in (**h**–**i**). Significant levels were determined using a two-sided Wilcoxon test in (**e**–**g**), and an unpaired two-sided *t*-test in (**h**) and (**i**). Source data are provided as a Source Data file.

our assembly-based callset, we obtained a non-redundant set of 187,412 SVs. Using this unified SV set, we applied the vg toolkit[39] to build a pangenome graph with S076 as the reference backbone. We then used the vg pipeline to genotype graph-embedded SVs (i.e., deletions and insertions) across all accessions using short-read data. The resulting genotypes yielded a derived allele frequency spectrum (dAFS) similar to that obtained from the smaller set of assembled genomes, confirming that most SVs segregate at low frequency in the population (Supplementary Fig. 11d). Because our accessions primarily represent major regions of East and Southeast Asia and Europe, with limited sampling from other regions (for example, India, Korea, and Japan), we refer to this resource as a sub-pangenome.

Given that the pangenome graph above was built from a reference-based variant callset and may therefore be subject to reference bias, we additionally constructed a reference-free pangenome graph using the PanGenome Graph Builder (PGGB) pipeline[40] to capture potentially missing genetic diversity. The PGGB graph, constructed from the genome assemblies of 16 cultivated eggplant accessions, contains 31.8 million nodes and 43.9 million edges, corresponding to a mean node degree (i.e., the number of edges connected to a node) of 1.4. The total sequence length represented in the graph (the sum of node lengths) is 1.96 Gb. Deconstructing the graph using S076 as the reference backbone, yielding 31,793 structural variants (SVs; ≥ 50 bp), 2,045,873 InDels, and 6,939,066 SNPs. This reference-free pangenome graph provides an important complement to the reference-anchored graph and establishes a foundation for dissecting the functional effects of SVs on phenotypic variation in eggplant.

## A large inversion (~12.4 Mb) strongly associates with fruit color in eggplant

The fruit color of eggplant is a major quality attribute that significantly influences commercial value. We initiated a GWAS across our sample of 201 cultivated eggplant accessions that exhibit a broad spectrum of fruit colors. For simplicity, we classified the color pattern into two groups: purple (*n* = 148) versus non-purple (i.e., green and white, *n* = 53). Variants derived from the PGGB pipeline largely overlapped those identified by reference based methods. For example, nearly 97.61% of PGGB SNPs were present within or near 2 kb of GATK SNPs (Supplementary Data 8). We therefore first performed GWAS using filtered GATK-derived SNPs and InDels, together with reference-based vg derived SVs, which consistently identified a major association signal spanning a large genomic segment on chromosome 10 (Fig. 4a and Supplementary Fig. 13). In addition, we detected multiple significant loci across the genome, including regions overlapping known anthocyanin pathway genes such as *SmANS*[41], *SmelAAT*[42], *Sm3GT*[43,44], *SmMYB1*[19,45,46], and *SmPAL*[43,47] (Fig. 4a and Supplementary Data 9). Several of these loci were also recovered when GWAS was performed using the reference-free PGGB derived variants, although fewer significant loci were detected, likely due to the smaller variant set (Supplementary Fig. 14 and Supplementary Data 10). Together, these results indicate that fruit color is influenced by multiple loci in eggplant, with the chromosome 10 locus showing a particularly pronounced effect in our panel.

By focusing on the genomic region that harbors the most significant signal on Chr10, we found significant loci spanning over a genomic segment >10 Mb (Fig. 4a). This segment contains four previously identified genes reported to be associated with anthocyanin synthesis in eggplant[19,43,48]. Interestingly, this region coincides with a 12.4 Mb inversion segregating within the population, previously identified in our synteny analysis (Supplementary Fig. 9). The significant loci are enriched at the two breakpoints of this inversion (Fig. 4b). Moreover, this inversion polymorphism shows a strong correlation with purple and non-purple coloring across 14 genome assemblies (Fig. 4c). Using *S. violaceum* 'S098' as the outgroup, we inferred this inversion to be derived in accessions with purple fruit color. By focusing on the inversion breakpoints and examining the short-read mapping result with S126 (green fruit without inversion) as the reference, we were able to further genotype this inversion across a total of 227 eggplant accessions (198 in this study, 28 from the previous work, and the 'NO211') (Fig. 4d). A total of 115 accessions harbor the derived inversion; of these, 111 (96.52%) have purple fruit color and four are green (Fig. 4d). Of the four exceptional samples, we found 3 accessions that possess a very close phylogenetic relationship, and contain an 'A/T' SNP leading to a premature stop codon in *SmANS*[19] that functions in anthocyanin biosynthesis. Out of the 112 accessions without the inversion, 55 (49.11%) have purple color and 57 (50.89%) display non-purple color (Fig. 4d). These results suggest that the inversion on chromosome 10 is strongly associated with the purple phenotype; however, if it is causal, it is not the sole determinant of purple coloration in eggplant.

We next attempted to examine the functional impact of this inversion on color formation by examining the expression of genes within and surrounding the inverted region. We focused on a total of 1178 genes annotated within a target region, which included the inverted region as well as 500 kb upstream and downstream of both breakpoints. We compared RNA-seq data from fruit between samples that were polymorphic for the inversion, including 55 samples with the inversion and 55 samples without the inversion. Of the 1178 genes, we identified only one gene, *SmMYB1*, which exhibited a significantly different expression pattern (*p* ≤ 0.05, fold change ≥ 1.5, and TPM ≥ 10) and is located at the inversion breakpoint (Fig. 4e). This significant difference was further validated by qRT-PCR in 62 eggplant accessions (23 with the inversion and 39 without) (Supplementary Fig. 15a). *SmMYB1* has previously been shown to play a role in anthocyanin synthesis in eggplant[19]. A 6-bp deletion in its coding region was found exclusively in eggplant accessions with green and white fruit colors (*n* = 26) across the resequencing samples (Fig. 4f). In contrast, those without this deletion were predominantly purple (162/193, 83.94%) (Fig. 4g and Supplementary Data 11). Moreover, the qRT-PCR result showed that eggplant accessions lacking this deletion exhibited significantly higher *SmMYB1* expression than those with the deletion (Supplementary Fig. 15b). Therefore, *SmMYB1* might be a strong

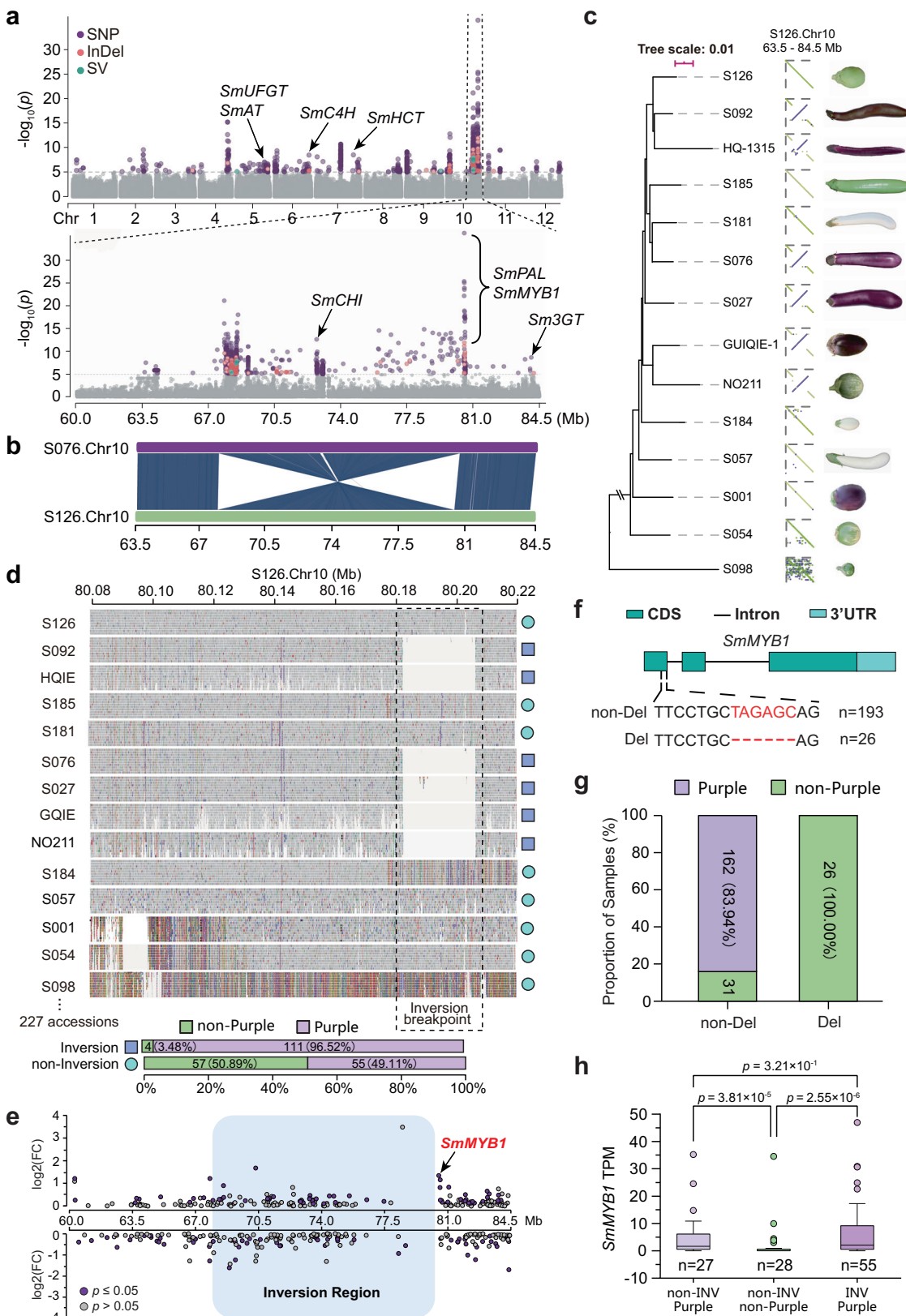

candidate for interacting with the inversion to influence the formation of the purple color.

To further investigate the causal relationship between the inversion and the expression of *SmMYB1*, as well as its subsequent effect on the purple color pattern, we categorized the eggplant accessions with available fruit RNA-seq data into three groups: purple without inversion ($n = 27$), non-purple without inversion ($n = 28$), and purple with inversion ($n = 55$). Pairwise comparisons of *SmMYB1* expression revealed that differential expression of *SmMYB1* was observed only between groups with different fruit colors, but not between groups with or without the inversion (Fig. 4h). This result therefore indicates that the inversion is not directly involved in the expression change of

**Fig. 4 | A high frequency large inversion (~12.4 Mb) is strongly associated with fruit color in eggplant. a** Genome-wide association study (GWAS) results highlight a pronounced signal on chromosome 10 linked to fruit color. Several known anthocyanin biosynthesis pathway-related genes overlap with these GWAS signals, refer to Supplementary Data 9 for more details. *P* values were calculated using a two-sided Wald test based on a linear mixed model. **b** The GWAS signals on chromosome 10 coincide with a large inversion of ~12.4 Mb. **c** A high frequency of this inversion is observed in eggplant accessions with genome assemblies, particularly those exhibiting purple fruit color. Using 'S098' as the outgroup, it is inferred that the inversion occurred in purple varieties. **d** This inversion was further genotyped across 227 eggplant accessions using short-read mapping results. Examining the breakpoint region with the Integrative Genomics Viewer (IGV) revealed a large deletion associated with this inversion. **e** Expression profiles within this inversion region (highlighted in shadow) were compared between samples with (55 individuals) and without the inversion (55 individuals). The significantly differentially expressed gene, *SmMYB1*, is indicated with an arrow. **f** Genotyping of a 6-bp deletion in *SmMYB1* coding region across 219 eggplant accessions. **g** The proportion of eggplant accessions with purple and non-purple fruit color is compared between groups with and without the 6-bp deletion. **h** RNA-seq data from 110 eggplant accessions reveal that differential expression of *SmMYB1* is observed only between groups with different fruit colors, and not between groups with or without the inversion. In boxplots, the lower and upper edges of the box represent the first and third quartiles, respectively, and the central line indicates the median. The whiskers extend to the smallest and largest values within 1.5 × IQR. The *p* values above the bars indicate significantly different values ($p < 0.05$) calculated using unpaired *t*-test (two sided). Source data are provided as a Source Data file.

*SmMYB1*, raising the question of how this inversion is tightly linked to the purple color of the fruit in eggplant.

## The 12.4 Mb inversion linked with reduced genetic diversity and extended haplotype block in eggplant population

Inversions play a role in a wide range of biological processes and their maintenance within populations can be attributed to intricate evolutionary forces such as local adaptation, balancing selection, and natural selection. Among the 15 large inversions (> 5 Mb) (Supplementary Data 5) we identified a 12.4 Mb inversion on chromosome 10 that stood out due to its high allele frequency (50.44%) within the population. This observation presents an excellent opportunity to explore how large chromosomal rearrangements can be sustained in domesticated populations by tracing the evolutionary trajectory of the inversion.

Our study revealed that this inversion occurred predominantly among eggplant accessions from China from groups C3 and C4, with only a handful of instances (n = 5) occurring in samples from C1 representing other regions (Fig. 5a). This suggests that this inversion arose or has risen to high frequency during the selective breeding of eggplant in China, although it remains present globally in eggplant germplasm. To explore the population properties of this inversion and its potential effects on genomic diversity, we classified the 226 eggplant samples into two groups: one with samples carrying the inversion (n = 114) and the other presenting the ancestral status (n = 112). Our analysis of nucleotide diversity (π) on chromosome 10 revealed that the ancestral group (i.e., without the inversion) exhibited higher genetic diversity across the entire chromosome—approximately twice that of the inversion group. This difference was even more pronounced within the inversion region and its surrounding breakpoints (Fig. 5b). We also observed a significant increase in the *Fst* value (a measure of genetic differentiation) across the inverted region between the two groups (Fig. 5c). Furthermore, Tajima's D values, which can be an indicator of selection, showed a greater difference in this inverted region between the two groups. Collectively, these findings suggest that this inversion is associated with a substantial reduction of genetic diversity in the eggplant population.

We observed that the 12.4 Mb inversion is linked to a large haplotype encompassing the entire inverted region (Fig. 5e). A phylogenetic tree constructed using SNPs from this inversion region reveals that nearly all eggplant accessions carry the inversion cluster together, forming a distinct group at the bottom of the tree. This suggests a single haplotype among these accessions, likely resulting from a recent and singular origin. Additionally, we identified a large linkage disequilibrium (LD) block of ~6.0 Mb within this inversion (Fig. 5e), indicating that recombination events in this region are significantly suppressed. The suppression of recombination within inversions is well-documented; inversions can inhibit recombination between heterokaryotypes, leading to the accumulation of genetic changes and the establishment of divergent haplotype blocks. To further examine the effect of this inversion on LD patterns, we plotted LD for each 10 Mb window downstream, upstream, and spanning the inversion region.

We observed substantial LD differences among these windows, suggesting a significant impact of the inversion on local LD patterns (Fig. 5f). Considering these findings alongside the strong association of this inversion with fruit color, we propose that its high frequency in the population is likely due to a de novo origin followed by artificial selection during the cultivation and breeding of eggplant.

## Genomic loci associated with bacterial wilt resistance

Just as fruit color is a crucial agronomic trait, so is the ability to resist disease, particularly bacterial wilt (BW). To identify genomic regions associated with BW resistance, we planted 197 accessions in four batches (three batches in 2022, one batch in 2023), and scored every seedling of these accessions as resistant or susceptible (dead or wilt) consecutively for 5 weeks after infection with *Ralstonia solanacearum* at the 4-5 true leaf seedling stage (Supplementary Data 12). The incidence rate for each accession was calculated as the number of dead or wilt seedlings divided by the total number of seedlings (at least 20 for each accession). The incidence rates ranged from 0 - 100% (Fig. 6a, b). Similarly, we performed SNP-based GWAS and InDel-based GWAS using the filtered GATK callsets, and SV-based GWAS using the reference based vg derived SVs, for each of the four sets of phenotypic data. Strong association signals were primarily identified on Chr02, Chr03, Chr04, and Chr05 (Fig. 6c; Supplementary Figs. 16 and 17; and Supplementary Data 13) for SNP-GWAS and InDel-GWAS, and these were identified consistently across datasets. Strikingly, the top associated SNPs on Chr04 were located within a gene, *evm.model.Chr04.2518*, homologous to *CYP82D47* and *CYP82A3*, which confer resistance to powdery mildew in cucumber and to biotic/abiotic stresses in soybean[49,50]. Importantly, we identified a SNP located within the first exon of this gene, leading to a premature stop codon that reduced the protein length to less than a half. As a result, the gene product is predicted to lack most of the conserved domain of cytochrome P450 family protein (Fig. 6d). The BW incidence rates of accessions with a TGG genotype were significantly lower than those with a TGA (stop codon) genotype ($p = 4.90 \times 10^{-11}$, two-sided *t*-test).

By searching candidate genes on Chr05, we identified that the top associated SNPs and InDels were directly located within a gene, *evm.model.Chr05.2618*, which was truncated and not expressed. This truncated gene was homologous to *EPS1*, which controls salicylic acid (SA) biosynthesis[51,52]. This gene region turned out to also harbor a *Roq1-like* homolog (directly adjacent to the truncated *EPS1* homolog) after GSAman correction as mentioned below. This misannotation was likely due to the erroneous fusion of two neighboring genes. Consistently, SV-GWAS identified two SV association signals (Chr05:84617337 and Chr05:84619808) on Chr05 (Supplementary Fig. 18), which were also located within the above *evm.model.Chr05.2618* region (Supplementary Data 13). Interestingly, analysis of *EPS1* homologs based on the gene family-based pangenome result revealed substantial variation in its copy numbers across the 11 genomes examined, leading us to speculate that a single reference genome may not reveal the whole

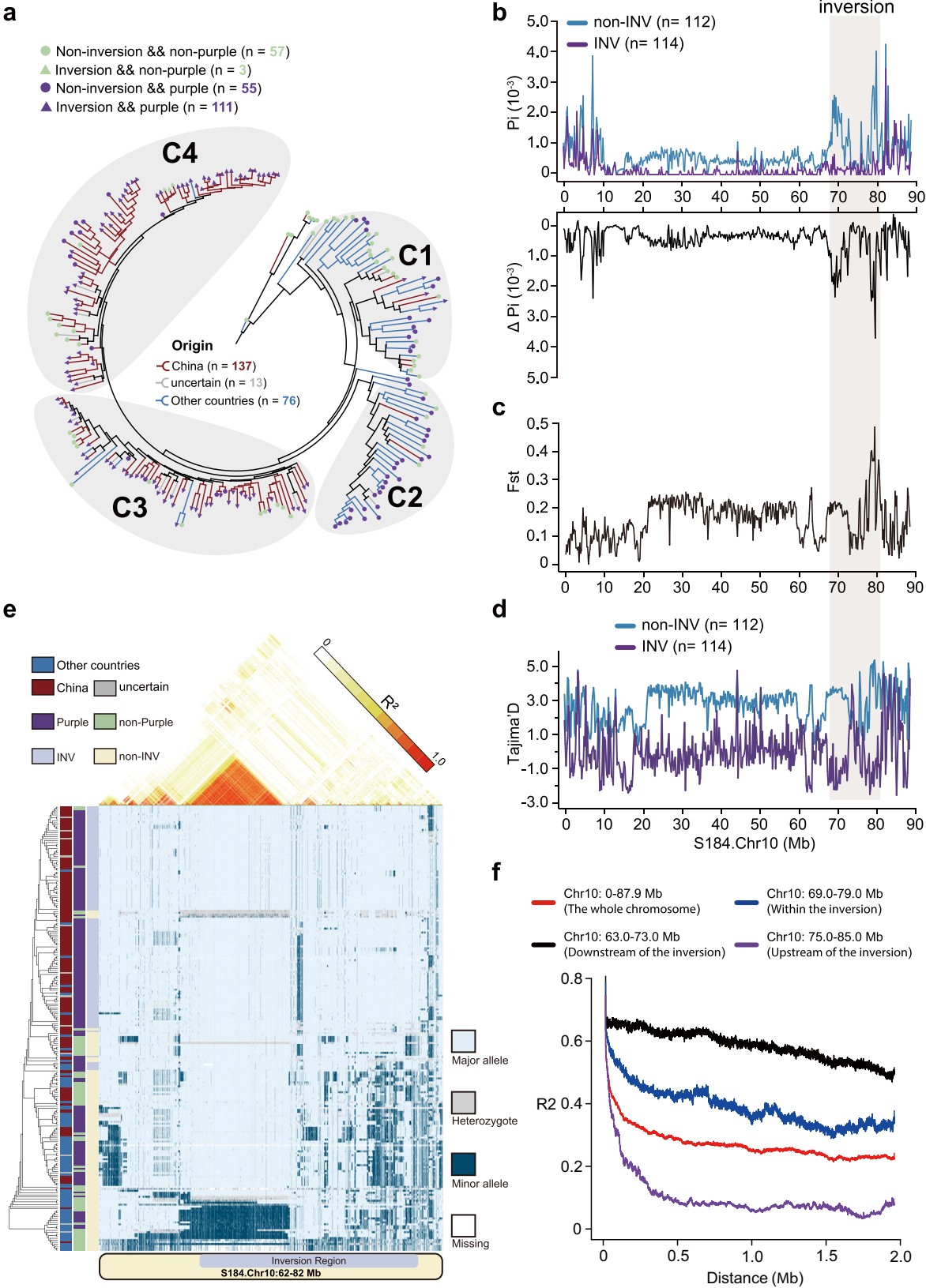

**Fig. 5 | Genomic diversity reduction and extended haplotype across the 12.4 Mb inversion region. a** Distribution of the 12.4 Mb inversion across 226 eggplant accessions. **b** Genomic diversity (π) along chromosome 10 comparing individuals with and without the inversion, with the delta pi between groups displayed below. **c** Population differentiation (*Fst*) indicating genetic divergence between populations. **d** Tajima's D, a measure of genetic variation, along chromosome 10. **e** Haplotype and linkage disequilibrium (LD) patterns across the entire inversion region. **f** LD decay across the whole chromosome and within the left, middle, and right regions of the inversion, highlighting substantial variation in LD across the inversion region. Source data are provided as a Source Data file.

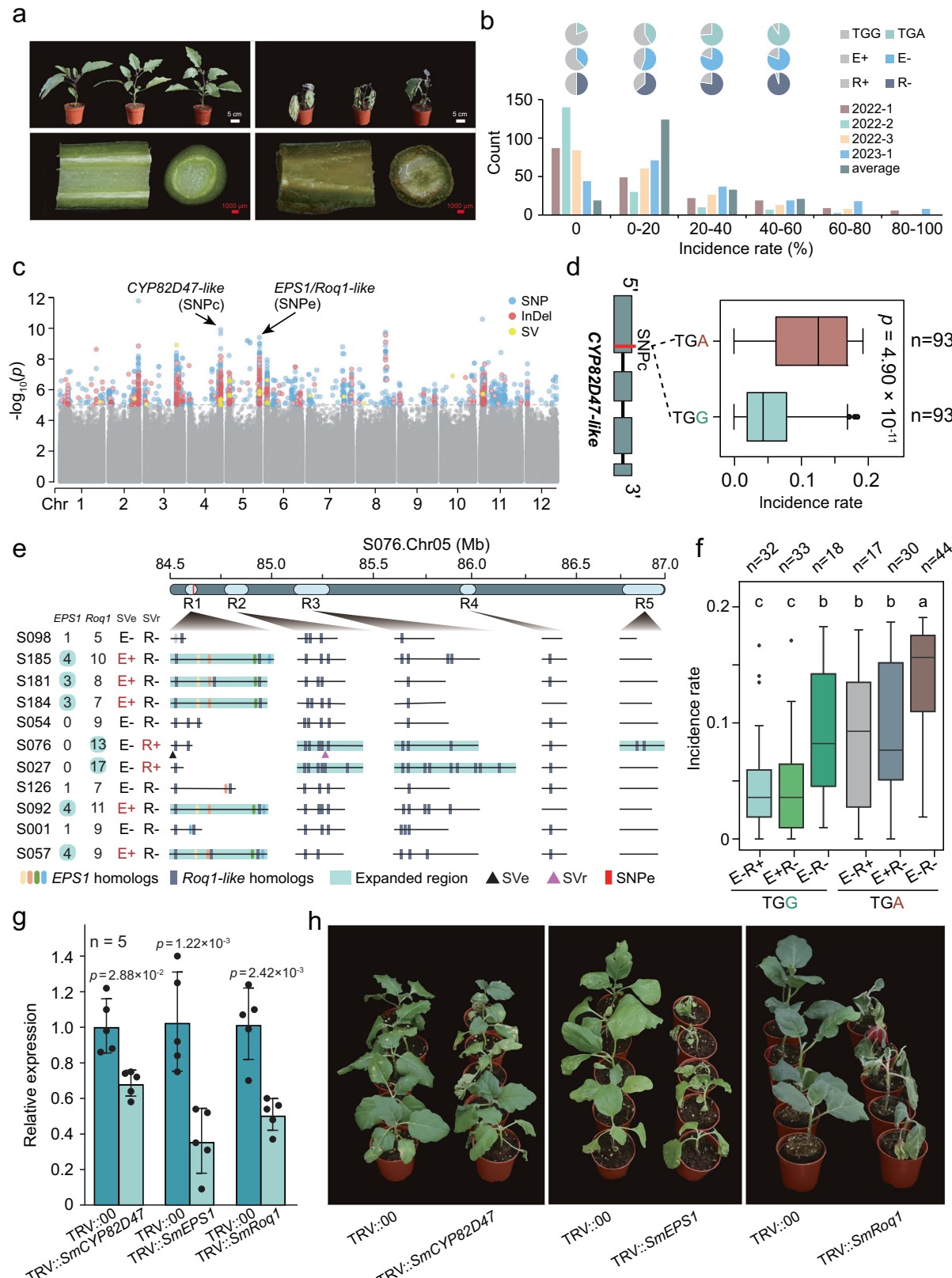

picture of this region. Similarly, both SNP- and InDel-based GWAS analysis with another accession 'S126' as a reference revealed that the majority of top association signals were located within the homolog of *EPS1*, *evm.model.Chr05.2786*, as well as within a *Roq1-like* gene, *evm.-model.Chr05.2787*, encoding putative disease resistance proteins (Supplementary Data 13). Importantly, *Roq1* was previously reported to

confer resistance to three bacterial pathogens, including *Ralstonia* in tomato[53].

Although SNP-, InDel-, and SV-GWAS signals all converged on this region, the specific causal variants or genes responsible for BW resistance or susceptibility remained unresolved. In particular, the SNPs and InDels within these genes did not show strong effects. This

**Fig. 6 | Candidate loci associated with bacterial wilt resistance obtained from genome-wide association study. a** Phenotype of whole plant and stem sections after bacterial wilt pathogen infection for resistant and susceptible plants. **b** Frequency distribution of bacterial wilt disease incidence rates of ~200 accessions for four batches and the averaged rates across two years. Pi charts indicate proportions of different genotypes at the three markers described in (**d**, **e**). **c** Manhattan plot of GWAS analysis. The horizontal dashed line marks the significance threshold ($-\log_{10}(p) \geq 5$). Two arrows point to candidate genes (SNPs) on Chr04 and Chr05. *P* values were calculated using a two-sided Wald test based on a linear mixed model. **d** A candidate gene on Chr04 homologous to *CYP82D47*. The SNP (G to A) leads to a premature stop codon. Accessions with the G/G genotype show significantly lower disease incidence than those with A/A genotype. **e** A candidate genomic region with strong association signals on Chr05, with varying numbers of *EPS1* (0 - 4) and *Roq1-like* (5 - 17) homologs across 11 assemblies in five

sub-regions (R1 - R5). Marker SVe (4,956 bp Indel) distinguishes accessions with many (E + , 3 - 4) versus fewer (E-, 0 - 1) *EPS1* homologs, while marker SVr (167 bp Indel) differentiates accessions with abundant (R + , ≥13) versus fewer (R-, 5 - 11) *Roq1-like* homologs. **f** Normalized bacterial wilt incidence rates of 174 accessions with different genotype combinations of *CYP82D47*, SVe, and SVr. In boxplots (**d**, **f**), the lower and upper edges of the box represent the first and third quartiles, respectively, and the central line indicates the median. Whiskers extend to the smallest and largest values within 1.5 × IQR. *P* values above bars denote significant differences (**d** two-sided *t*-test, *p* < 0.05), and different letters indicate significance (**f** two-sided Wilcoxon rank-sum test, *p* < 0.05). **g** Relative expression of *SmCYP82D47*, *SmEPS1*, and *SmRoq1* in control (TRV::00) and gene-silenced plants (mean ± SD, *n* = 5); significance was assessed by two-sided Student's *t*-test. **h** Phenotypes of control and gene-silenced plants at 10 days post *R. solanacearum* infection (*n* = 5). Source data are provided as a Source Data file.

prompted a detailed comparative genomic analysis. To obtain a clear landscape of *EPS1* and *Roq1-like* homologs in this area, we manually curated and corrected gene models in these regions of the 11 genomes utilizing transcriptomes of five organ types including roots, stems, leaves, flowers, and fruits, with GSAman (Fig. 6e and Supplementary Data 14). As expected, the number of expressed *EPS1* (0 - 4) and *Roq1-like* (5 - 17) homologs and the lengths of syntenic regions were highly variable across the 11 genomes. To evaluate their potential correlation with BW resistance, we identified and utilized another two SV markers in these regions, with SVe (4,956 bp InDel, Chr05:84599196) differentiating accessions with expanded *EPS1* homologs (≥ 3, E + ) from those with only one or zero *EPS1* homolog (E-), and with SVr (167 bp InDel, Chr05:84828646) differentiating accessions with relatively more *Roq1-like* homologs (≥ 13, R + , like 'S076' and 'S027') from those with less *Roq1-like* homologs (5 - 11, R-, like the remaining accessions). When considered together or in combination with the Chr04 locus (*SmCYP82D47*), the two SV markers were significantly and highly associated with BW resistance (Fig. 6f). By fixing the Chr04 locus as either 'TGG' or 'TGA' genotype, the comparison revealed that accessions with either E-_R+ (indicating less *EPS1s* and expanded *Roq1-like* homologs) or E + _R- (indicating expanded *EPS1s* and less *Roq1-like* homologs) genotype were significantly more resistant to the disease than those with E-_R- (indicating less *EPS1s* and less *Roq1-like* homologs) genotype. Accessions with TGA_E-_R- genotype were the most susceptible to the disease. Other genotypes were not compared due to a small number of available accessions (n < 10). Another approach calculating the normalized mean coverage of *SmEPS1* and *SmRoq1* revealed similar results (Supplementary Fig. 19). The comparison of gene expressions of different *EPS1* homologs revealed that those for genomes with multiple copies showed much higher expressions (Supplementary Figs. 20 and 21). To evaluate the functions of *SmCYP82D47*, *SmEPS1* and *SmRoq1* homologs, we carried out virus-induced gene silencing (VIGS) experiments targeting the *SmCYP82D47* homolog using accession 'S065' (TGG_E-_R-), targeting a conserved fragment of *EPS1* and *Roq1* homologs using accession 'S092' (with TGA_E + _R- genotype) and accession 'S050' (with TGA_E-_R+ genotype), respectively. Results showed that the *SmCYP82D47*-silenced, *SmEPS1*-silenced, and *SmRoq1*-silenced plants displayed typical wilt symptoms at 10 days after infection with *R. solanacearum*, while their control plants showed less or no wilt symptoms (Fig. 6g, h). These results support a model in which *SmCYP82D47*, *SmEPS1*, and *SmRoq1* homologs, perhaps together with additional genes, jointly contribute to BW resistance. Collectively, we identified three major candidate genes or gene types associated with BW resistance, as well as three associated molecular markers, including one SNP on chromosome 4 and two SVs on chromosome 5, that can be directly applied in eggplant breeding for BW resistance. These three candidate genes, *SmCYP82D47*, *SmEPS1*, and *SmRoq1*, were also among the significant association signals in the GWAS using PGGB-derived variants (Supplementary Fig. 14 and Supplementary Data 10 and 15).

## Discussion

Pangenomics has recently emerged as a powerful tool for exploring genome diversity, origin and domestication, as well as for genetic analyses of important phenotypic traits in crop species[54–57]. Eggplant is among the top highest important horticultural crops, yet application of its genomic resources in breeding lags behind those of other Solanaceae crops, like tomato[58], potato[15,59], and pepper[33,60–63]. To fill this gap, we have generated and collected resequencing data for a diversity panel of 226 eggplant accessions, de novo assembled 11 of them, and also included eggplant genomes from previous works[9–14], to represent so far the most comprehensive sub-pangenome resource in eggplant, with a primary focus on capturing its genetic diversity from East and Southeast Asia. We have demonstrated that this sub-pangenome dataset can effectively identify large hidden genomic variations associated with crucial traits, including a 12.4 Mb inversion linked to fruit color. Additionally, this sub-pangenome coupled with other genomic resources have facilitated the identification of complex loci associated with bacterial wilt (BW) resistance, a serious disease in eggplant and other Solanaceae crops. This sub-pangenome resource expands the repertoire of sequence variations accessible for genetic analysis and will further enhance breeding efforts in eggplant. It is also expected to have utility beyond the scope of this study, potentially benefiting broader population genomics research and trait association studies in eggplant.

Although the origin and domestication history of eggplant remains somewhat ambiguous, some evidence suggests that eggplant was independently domesticated in multiple locations, particularly in Southeast Asia and India[7]. Our germplasm for this study was largely derived from Southeast Asia (*n* = 35) and China (*n* = 137), mostly belonging to Asia which contributes ~93.5% of the global production of eggplant (FAO, 2021-2023). The population structure points to a strong correlation between the genetic relationships and geographic origins in eggplant (Fig. 1e), consistent with previous studies[7,64]. The samples from group C1 and C2, mainly from Southeast Asia, South America, and Europe displayed significantly higher genomic diversity (π) than the samples from China (groups C3 and C4), likely reflecting that Southeast Asia is a major center of eggplant domestication. Alternatively, the differences in diversity could reflect genetic bottlenecks caused by breeding during eggplant improvement in regions of China[1,2,7]. One insight from our analyses is that it may be important to include genetic materials from group C1 and C2 for breeding programs in China to expand the genetic base, both to meet the diverse consumer habits and to aid better adaptation to changing environments. Considering that our samples are primarily from China and Southeast Asia, future studies incorporating more samples from diverse origins, particularly from India, the Middle East, Africa, Korea and Japan, will be essential to expand our comprehensive understanding of eggplant domestication, migration routes, and genomic diversity.

We investigated the gene space by performing a gene-based pangenome analysis utilizing 17 genomes, 11 of which were produced

for this study. The quality of our de novo assemblies was clear from two observations. First, the N50 lengths of assembled genomes in this study were higher than[12,13] or similar to[15] that of previously published eggplant genomes (Fig. 2b). Second, BUSCO completeness scores all exceeded 98.70%. Using these high-quality genomes along with published ones, we have constructed a gene-based eggplant sub-pangenome comprising 37,435 gene families, including 15,406 core gene families. Both numbers are lower than those reported in the previously published pangenomes of tomato (40,457 pangene families, 21,847 core gene families)[28] and pepper (42,972 pangene families, 19,662 core gene families)[65], possibly due to species differences or sampling issues. Nonetheless, these pangene families may well represent the genetic diversity and gene repertoire of *S. melongena*, since the number of pangene families neared a plateau (Fig. 2c). However, the number of pan genes continued to increase without reaching a plateau when we included genes unassigned by OrthoFinder (Supplementary Fig. 22). The core gene families are conserved in all sampled genomes, indicating they are essential for the basic biology of *S. melongena* and probably mainly play housekeeping functions. A considerable proportion (47.79%) of the pan-gene families were dispensable (Fig. 2d), highlighting the genetic diversity of eggplant germplasms and a potential source for genetic improvements. This was especially demonstrated by the scenario of *SmEPS1*, in which case its presence or absence (or copy number) influences the levels of resistance to BW in eggplant. To rule out potential biases in gene annotations between the published genomes and our de novo sequenced genomes, we also performed a pangenome analysis using a subset of 11 genomes, which yielded similar results.

SVs play a profound role in genome evolution and phenotypic diversity in plants[38,66,67]. To ensure the accuracy and completeness of SV identification in our samples, we employed two complementary strategies. First, we utilized an assembly-versus-assembly pipeline to detect SVs from the chromosomal-scale assemblies of 16 diverse eggplant accessions and two outgroup species, *S. insanum* and *S. violaceum*. These genomes were selected from major branches of the eggplant phylogenetic tree (Fig. 2a, and Supplementary Fig. 4), representing a wide range of genetic diversity. This dataset is expected to capture most high-frequency SVs within the eggplant population that originated from the domestication center in Southeast Asia and has proven particularly effective at identifying insertions and larger SVs, including inversions. Second, we applied short-read-based methods to detect SVs from the resequencing data of 226 eggplant accessions, which is likely to reveal smaller, rarer SVs due to the large sample size. By integrating both datasets, we attempted to build a comprehensive SV catalog for eggplant.

Our investigation of SVs in eggplant reveals features that suggest they generally have deleterious effects, which resemble observations in other crop species[36,38]. SVs are typically found at low frequencies within populations and are less common within gene bodies (Fig. 3c). We identified several large inversion polymorphisms in the cultivated eggplant population, some of which segregate at high frequencies. Inversions can profoundly affect various genomic properties, including recombination, linkage disequilibrium (LD), genome architecture, and gene expression[32,68–70]. Population genetic studies in crop species have shown that inversions often have lower fitness than other mutations, as they usually segregate at lower frequencies[37,38]. Nonetheless, inversions can sometimes reach high frequencies through selection due to their beneficial effects, such as facilitating adaptive mutations, preserving advantageous gene combinations, or being linked to favorable phenotypes[71–74].

One notable example of inversions in our study is a 12.4 Mb inversion on chromosome 10, present in 50.44% of all samples, with a higher frequency of 70.80% (97 out of 137) in samples from China (Supplementary Data 11). Population genomic analyses reveal that individuals carrying this 12.4 Mb inversion on chromosome 10 exhibit reduced overall genomic diversity compared to those without the inversion, which represents the ancestral state. This reduction in diversity, along with the extensive haplotype and linkage disequilibrium (LD) spanning the inversion, suggests a genetic bottleneck in the subgroup with the inversion, likely due to the founder effect during the domestication and breeding history of eggplant. The origin of this 12.4 Mb inversion on chromosome 10 remains unknown. However, artificial selection favoring the purple fruit color may have contributed to its maintenance and increased frequency in China. A model describing the origin and evolution of the inversion and its association with fruit color is depicted in Supplementary Fig. 23.

The color diversity in eggplant fruits is primarily determined by the interaction of two key pigments: anthocyanin and chlorophyll. Anthocyanin, responsible for the purple to black hues, and chlorophyll, which gives green coloration, are controlled by a few dominant genes[19,44,75,76]. Historically, genetic studies have identified three dominant genes (*C*, *P*, and *D*) as central to anthocyanin production in eggplant, as outlined by Tigchelaar *et al.*[77]. Our recent research has provided deeper insights into these genes, identifying the *SmANS* gene on chromosome 8 and *SmMYB1* on chromosome 10 as likely candidates for the *P* and *D* genes related with anthocyanin biosynthesis[19], respectively. In experiments where these genes were silenced, a reduction in anthocyanin production was observed, confirming their role in fruit coloration. Our study also discovered several alleles of the *SmMYB1* gene, including a specific 6-bp deletion in its coding region. This deletion correlates with non-purple fruit, consistent with decreased expression of *SmMYB1* (Supplementary Fig. 15b)[78]. Additionally, a 12.4 Mb inversion was found ~5 kb upstream of *SmMYB1* (Fig. 4c). This inversion is mutually exclusive with the 6-bp deletion and is associated with purple fruit, a finding validated across a large sample of 201 accessions, along with 25 accessions from a previous study[11]. Four exceptional accessions were identified that carried the inversion but exhibited green fruit; three of these can be explained by functional mutations in the structural genes of the anthocyanin pathway. However, the inversion is most likely not the causal mutation directly affecting *SmMYB1*. Instead, it appears to be a closely linked variant associated with the purple fruit phenotype, which likely involves multiple genetic factors. Further studies are needed to explore the underlying mechanisms and the complex genetic determinants of eggplant fruit color.

Bacterial wilt is a devastating disease of eggplant caused by *R. solanacearum*, which also infects other Solanaceous crops and a wide range of plant families[79]. Previous studies using genetic mapping populations (e.g., F$_2$, RILs, and DHs) have identified numerous QTLs and putative resistance genes associated with BW resistance in eggplant (Supplementary Data 16)[23–25,80–82]. However, these findings were obtained within specific or narrow genetic backgrounds. In this study, we addressed this gap by detecting association signals for BW resistance through SNP-, InDel-, and SV-based GWAS analyses across a broad and diverse eggplant germplasm panel. We identified three loci that were significantly associated with BW resistance, likely linked to functional mutations, including a premature stop codon in *SmCYP82D47* and copy number or other unidentified genomic variations in *SmEPS1* and *SmRoq1*. The numbers of *SmEPS1* (0–4) and *SmRoq1* (5–17) homologs varied substantially among genomes, likely because both genes are located near chromosomal ends where recombination rates are high[83]. These three loci appear to confer resistance through distinct mechanisms, potentially involving the salicylic acid (SA) pathway and immune receptor (R gene)–mediated responses. The expression of *CYP82D47* in cucumber is induced by salicylic acid, and multiple SA-related genes are upregulated in *CYP82D47*-overexpressing plants[49].

Although we observed a positive correlation between the copy numbers of *SmEPS1* and *SmRoq1* and bacterial wilt (BW) resistance, it remains unclear whether increased copy number directly enhances

resistance (a gene-dosage effect) or instead reflects linkage with other, as-yet-unidentified causal variants. Our VIGS experiments showed that silencing *SmEPS1* or *SmRoq1* altered BW resistance, supporting a functional involvement of these genes in disease resistance. Because both genes occur as multi-copy families, VIGS may co-silence closely related homologs due to sequence similarity[84]; nevertheless, sequence comparisons indicate that both the VIGS fragments and the qRT–PCR primers match only *SmEPS1* or *SmRoq1* homologs and not other genes (Supplementary Data 17 and Supplementary Fig. 24). Thus, the VIGS phenotypes likely reflect the combined effects of silencing one or a subset of homologs within each gene family, rather than unintended impacts on unrelated loci. Further targeted functional analyses will be required to pinpoint which specific copy or copies contribute to resistance given the pronounced variation in both copy number and predicted protein length among homologs, as exemplified by *SmEPS1* (Supplementary Fig. 20).

Given that multiple loci confer bacterial wilt (BW) resistance in eggplant, elucidating their synergic–epistatic interactions is essential for breeding highly resistant cultivars. Our results indicate that *SmCYP82D47* plays a central role in BW resistance, as when it carries a premature stop codon, even accessions with high copy numbers of *SmEPS1* and *SmRoq1* do not exhibit strong resistance. In contrast, when *SmCYP82D47* is functional, the presence of high copy numbers of either *SmEPS1* or *SmRoq1* homologs-or both-confers strong resistance, whereas accessions lacking these copies remain susceptible. Thus, *SmCYP82D47* exerts the most pronounced effect among the three loci, while *SmEPS1* and *SmRoq1* contribute comparably to BW resistance (Supplementary Fig. 25). The alleles of *SmCYP82D47* (TGA:TGG = 50%:50%), *SmEPS1* (E-:E + = 63.79%:36.21%), and *SmRoq1-like* (R-:R + = 71.84%:28.16%) are all prevalent in global eggplant germplasm (Fig. 6d, f), suggesting that pyramiding the beneficial alleles of these loci into a single cultivar is theoretically feasible through conventional breeding. This strategy could be further facilitated by the molecular markers linked to these loci developed in this study. Moreover, the BW resistance genes identified in eggplant may provide valuable insights into the genetic basis of BW resistance in other Solanaceous crops such as tomato, pepper, and potato.

## Methods

### Plant materials, growth conditions, and phenotyping

A total of 198 eggplant germplasm accessions were obtained from South China Branch of National Genebank for Vegetable Germplasm Resources at Vegetable Research Institute, Guangdong Academy of Agricultural Sciences, Guangzhou, China. This diverse collection included 193 eggplant accessions (*S. melongena*), 4 scarlet eggplant accessions (*S. aethiopicum*), and 1 wild eggplant species (*S. violaceum*), which were collected from 21 provinces of China, Southeast Asia, and South America. To evaluate their phenotype, these accessions were planted at Baiyun Experimental Station of Guangdong Academy of Agricultural Sciences, Guangzhou, China (113.40° N, 23.39° E). Fruit colors were observed in the field for four growing seasons in 2021-2023 and recorded as purple, green, and white. To evaluate the degree of resistance/susceptibility to bacterial wilt, the seedlings at 4-5 true leaf stage maintained in a disease nursery were inoculated with *R. solanacearum* (strain GMI1000, RS742, DG-1-2, 20181102). The roots of seedlings were immersed in a prepared *R. solanacearum* suspension ($10^8$ CFU mL$^{-1}$) for 20 min. Following inoculation, the seedlings were transplanted into the field for subsequent disease development. A total of four batches (three batches each with two completely randomized blocks in 2022, and one batch with three completely randomized blocks in 2023) of plants were conducted for bacterial wilt inoculation. Each seedling was evaluated as resistant (normal, score 1) or susceptible (dead or with wilt symptom, score 0) at five weeks post inoculation during the summers of 2022 and 2023. The incidence rate of each accession was calculated as the number of dead or wilt seedlings divided by the total number of seedlings (at least 20 for each accession per bach).

### Short-read and long-read sequencing

Genomic DNA extraction and library (350 bp) construction were performed by a commercial service (Biomarker Technologies, Beijing, China), which were sequenced using the Illumina NovaSeq 6000 platform (150 bp paired-end) for 198 accessions. For construction of Hi-C libraries, young leaves were collected from 11 selected accessions and immediately put into liquid nitrogen. Then samples were cross-linked with 2% formaldehyde via vacuum infiltration; 2.5 M glycine was added to the mixture to quench the crosslinking reaction. Nuclei were purified, digested with enzyme *Dpn*II, and end-labeled via biotinylation with biotin-14-dCTP. Ligated DNA was sheared into 200-600 bp fragments by sonication, which were end-repaired, A-tailed, and purified. Hi-C sequencing libraries were amplified by PCR (12-14 cycles) and sequenced on Illumina NovaSeq 6000 platform (150 bp paired-end). For long-read sequencing, genomic DNA was extracted from young leaves of the 11 selected accessions and was used to construct PacBio HiFi SMRTbell libraries. The libraries were sequenced on a PacBio Sequel II platform to generate HiFi long reads with the circular consensus sequencing (CCS) mode in the SMRT Link software (v11.0).

### De novo assembly and chromosome anchoring

HiFi long reads were used to generate raw contigs using Hifiasm (v0.16.1)[85] with default parameters. The resulting contigs and Hi-C data were used for chromosome anchoring with ALLHiC (v0.9.8)[86] (--min-REs 50 --maxlinkdensity 3 --NonInformativeRabio 2) and Juicebox[87] with manual adjustment. Then sequence order and direction were adjusted according to the genome collinearity with the published eggplant genome[12] using Mummer (v4.0.0rc1)[88] (parameter: -i 90 -l 5000). To further evaluate the assembly quality, we assessed the gene completeness with BUSCO (v.5.2.2)[89]. Then, we obtained the mapping rates of short-reads to the assemblies using the BWA (v2.2.1)[90].

### Repeat annotation

A strategy combining homology and de novo prediction was used to identify transposable elements (TEs) in the assembled genomes. Tandem Repeats were identified using TRF (v4.09)[91] by ab initio prediction. RepeatMasker (v4.1.0[92]) was used for the homology-based method, employing the Repbase database[93] with default parameters. The ab initio prediction was carried out by LTR_FINDER v1.06[94], RepeatScout (v1.0.5)[95], and RepeatModeler (v2.0.1)[96] with default parameters. Subsequently, all repeat sequences with lengths >100 bp and 'Ns' (gap) <5% constituted the raw TE library, which was combined with Repbase for redundancy removal by UCLUST[97]. The resulting non-redundant TE library was supplied to RepeatMasker for final repeat identification.

### Gene annotation

To annotate protein-coding genes, we integrated homology-based, ab initio, and RNA-Seq assisted approaches to predict gene models. We downloaded the protein sequences of *S. lycopersicum* (GCF_000188115.4), *S. tuberosum* (GCA_000226075.1), *Capsicum annuum* (GCF_000710875.1), *S. melongena*[11], *Nicotinana tabacum* (GCF_000715135.1), and *Arabidopsis thaliana* (GCA_000001735.1) from Ensembl (https://useast.ensembl.org/) and NCBI. Then protein sequences were aligned to the genome using TblastN (E-value ≤ 1e − 5), and the matching proteins were aligned to the homologous genome sequences for accurate spliced alignments using GeneWise (v2.4.1)[98]. For ab initio prediction, we used Augustus (v3.2.3)[99], Geneid (v1.4)[100], Genescan (v1.0)[101], GlimmerHMM (v3.04)[102] and SNAP (2013-11-29)[102,103] for our automated gene prediction pipeline. For RNA-Seq assisted prediction, total RNAs were extracted using TRIzol (Invitrogen, USA) from five different tissues, including roots, stems, leaves, flowers, and

fruits, which were used for transcriptome sequencing on the NovaSeq 6000 platform. In addition, the RNAs of these tissues were also pooled for full-length transcriptome sequencing on the PacBio Sequel II SMRT platform (except for S001 and S057). RNA-Seq reads were assembled using Trinity (v2.1.1)[104] for the genome annotation. The RNA-Seq reads were mapped to the corresponding genome using HISAT (v.2.0.4)[105] with default parameters. The full-length transcript sequences of each genome were assembled using StringTie2 (v.1.3.3)[106] with default parameters. Finally, a non-redundant gene set was obtained by merging gene models from the above three methods using EvidenceModeler (EVM, v1.1.1)[106,107] and Program to Assemble Spliced Alignment (PASA)[108]. The tRNAs were predicted using the program tRNAscan-SE (v1.4)[109]. The rRNAs were predicted by Blast, while other ncRNAs were identified using the Rfam (v14.1)[110] database with default parameters with the Infernal software (v1.1.4)[111]. Gene functions were assigned according to the best match by aligning the protein sequences to the Swiss-Prot database using Blastp (E-value ≤ 1e − 5). Genes were also functionally annotated using InterProScan (v.5.31)[112].

## SNP and indel calling

Raw Illumina short reads were filtered with Trimmomatic (v0.39)[113] (LEADING:3 TRAILING:3 SLIDINGWINDOW:4:15 MINLEN:100) to remove low-quality bases and adapter sequences. The clean reads were aligned to the 'S076' (named 'R06112') or 'S126' reference genome using the BWA (v2.2.1)[90] with default parameters. SNPs and Indels were called using the Genome Analysis Toolkit HaplotypeCaller (v4.2.6.1)[114]. The raw variants were filtered using following parameters: QD < 2.0, FS > 60.0, MQ < 40.0, MQRankSum < −12.5, ReadPosRankSum < −8.0, SOR > 3.0; minor allele frequency ≥ 0.05, missing rate ≤ 0.2.

## Phylogenetic and population genomic analyses

To investigate the population structure and phylogenetic relationships of eggplant germplasm accessions, the SNPs called based on 'S076' as the reference were further processed. The SNP dataset was pruned using Plink (v0.98.3)[115] (--indep-pairwise 50 10 0.2), resulting in 153,934 SNPs. A neighbor-joining (NJ) tree was constructed using MEGA7 software[116] with the wild species *S. violaceum* (S098) as the outgroup, and the results were visualized using the iTOL (v6) tool (https://itol.embl.de/). The genetic structure and ancestry information were inferred using ADMIXTURE (v1.30)[117] (-C 0.01 -j 24) with the number of $K$ ranging from 2 to 10. The results were subsequently visualized using TBtools (v1.123)[118]. For PCA analysis, the top 20 principal components and their corresponding eigenvalues were analyzed using Plink, and the clustering patterns were visualized using R. The optimal grouping of the 226 accessions was inferred based on the comprehensive consideration of their diverse phenotypic characteristics, phylogenetic relationships, geographical origins, PCA, and population structure. The nucleotide diversity (π) within subgroups and fixation index (*Fst*) representing genetic differentiation between each subgroup pair were calculated using VCFtools (v0.1.16)[119]. The linkage disequilibrium (LD) decay was evaluated using PopLDdecay (v3.41)[119,120] with a maximum measurement distance set at 1000 Kb.

## Structural variant identification, analysis, and graph-based pangenome construction

We used two strategies to identify structural variants for eggplant. A custom pipeline named SVGAP[34] (https://github.com/yiliao1022/SVGAP) was used to identify SVs (≥ 50 bp) from genome assemblies. This pipeline consists of six major steps: (1) Pairwise whole genome alignment using MUMmer4[88] with the default settings. In this step, the genome assembly 'S076' was used as the reference, and the other genome assemblies were used as the query genomes. (2) Identification of syntenic and orthologous alignments with the Chain/Net pipeline[121]. (3) Identification of SVs from pairwise genome alignments. (4) Generation of a non-redundant callset by merging SVs across samples. (5)

Genotyping each SV again across each individual genome. (6) Annotation of SVs: SVs were classified into different types based on their overlap with corresponding genomic features, including insertions of transposable elements (LTR retrotransposons and DNA transposons), tandem duplications (i.e., sequences that are inserted or deleted and are homologous to their flanking regions), gene fragments, 'TATA' satellites, and complex SVs consisting of more than two genomic features. Using this pipeline, we identified and genotyped a total of 34,938 deletions and 41,064 insertions, ranging in size from 50 bp to 100 kb, along with 315 inversions (>10 kb) and 164 translocations (>100 Kb). These SV datasets are available at Zendo (https://doi.org/10.5281/zenodo.18425195). Additionally, we also called InDels and SNPS using this pipeline to generate fully genotyped VCF files. The accuracy of the SVGAP pipeline for SV detection verified by PCR experiment is 87.50% (21/24) (Supplementary Fig. 26).

For short-read resequencing data, we used three programs-SvABA (v1.1.0)[122], DELLY (v1.1.6)[123], and Manta (v1.6.0)[124]-to call structural variants (SVs) for each sample. SvABA was run with the following parameters: 'svaba run -t $BAM -p 20 -L 6 -I -a germline_run -G $REF', while Manta and DELLY were executed with their default settings. SVs identified by these three programs for each sample were merged using SURVIVOR (v1.0.7)[125] with the parameters: 'SURVIVOR merge $sample.vcf.files 1000 2 1 1 1 50 $sample.merged.vcf', retaining only those SVs supported by at least two programs. To further consolidate SV calls across samples, we used two programs: Jasmine (https://github.com/mkirsche/Jasmine) and Panpop[126]. After comparing their results, we retained the Panpop output for downstream analyses, as it effectively merges and optimizes the majority of multiallelic SVs, while nearly encompassing all the calls made by Jasmine (Supplementary Fig. 27).

To calculate minor allele frequency (MAF), we used VCFtools with the --freq command. To estimate derived allele frequency (DAF), we used the genotype of *S. insanum* in the VCF file as the ancestral state. Specifically, when the genotype of *S. insanum* was "0/0", the alternative genotype "1/1" was treated as the derived allele, and vice versa. For the genome assembly–based SV callset generated using SVGAP, the genome of *S. insanum* was obtained from a recent study; ~74.32% (56,486 out of 76,002) of presence/absence variants (PAVs) were assigned an ancestral state. We also tested *Solanum violaceum* (S098) as an alternative outgroup reference, but only 28.09% (21,351/76,002) of sites could be assigned ancestral states. Nevertheless, the derived allele frequencies estimated using both references were highly consistent (Fig. 3d and Supplementary Fig. 11c). For the short-read–based SV callset, the genotype of accession S225 (*S. insanum*) was used as the ancestral reference.

We constructed graph-based pangenomes using deletions and insertions identified by the SVGAP pipeline and complementary short-read variant callers, using S076 as the reference and the vg toolkit (v1.55.0)[39]. Presence/absence variants (PAVs) were then genotyped across the eggplant population panel using short-read data. Briefly, reads were mapped to the pangenome graph with 'vg giraffe', read support was summarized with 'vg pack', and genotypes were generated with 'vg call'. In addition, we built a reference-free pangenome graph from 16 high-quality genome assemblies using PGGB (v0.7.4)[40] with default settings. Basic statistics of the graph were summarized using MultiQC v1.19[127]. We then deconstructed the PGGB graph using S076 as the reference path to extract SNPs, InDels, and SVs. For population-wide genotyping, variants of each type were anchored to the S076 genome coordinates, and corresponding pangenome graphs were subsequently constructed using the vg toolkit. These graphs were then used for read mapping and variant genotyping as described above.

## Gene-based pangenome construction

To illustrate the phylogenetic relationship of the four published accessions (PI180485, PI196043, PI200854, NO211)[15] relative to the 226

accessions, we first obtained SNPs by comparing their assemblies with that of 'S076' using the SVGAP pipeline. Subsequently, the shared SNP variants among the 230 accessions were used to construct a phylogenetic tree using the Neighbor-Joining approach and 4-fold degenerate sites. Besides, to investigate to what extent these assemblies and re-sequencing data represent the genetic diversity of eggplant, the sequencing data for over 3400 SPET-genotyped accessions were downloaded and mapped to the S076 reference genome for variant calling following the GATK pipeline. Common SNPs for a total of 3,673 accessions were used for phylogenetic tree construction with IQ-Tree (v1.6.12)[128]. The protein sequences of 17 eggplant accessions were assigned to gene families using OrthoFinder (v.2.5.4)[129]. The core (in all 17 genomes), softcore (in 16 genomes), dispensable (in 2-15 genomes), and private (only in one genome) genes were further defined based on their presence or absence in each of the 17 genomes. For the non-synonymous to synonymous substitution ($K_a/K_s$) calculation, the ortholog gene pairs for all possible pairwise comparisons among the 17 genomes were obtained using JCVI (v1.3.5)[128,130]. The paralog pairs were also obtained by comparing each genome with itself. Only gene pairs supported by both JCVI and OrthoFinder were retained for $K_a/K_s$ comparison. $K_a/K_s$ was calculated in two ways, either including only the ortholog pairs or including both the ortholog and paralog pairs. TBtools-II[118] was used to calculate $K_a/K_s$ ratios as well as for Gene ontology (GO) enrichment analysis. The above analysis was also applied for a subset of 11 de novo genomes using the same approach.

## Genome-wide association study

The filtered SNPs, Indels, and SVs based on 'S076' reference genome were used for genome-wide association study (GWAS) analyses, respectively, to identify the genetic loci associated with traits including fruit color and bacterial wilt incidence in eggplants. In addition, the SNP- and Indel-based GWAS analyses were also performed based on S126 as a reference genome. The variant datasets were filtered using Plink (v0.98.3)[115] with the criteria of missing rate ≤ 0.2, allele frequency ≥ 0.05, and no multi-allelic variants. Principal component analysis (PCA) was performed using Plink (-allow-extra-chr -pca 20) to extract the top three principal components. A standardized kinship matrix generated by GEMMA (version 0.98.1)[131] (parameter: -gk 2) was used as covariates. GWAS analysis was carried out using the mixed linear model (LMM) in GEMMA (-lmm 1). To determine the GWAS significance threshold, the GEC software (v0.2)[132] was used to calculate the threshold at the α = 0.05 level. However, for quantitative traits such as disease incidence, a relatively relaxed criterion ($P < 1 \times 10^{-5}$) was adopted as a secondary threshold. Finally, Manhattan plots were generated using the R software (v4.2.0)[133] with packages CMplot (v4.4.1) (https://github.com/YinLiLin/CMplot) and qqman (v0.1.9).

To evaluate the potential correlation between copy numbers of *SmEPS1* and *SmRoq1* with bacterial wilt resistance, the normalized mean coverage values of both genes were calculated as follows. The resequencing data were aligned to a custom reference sequence consisting of a single *SmEPS1* (S126_evm.model.Chr05.2786), a single *SmRoq1* (S126_evm.model.Chr05.2787), and a single-copy reference gene (S126_evm.model.Chr05.2898) identified by OrthoFinder gene family assignment among the 17 genomes. The mean coverages of two conserved regions within *SmEPS1* and *SmRoq1* were obtained using Bedtools (v2.31.1; coverage -mean)[134], and subsequently normalized to that of the single-copy gene. Samples were classified as E+ or E− if the normalized mean coverage of *SmEPS1* was greater or less than 2, respectively, and as R+ or R− if the normalized mean coverage of *SmRoq1* was greater or less than 15, respectively.

## Quantitative real-Time PCR

To analyze the expressions of candidate genes for fruit color and bacterial wilt resistance, fruit peels from samples with different colors and leaves from the seedlings in the VIGS experiment were collected with three biological replicates. Total RNA was extracted using the MagicPure® Total RNA Kit (TransGen Biotech, Beijing, China). First-strand cDNA was synthesized using the TransScript® First-Strand cDNA Synthesis SuperMix, and qRT-PCR was performed with the TransStart® Top Green qPCR SuperMix (TransGen Biotech, Beijing, China), following the manufacturer's instructions. Three technical replicates were performed for each cDNA sample. All primers used in this study were designed with Primer3Plus (https://www.primer3plus.com), and the primer sequences were provided in Supplementary Data 18. The qRT-PCR results were analyzed using the $2^{-\Delta\Delta Ct}$ method, and the *SmActin* was used as the reference gene.

## Virus-induced gene silencing

To assess the functions of *SmCYP82D47*, *SmEPS1s*, and *SmRoq1* in eggplant in response to *R. solanacearum* inoculation, conserved fragments of 274 bp, 228 bp, and 678 bp targeting *SmCYP82D47*, *SmEPS1s*, and *SmRoq1* were cloned into the pTRV2 vector, respectively (Supplementary Data 18). The VIGS assay was conducted as follows. The *Agrobacterium tumefaciens* strain GV3101 cells transformed with pTRV2:00, pTRV2: *CYP82D47, EPS1*, and pTRV2:*SmRoq1* were mixed with GV3101 cells transformed with pTRV1 at a ratio of 1:1, respectively. Each mixed bacterial solution was injected into three to four leaves of eggplant seedlings, which were incubated at 16 °C for 24 h under dark. Then, they were cultured under normal conditions (16 h light at 28 °C; 8 h dark at 22 °C) for one to two weeks. Subsequently, the silenced plants were infected with *R. solanacearum* strain RS742 using a root-dipping inoculation method[25]. The strain RS742 was grown in nutrient broth at 28 °C overnight and then suspended in sterile distilled water. The bacterial suspension was adjusted to a final concentration of $10^8$ CFU mL$^{-1}$. Roots of the seedlings were dipped in the prepared suspension for 20 min. After inoculation, plants were grown under a 16 h light/8 h dark photoperiod at 30 °C/32 °C (day/night) for disease development. There were at least 10 replicates for each treatment. To evaluate potential off-target effects, the VIGS target fragments were aligned against all annotated coding sequences (CDSs) of S092 (for *SmEPS1*) and S076 (for *SmRoq1*), using BLASTn with an E-value cutoff of 1e − 5. Similarly, the corresponding qRT-PCR primer pairs were mapped using Bowtie[135] with the parameters -S -f -v 2 -I 100 -X 5000 -y.

## Reporting summary

Further information on research design is available in the Nature Portfolio Reporting Summary linked to this article.

## Data availability

All the raw sequencing data, genome assemblies and annotations have been submitted to China National GeneBank (CNGB) database under Project accession number CNP0006177. The variant (VCF) and pangenome graph files have been deposited in the Zenodo database [https://doi.org/10.5281/zenodo.18425195]. Source data are provided with this paper.

## Code availability

The scripts associated with the pangenome analysis are available at Github (https://github.com/yiliao1022/eggplantpangenome) and Zendo (https://doi.org/10.5281/zenodo.18467477).

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

## Acknowledgements

We thank Y. Wang and C. Yu for help with the germplasm collection. This work was founded by grants from Guangdong S&T Program (Grant No. 2025B0202070003 to T.L.), the Guangdong Provincial Natural Science Foundation (Grant No. 2023A1515012563 and 2025A1515012414 to Q.Y.),

the Guangdong Provincial Rural Revitalization Strategy Special Fund Seed Industry Revitalization Project (Grant No. 2022-NJS-00-005 and 2023-NJS-00-003 to Q.Y.), the Special fund for scientific innovation strategy-construction of high level Academy of Agriculture Science (Grant No. R2021YJ-YB3019 and R2023PY-QY004 to Q.Y.), Modern Seed Industry Innovation Capability Enhancement Project of Guangdong Academy of Agricultural Sciences (Grant No. 2025ZYTS0505 to T.L.), the Department of agriculture and rural areas of Guangdong province of China (Grant No. 2025-NBH-00-001 to B.S.), the Basic Research Project of Guangdong Vegetable Research Institute (Grant No. 202110 to Q.Y.), and Research Start-up Funding from South China Agricultural University to Y.L.

## Author contributions

Q.Y., Y.L., T.L., and B.S. conceived and designed the study. Z.L., T.L., and B.S. prepared the materials. Z.P., Y.L. and M.H. performed the pangenome and structural variation analyses. Q.Y., Y.J., P.W. and Y.Z. performed the GWAS analyses. W.Z., S.Z., H.C. and H.Z. contributed to the field phenotyping. Q.Y., W.Z., and S.Z. performed the gene silencing experiments. Y.L., Q.Y., Z.P., Z.L., Y.J., P.W., T.L., Y.Z., B.S., C.C., and B.S.G. wrote and revised the manuscript. All authors have read and approved the final manuscript.

## Competing interests

The authors declare no competing interests.
