## [Peer Review file · Nature Communications]

The eggplant sub-pangenome reveals hidden structural variants associated with fruit color and bacterial wilt resistance

Corresponding Author: Dr Qian You

Version 0:

Reviewer comments:

Reviewer #1

(Remarks to the Author)

You and colleagues reported the populational and pan-genome analysis of eggplant, an important horticulture crop with both scientific and economic importance. They newly assembled 9 genomes, constructed a graph-genome, investigated the effect of SVs on gene expression, and discovered candidate loci involved in eggplant fruit color and disease resistance. Overall, this work offers important genetic resources to understand the historical evolution and to improve eggplant breeding. However, I have some major concerns, e.g., the quality of SV datasets and the association results of fruit colors.

Major:

1. The authors used a customized SV calling pipeline to identify SVs between newly assembled genomes and the reference genome. In this pipeline, three commonly used tools including SvABA, Delly and Manta were used, and those SVs supported by two callers were retained. Although this pipeline seemed reasonable, please offer more details regarding the SV statistics among three callers. In fact, I am more concerned about the merging of SVs from different accessions using SURVIVOR. The strategy of this tool was to collapse SVs with mutual overlaps into a non-redundant SV set, which was not sensitive enough when dealing with complex SV breakpoints. Meanwhile, the integrated populational SVs were not genotyped accurately for each accession. The authors can try additional SV mergers and cross-compare with SURVIVOR results to increase accuracy of population SV, for example, PanPop (Nat Commun, 2024), a recently developed population SV merger which had been optimized for plant species with relatively high repeat contents.

2. The authors applied association analysis to investigate the genetic basis of eggplant fruit color. They concluded that the large inversion, 6-bp deletion in coding regions, as well as the gene expression variation of MYB1 gene were all strongly associated with fruit color. Chromosome inversions may have impact on gene expression, especially for those genes that near breakpoints. However, inversions have less impact on gene coding regions only if the coding regions overlapped with breakpoints, while it seemed that the coding variation had strongest evidence associated with fruit color (Fig. 4g)? I suggest the authors to re-organize these results to make more clarification regarding the inference of potential causal variant (coding or regulatory) of this trait.

3. The association analysis of SVs and gene expression was performed by RNA-seq data of only 20 accessions (Line 227-247). However, the sample size was a little bit small to have high-confidence conclusions regarding SV effects on gene expression. To meet the standard of a high-quality publication in Nat Commun, I suggest the authors to further incorporate more populational RNA-seq data (>100 accessions) which can better resolve the causality between SV and gene expression, as well as to help the annotation of GWAS results through eQTL mapping.

Minor:

1. The pan-genes in this work was constructed by comparing 9 newly assembled and 5 previous published genomes. The authors should be aware of the potential biases of pan-gene construction originated from different gene annotation methods.

2. Line 123-124, the genetic diversity (π) was unconventionally high in eggplant (0.181, 0.187). It was more common to see an estimation of genetic diversity between 1.0×10^{-3} to 6.0×10^{-3} among major crops including rice, maize and millet (e.g., Nat Genet 54, 1736–1745 (2022)).

3. Line 187, it should be 'By comparing the other 13 genome assemblies against the reference genome of S076'.

4. The impact of large inversion on recombination and linkage disequilibrium should be further discussed.

5. Line 344, the authors found a 15-fold increase of genetic diversity near breakpoints of inversions. If this inversion

encountered consistent artificial selection during breeding, the genetic diversity of this inversion should exhibit significant reduction due to bottleneck effects. I mean, the authors should be careful to these types of observations which might be originated from the mis-mapping of paralogous reads resulted from SVs.

Reviewer #2

(Remarks to the Author)

NCOMMS

I found the manuscript beautifully detailed and compelling that there is tremendous scientific value in pangenome studies. The use of fruit color and disease resistance are excellent examples. My recommendation for major revision is because of the misportrayal of the work as encompassing global or at least dominantly Asian biodiversity of eggplant but really collections are mostly from China (a major place of great interest!) and the ways eggplant accessions are lumped into larger groupings like countries for some and continents for others makes the study feel inherently biased and hard to follow or trust. The authors should consider how they are categorizing the germplasm and arriving at their conclusions. To me, the value is in the pangenome and GWAS work, not in the domestication confirmation part, which is where the bias and lack of clarity on what the results mean really shows through.

My specific comments are below. Thank you for the opportunity to review this manuscript.

line 44: change 'or' after SE Asia to 'and'

line 47: about *S. macrocarpon* and *S. aethiopicum*, they are now cultivated around the world as well. I've seen plenty of *S. macrocarpon* growing in China (e.g. Guizhou) and both species are getting more popular in the Americas as they are consumed by people with African heritage such as Brazil's massive African diaspora. Please revise to explain they are not just locally cultivated. You may also wish to mention *S. macrocarpon* is consumed for its leaves as well as its fruits, and *S. aethiopicum* is a popular ornamental crop looking like pumpkins in floral bouquets.

line 63: Break up the paragraph into two starting with 'the concept'

line 64: 'the full' is impossible. 'much more' should replace 'the full'. Let's not date ourselves....

line 65: obviating  reducing

line 75: I don't think the goal of 'to enhance the power of genetic discovery' is crisp enough. Simply saying 'to enable domestication and artificial selection studies to be enhanced with less bias' or something similar is more clear and direct.

line 104,105: This sentence is in direct conflict with the sentence on lines 88-89. You mainly have China and SE Asia representation of cultivar diversity, not worldwide phenotypic and genetic diversity.

line 125: decreased at 69kb in which group?

line 128: South Asia is included in C1 too. If South America is pointed out, SE Asia should be pointed out. I also don't understand how the C2 and C3 could be parsed out of PCA. There is no explanation of a geographic pattern or region that is consistent with the map. The clusters are also not separated. It is not ideal to group them. Seek a less subjective grouping system.

Fig 1: I don't understand how admixture could split the populations. I would support these three or four populations based on the phylogenetic tree but there are plenty of accessions in the admixture plot that could be rearranged into different groups based on pop assignment color, like some in the purple-blue group that have more than 50% red. Maybe explaining the decision making process would be helpful here. I am also curious if you go with $k=4$ if the light blue are a clade of their own in the phylogeny.. If so, it would be nice to highlight on f.

line 159: a matter of style, but it would help readers to say A total of 16,512 (44.81%) core, 3570 (9.69%) softcore, 16,...dispensable, and 391 (1.06%) private gene families were obtained ().

Fig 2a: these colors are similar to the C1-C3 purple and green. I suggest you change the colors so they don't look similar. The colors may also not be good for colorblind people. What about a dark red-orange for new genomes and black for published genomes?

Fig 2: I don't understand the calculations of GO counts and p values with fold change. What is being compared to what for examining representation as fold enrichment? The legend could be clearer if a little longer for this section .

Fig 3: Here too I cannot easily see the differences in shade. Please choose colors and shades that are more distinguishable from each other.

line 315: 'Known color-related genes are depicted.' These could be more specifically described as the genes in the Hydroxycinnamic acid synthesis pathway that take from anthocyanin biosynthesis precursors. C4H is the known key enzyme shunting molecules to HCAAs or pigments. I would just advise the authors to make sure they highlight the literature on these pathways and not to oversimplify as color-related genes.

Fig 5a: I cannot easily see the differences between uncertain and other countries but I think this section needs some reconsideration. You have purple arising in another country very early in the tree, both with and without the inversion. The authors don't acknowledge where that came from. It was favored in China but also most accessions chosen were purple eggplants in china -- look how few non-purple eggplants there are from China. I am not convinced the inversion arose in China. It looks like it arose in multiple places. It does seem to be favored in China, so I am in agreement with the authors on that. What would make this more interesting is to get more fine-scale, where the purple eggplants without the inversion from China (the bottom cluster) could be associated with a geolocation (province level? Latitude? Ecoregion?) and so could the varieties on the top from China that are green. Then regarding the purple inversion accession from other countries, how thoroughly were those countries sampled here? Without naming the countries we cannot know. I bet one is from India and there is only one representative. This study unfortunately suffers from being biased geographically toward China while doing a broader landscape analysis and thus requires more precaution with caveats to not jump to conclusions about what happened in China versus the rest of Asia. Then it suffers from underutilizing the high resolution of Chinese germplasm in the study, not doing a proper landscape-level analysis.

line 477: This stat of 65% of eggplant being grown by South America is incorrect. It seemed high so I calculated on my own, and total yield for 2021 (the most recent FAO data), defined by yield in 100g/ha, was 2322313. The proportion of that for South America is 11.2%. If you use another metric, like tons, South America is not even a percent of eggplant in tons. Asia is

94% of eggplant tons production.

line 486: 'confined to one center'  make sure it's clear if China is expected to be derived from the SE Asia center, a South Asian center, or a China center. This will be confusing to readers and it sounds like the authors are unsure themselves about how Chinese germplasm relates to the domestication narratives.

line 519: 'we ensured a comprehensive'  you cannot ensure a comprehensive catalog of SVs from a species when you are not sampling everything and you are limited by technology. Avoid hyperbole and say 'we attempted to build a comprehensive catalog' or 'we aimed for a comprehensive catalog'.

line 532: it would be beneficial to relate this to the experiences of breeders. has breeding for purple color ever been a challenge because this inversion was not known? Probably impossible to find such information but I am curious what reports are out there on breeding for various gradations of purple in eggplant.

line 535: not global, YOUR samples.. Make sure to describe accurately. The big flaw in this paper is the misportrayal of the diversity examined herein. Be careful.

line 544: studied in detail in China. Not especially in China. You didn't look enough elsewhere to say 'especially'

line 658: citation marked twice, no closed parentheses.

Data Availability: I see August 2026 as the date the data are made public. Will that change once the manuscript is in press to align with publication?

-Rachel Meyer

Reviewer #3

(Remarks to the Author)

The present study re-sequenced 198 eggplant accessions and de novo assembled nine genomes from them to represent eggplants' genetic diversity. They also identified a large inversion associated with purple fruit color and two genes associated with bacterial wilt resistance. This study provides a valuable resource for future studies, however, I have some concerns.

Major ones:

(1) As claimed in the title, pangenome analysis was applied in this study, but the benefits of constructing a pangenome are unclear.

(2) To what extent the assembled genomes can represent the genetic diversity of eggplants? Inclusion of the 25 accessions re-sequenced by Barchi et al. (2021) and the over 3400 SPET-genotyped accessions sequenced later (Barchi et al., 2023) may help clarify the representativeness of this genome collection. Especially, it can help estimate the power of the pangenome in SV genotyping with short reads.

(3) The association between the large inversion and the fruit color may not 'underlie' the formation of purple eggplant fruit. I have several questions related to this part of the analysis.

(3.1) Line 276-311: Is the strongest signal for related to SmPAL and SmMYB1 genes on the inversion? Given that nearly half of accessions without this inversion are purple, the association between this inversion and the purple color is likely hitchhiked by the suppression of recombination, because the 6-bp deletion on SmMYB1 can better explain the expression change. There might be more surrounding SNPs and SVs with higher association. How far are the two genes, SmHIPP and SmMYB1, distant from the inversion? To further dissect the function of the large inversion and 6-bp deletion, the two variations can be combined to contrast the haplotype versus phenotype in figure 4h.

(3.2) Line 335-337: It is more likely that this inversion was inherited from the accessions at the base of the tree, including accessions from both China and other regions, if that tree is rooted. No group information is available in this tree.

(3.3) Line 339-352: No evidence was shown that the allele of SmHIPP linked with this inversion is beneficial to the stress response. The nearly fixed genotype within the inversion-harboring monophyletic group could be explained by narrow genetic diversity caused by founder effect rather than selection against an inversion, especially when this inversion may not be causal.

(3.4) Line 354-362 & figure 3e & 3f: It is surprising that the LD decays much faster in the INV group than in the non-INV group, given that the nucleotide diversity is much higher in the non-INV group. I would suspect it of being an artificial error caused by using the wrong reference for physical distance calculation. The high LD could disappear if a 'non-INV' genome was used as the reference genome.

Besides, since 'S076' was used as reference to talk about variants, the 'INV' group should be those harboring the 12.4-Mb inversion which are different from 'S076'.

I also have minor concerns related to the illustration and interpretation of results and details of methods:

(4) Need more information to illustrate the association between geographic origin and C1, C2, and C3 grouping, for example summarizing location versus population in table S1.

(5) Line 130-131: Need more evidence to support the conclusion. Higher genetic diversity doesn't necessarily mean that genetic variants are more important.

(6) Line 160-162 & 729-731 & figure 2f & 2g: How were the expression profiles obtained? Are they representing the 14 genomes used in this study? Need further information on how Ka/Ks values were calculated. What was the purpose of using an outgroup in TBtools? What type of gene pairs was compared? How to pair genes if there are paralogs?

(7) Figure 3a & 3b & line 252: The figure legend for 3b is unclear. Additionally, if the value 19,824 represents the number of SVs present in PI180, it doesn't align with the length of the bar in 3a. The bars in 3a don't match numbers in supplementary table S4 either. The 'growth rate' is sensitive to the order to include samples, so the decline is artificial.

(8) Line 197-199: 'S076' is the reference genome use for comparison, then why is the 12.4-Mb inversion described to happen in 'S076' instead of other accessions in supplementary figure 7? is there evidence supporting the ancestry state?

(9) Line 199-201 & figure 3c: More details are required for this analysis. Different types of TEs can be harbored in the same

insertion or deletion. 'Tandem duplication' is not a type of TE, so it is weird to include it here. Please also define 'the most common SVs'.

(10) Line 201-207: More details are required in the SV calling and genotyping methods. No information is available on the pipeline (SVGAP, <https://github.com/yiliao1022>) used to call SVs based on genome comparison. The parameters used for each software should be provided. Additionally, how many SVs were included in the final graph? Were there any size filtering on these SVs and in the downstream analyses?

(11) Line 208-226: The number of sampled genomes is too small to represent the real allele distribution in the population. A reasonable way is including the 201 re-sequenced accessions.

(12) Line 229-230: Like figure 3c, definition of the SV type classification needs to be clarified. And there is no evidence supporting that LTR-TE insertions have higher 'deleterious impacts' on genes.

(13) Line 231-239: Supplementary figure 6 and 7 mentioned are missing. The large sample sizes for comparisons in figure 3g increase the power, which may increase type I error. Please add a bootstrap-based method to provide a confidence interval.

(14) Line 266-270: Signal for SmTT8 is absent in figure 4a.

(15) Figure 6c: GWAS for different phenotypic data should be plotted separately to improve clarity. These plots can be aligned to compare the consistency of signals.

(16) Line 414-434: Are these two selected SVs signaled in SV-based GWAS as described in line 401? What are the relative locations of selected SVs related to the target gene SmEPS1?

Reviewer #4

(Remarks to the Author)

In this study, the role of structural variation in skin color and bacterial wilt resistance of eggplant was discussed by pan-genomic analysis. In addition, the genetic information of 201 eggplant germplasm was analyzed to provide guidance for genetic domestication of eggplant. The results of this study are detailed and have certain value for basic research and molecular breeding of eggplant in the future. In response to this study, the reviewers have the following questions:

1. What was the basis for the selection of S076 as a reference genome in the population structure analysis? We know that the authors assembled nine genomes from scratch, and I think the authors need to explain the basis for selecting reference genomes for population structure analysis.
2. Please refine the genome assembly strategy and the different technical approaches to obtain a specific description of the amount of data available.
3. It is suggested that each attached figure in Figure 2 should choose a color match with more vivid color contrast, which will help readers to distinguish better.
4. For the obtained structural variation, the author needs to consider the molecular verification of some representative important structural variation to prove the reliability of the data.
5. Based on numerous previous reports, fruit shape of eggplant is also an important selected trait. Has the author tried to analyze structural variation and fruit shape selection of eggplant?
6. Regarding the classification of fruit color, we believe that the distinction between purple and non-purple is too simplified. Currently, there are abundant reports on the color of eggplant fruits, and the results obtained by the author are not enough in terms of novelty compared with previous reports. This paper systematically expounds the formation and domestication of eggplant peel color.
7. The description of bacterial wilt inoculation methods and evaluation criteria as well as biological and technical duplication is not clear enough, please add a detailed description.
8. This study provides a comprehensive analysis of structural variants (SVs) in eggplant; however, the SV data are not publicly available. The authors should make these data accessible to promote transparency and facilitate further research. Additionally, the description of the methods used for SV detection is insufficient. The authors should provide more detailed information about the methodologies employed for identifying the SVs.
9. One striking finding in this study is the observation that a large inversion is strongly associated with peel color in eggplant, a phenomenon also observed in other organisms. Although the authors present compelling evidence to identify the causal variants and provide some population genetic analyses to demonstrate the potential artificial selection of this inversion, the current explanation could be further strengthened. I encourage the authors to incorporate additional population analyses or discussion to better elucidate the domestication history and evolutionary significance of this inversion.
10. To clarify how SmEPS1 associated with bacterial wilt resistance, I encourage the authors provide more analyses, such as comparing the expression and the protein sequences among the different copies of SmEPS1.

Version 1:

Reviewer comments:

Reviewer #1

(Remarks to the Author)

The authors have done a nice work in addressing my concerns. The manuscript is now much improved. Thank you! For additional suggestion, I will not put the result of two pan-gene versions into the main text, which will reduce the readability of this section. I agreed with the authors to retain the pan-genes of "11denovo + 6published" in Results, while to re-organize the "version2 pan-genes", emphasizing the potential biases of gene annotations into Discussion.

Reviewer #3

(Remarks to the Author)

Thank you to the authors for the revised manuscript. I appreciate your efforts in addressing my previous comments and enhancing the overall quality of the paper. The manuscript provides valuable genomic resources and presents interesting trait associations. Most of my concerns have been addressed. However, after reviewing this revised version, I still have a few remaining concerns.

The authors have made notable improvements in clarifying the representativeness of the assembled genomes, tracing the origin of the large inversion, refining the methodology, and removing inconclusive inferences. Despite these improvements, several concerns remain unaddressed or insufficiently justified.

I use numbered tags provided by the authors to trace the comments.

[R3C1] This response does not adequately address my concern.

(1) I infer that the authors' statement regarding "benefiting from the gene number comparisons" refers to the results described in lines 447–461. However, this section does not provide substantive evidence from gene-based pangenome gene family data to support the association between gene copy number variation (CNV) and phenotype. Moreover, the results presented raise concerns about the quality of gene annotation. Specifically, the structure of the gene *evm.model.Chr05.2618* may have resulted from an erroneous fusion of two neighboring genes, *evm.model.Chr05.2786* and *evm.model.Chr05.2787*. This potential annotation error is partially supported by subsequent manual curation (lines 462–465). Figure 6e illustrates CNV, but the distinction between "R+/R-" needs clarification. Additionally, based on the gene distribution patterns, genotypes carrying "E+" seem to originate from the same ancestral sequence, and similarly, those with "R+" appear to share a common ancestral sequence. This suggests that resistance in accessions carrying these genotypes may be driven by functional gene variations rather than solely by a potential dosage effect resulting from gene expansion. Furthermore, if the pangenome analysis only contributes to the identification of the *EPS1* gene, then the mention of structural variants (SVs) related to fruit color in the title could be misleading.

(2) The genetic diversity of SVs within the studied populations is one of the key contributions of this study. However, the analysis of this diversity does not substantiate the claim made in the title that the study "uncovers large hidden structural variants associated with fruit color and bacterial wilt resistance."

(3) There is a substantial discrepancy between the revised Suppl. Fig. 14d and the original Suppl. Fig. 9B. Notably, the authors used two different SV datasets, "SV_panpop" and "SV_SVGAP," in Suppl. Fig. 14d and 14e, respectively. This inconsistency is confusing and does not align with the uniform description of SV detection and genotyping in the main text. Similar inconsistencies are also present in Suppl. Fig. 18. These contradictions undermine the reliability of the analysis.

(4) The graph-based pangenome and SV population diversity are highly valuable resources. However, they do not sufficiently address the core concern—how these resources facilitate the identification of phenotype-associated SVs in this study.

[R3C3] Fig. 5e provides strong evidence. Please add 'Origin' information as in 5a to the figure to support the inferred origin of the Chinese-type inversion.

[R3C7] The authors should first carefully review their analyses, finalize the dataset they intend to use, and ensure consistency before re-computing SV distributions and conducting subsequent analyses, including figure generation. This will help maintain the credibility of their results by ensuring that the data used is consistent throughout. The revised Figure 3b still does not align with the reported results. According to the results section and Table S4, there are a total of 76K SVs, yet Figure 3b displays fewer than 71K SVs. Additionally, the authors have released two SV datasets containing 75,683 and 140,500 SVs, neither of which matches the reported results. Similar inconsistencies are present in other parts of the study. Unless there is a specific reason, which should be clearly explained, SV-GWAS should be conducted using the final SV genotyping dataset rather than one of these intermediate versions.

[R3C10] One of the key contributions of this study is the construction of a graph and the precise SV genotyping using *vg* toolkit, enabling more accurate and comprehensive graph-based SV analysis. However, according to this response, the actual SV-aware population genomic analyses were conducted using SV genotyping data derived from SVGAP, which is based on a single reference genome. This approach does not fully demonstrate the advantages of assembling multiple genomes and constructing a graph, which is a central strength of this study.

[R3C16] Please incorporate the genotype data (E/R) from Figure 6f into the dataset presented in Figure 6b. Additionally, the authors should release the SV genotyping results that were consistently used throughout the study, ensuring that they include all relevant accessions.

Overall, the manuscript has improved, but the emphasis on gene-based and graph-based pangenome analyses has not yet been sufficiently supported.

Reviewer #4

(Remarks to the Author)

The revised manuscript has been improved a lot and my concerns have been addressed. Only minor suggestions are given as below:

1. Authors used two accession names, 'GQIE' and 'HQIE', in Figure 2b, e, and in Figure 4c, d, but they were not defined in the text. Although 'GQIE' most likely refers to 'GUIQIE-1' and 'HQIE' should refer to 'HQ-1315', consistent names shall be used in order to avoid misunderstanding.

2. I suggest authors immediately release the genomes and sequencing data once the paper is in press.

Reviewer #5

(Remarks to the Author)

Review:

You et al performed a pangenome analysis of eggplant, based on 6 published and 11 novel chromosome-level genomes, and short read resequencing of 201 accessions. They then proceeded to compute PAVs, SNPs and SVs using a reference-based approach, construct a graph pangenome, and finally focus on two phenotypes: anthocyanin biosynthesis, where they find a large inversion associated (most likely through a founder effect), but not causally related to a difference in fruit color, and bacterial wilt resistance, where they conduct an elegant and thorough GWAS analysis of the genes involved, demonstrating the power of using multiple chromosome-scale assemblies. While most of the conclusions are expected, the bacterial wilt work is highly novel. However, the experimental approach presents some serious limitations:

1. The main one, as pointed out also by reviewer 2 (R2C0), is the sampling: the accessions used originate almost exclusively from one of the two recognized domestication centers (SE Asia) and from China, with just one accession from the second domestication center (Indian subcontinent) 2 from Central/Northern Europe, and 2 from the Americas (Fig 1a). Several important domestication/diversity centers (India, Middle East, Southern and Eastern Europe, Africa, Japan and Korea) are not represented, or represented just by one accession in India. This limitation is further exacerbated by the bias in the resequenced accessions, that cluster in a narrow region of the diversity tree (Supplementary figure 5). Thus, I don't think this concern by reviewer 2 has been appropriately addressed. A pangenome should give a more or less complete representation of genes within a species or clade, so this double geographic and genetic bias in the population is a serious limitation. The definition "regional pangenome" or "sub-pangenome" seems more appropriate for the present dataset. This bias in the accessions has a bearing also on other comments by reviewer 2, such as R2C16 (did the inversion arise in China? Its abundance is due to a founder effect?) or R2C19 (the catalog of eggplant SV is not comprehensive). Without a geographically and genetically balanced sample, these comments cannot be addressed.

2. A second limitation is the outgroup used for the phylogenetic analyses, composed by 4 *S. aethiopicum* and 1 *S. violaceum* resequenced accessions, of which the latter was also assembled to chromosome scale. Both species belong to the *Anguivi* grade, and are thus phylogenetically distant from *S. melongena*, whose direct progenitor and sister wild species are, respectively, *S. insanum* and *S. incanum*. This limitation questions the validity of many of the phylogenetic analyses made, such as inferring the ancestral state of SV (Fig. 3d) or the ancestral component analysis (Fig 1e). The analyses should be repeated using the direct eggplant progenitor *S. insanum* and/or the sister species *S. incanum*.

3. A third limitation is the approach used for pangenome graph construction. The main advantage of graph-based pangenomes is an unbiased all-vs-all comparison, thus avoiding the so-called reference bias (Secomandi, et al (2025). *Nature Genetics*, 1-14). Two main pipelines exist for this purpose: Minigraph-Cactus (Hickey, et al. (2024). *Nature biotechnology*, 42(4), 663-673) and PanGenome Graph Builder (Garrison, et al. (2024). *Nature Methods*, 1-5.). I therefore wonder why authors decided to first detect SVs with a reference-based approach and then build the graph. The resulting reference bias, summed to the geographical and genetic bias mentioned above, is likely to limit the value of this pangenome. Reference bias can be removed using an all-vs-all comparison, eg based on PGGB.

Other suggestions for improvement:

4. The fact that gene families in the pangenome "nearly reached a plateau after adding 17 genomes" (line 171) is largely expected. What about using all genes identified in the 17 genomes?

5. In the admixture analysis, authors tested 2-5 Ks. In the CV plot however, k from 2 to 8 are analyzed. Why did they select 3 as best K, since the CV plot shows a plateau at 8 (Fig. 1A)? This point was also raised by reviewer 2 on different grounds (R2C10) and needs to be better addressed.

6. Previous work by other groups has identified several candidate genes for anthocyanin biosynthesis in eggplant, as well as, in some cases, the causal mutations (see eg Florio, et al (2021). *International Journal of Molecular Sciences*, 22(17), 9174; Mangino, et al (2022). *Frontiers in Plant Science*, 13, 847789; You, et al (2023). *Horticulture Research*, 10(2), uhac268; Xiao, et al (2024). *International Journal of Molecular Sciences*, 25(10), 5241). Of these, the only paper that is cited (lines 305 and 616) is the You et al paper, by some of the authors of the present paper. In particular, the 6-bp deletion in the Chr10 Myb gene that is causing the non-purple phenotype (Fig 4f-g in the present paper) has been described by Mangino et al (2022) (fig 5j in that paper). Appropriate credit should be given to previous work in the field by other groups.

7. In the text, the authors conclude that the 12.4 Mb inversion on Chr 10 is not associated with the non-purple color, but it reduces genetic diversity in the surrounding chromosomal region. This conclusion is correct, but not thoroughly analyzed and inappropriately presented in the abstract:

a) both π and Tajima's D are reduced in the whole pericentromeric heterochromatin (Fig. 5b, d; Chr10 should be metacentric, Fang, H., et al (2025). *International Journal of Biological Macromolecules*, 284, 138094). How does an inversion in a chromosome arm affect these two parameters in pericentromeric heterochromatin, even on the other side of the centromere? Is it due to "gamete culling" (i.e. gametes that have a recombination event in the inverted region contain unviable chromosomes, thus reducing recombination over the whole chromosome)? If so, what is the predicted fitness on the chromosome carrying the inversion in a free-pollinating population? This point should be discussed.

b) The present formulation of the abstract "we identified a 12.4 Mb 27 inversion at a high frequency (50.7%) within the

population, which is associated with fruit color, likely due to genetic hitchhiking” is misleading, in that it suggests some sort of causal relationship between the inversion and fruit color. I suggest changing into “...associated with a previously identified mutation for fruit color..”

Minor points:

8. I don't understand fig 2g: It supposedly depicts Ka/Ks values for orthologous pairs in the core, softcore, dispensable, and private genes. But if a gene is private, how can it be part of an orthologous pair? Could the authors explain better how they did the analyses in this figure and suppl figs 7 and 8e?

9. Ref 14, which should refer to a chromosome level eggplant assembly, instead refers to phylogenomic discovery of deleterious mutations in potato. Ref 69, which should refer to fruit pigmentation, instead is a review on bacterial wilt resistance in Solanaceae. Please correct.

Version 2:

Reviewer comments:

Reviewer #3

(Remarks to the Author)

My concerns have been largely addressed in this revision. I have only one remaining comment and one minor suggestion.

The authors have effectively highlighted the advantage of graph-based pangenomes in accurately identifying structural variants (SVs). However, the proposed association between gene dosage effects and bacterial wilt (BW) resistance remains insufficiently supported. This association is further complicated by the finding that silencing a single gene, SmEPS1, results in a substantial reduction in BW resistance (Fig. 6g,h), even though the accession background of the wild-type control is not specified. Moreover, the connection between the representative SVs ('SVe' and 'SVr') and the observed copy number variation (CNV) is not yet strongly substantiated, particularly given the limited sampling of only 11 assembled genomes. The contribution of CNVs may be better interpreted as providing a genetic basis for potential mutational events rather than directly contributing to phenotypic resistance.

Minor suggestion: The criteria used to define and determine derived alleles should be clearly described in the Methods section.

Reviewer #5

(Remarks to the Author)

The authors have only partially addressed my main concern, which was the scarce representation of accessions from the Indian domestication center. They added few resequenced accessions from Europe and Asia, but the Indian domestication center remains crucially underrepresented, together with other important regions, such as Middle East, Africa, Korea and Japan (Fig 1a). Therefore, the definition “sub-pangenome” is the most appropriate. Its limitations should be more extensively discussed, indicating explicitly that it lacks a sufficient number of accessions from the Indian domestication center and expansions from that center, and therefore many genomic events that occurred in these regions are likely missing from this resource, limiting its usefulness.

As noted previously, this limitation is likely to be exacerbated by the reference bias introduced by graph construction using SVGAP. The authors, in their reply, show that the reference-independent PGGB software is performing as well as SVGAP in detecting bias. A logical next step is to perform GWAS on the PGGB reference-independent graph. This analysis is missing in the present revision and should be introduced.

My third comment, i.e. the lack of a proper outgroup to anchor the phylogenetic data, has been addressed by using a single, publicly available *S. insanum* chromosome level assembly that, as the authors note, “substantially strengthened” the resolution and robustness of their evolutionary inferences. It must be noted that *S. insanum* is heterogeneous, with some accessions being feral or admixed forms with *S. melongena*, and therefore the resolution and robustness of the evolutionary inferences would be further strengthened by including multiple assemblies of *S. insanum* and of the closely related *S. insanum*. Given the fact that generating additional chromosome-level assemblies is time-consuming and expensive, this is not an absolute requirement.

My comments 4-6 have been adequately addressed.

Additional comments:

Ralstonia resistance:

The relative contribution of CYP82, EPS and Roq1 genes in Ralstonia resistance remains elusive and should be better studied. In order to better understand it, a supplementary figure should be provided in which pie charts are shown for two of the gene classes in genotypes separately for the two classes of the third gene, together with the % average susceptibility of the two classes. Eg: pie charts for the four classes (E+R+,E-R+,E+R-, and E-R-) in TGG and TGA genotypes. And so on, for the E+/E- and R+/R- genotypes. % susceptibility and number of accessions should be indicated for the genotypes in each class.

Supplementary figure 19 is not very informative in clarifying the role of the different genes in ralstonia resistance. First, it is

unclear how genes in the different clusters correlate with those in fig. 6e. These should be marked with names or, better, with colors corresponding to the different clusters in supplementary fig 19. Importantly, supplementary figures (or if it's too complicated, tables) should be provided, giving the expression of individual genes in populations with different classes of susceptibility to ralstonia (eg, low, intermediate and high) and for all the CYP82, EPS and Roq1 genes analyzed. Alternatively, the CYP82, EPS and Roq1 gene composition and expression should be added to Table S10.

VIGS: Although the authors claim that they performed RT-PCR to evaluate the silencing of the different EPS1 and CYP82D47 homologs, I wasn't able to find anywhere the data. These data should be shown, together with a table with the % homology of the different homologs to the silencing fragments used. It would be also appropriate to show VIGS data also for the Roq1 homologs.

Minor: The pie chart data on CYP, EPS and Roq1-like genotypes in various classes of ralstonia susceptibility are shown only for four susceptibility classes (0 to 40-60). They should be shown also for the remaining two classes (60-80 and 80-100).

Once these data are provided, a discussion of the synergic-epistatic relations of the three classes of genes in determining Ralstonia resistance should be provided.

Other:

Supplementary Table 2 provides only the statistics for the 11 de novo assembled genomes. Statistics (except HiFi and HiC, if not available) should be provided for all 17 chromosome level genomes used in the analyses, including the newly added *S. insanum*.

Data availability: A search of the Chinese National Genebank with the Project or individual genome accession numbers found no results. Raw sequencing data, genome assemblies, annotations, must be released on a public database and the graph on a public pangenome browser such as PpanG (<https://doi.org/10.1186/s12864-024-10302-5>) upon paper acceptance.

Version 3:

Reviewer comments:

Reviewer #3

(Remarks to the Author)

All my previous concerns have been fully addressed.

Reviewer #5

(Remarks to the Author)

Overall Assessment

The authors have made a substantial effort to address my comments, and the manuscript is improved as a result. The clarification of the 12.4 Mb inversion on chromosome 10 as a case of genetic hitchhiking rather than a causal variant is a valuable scientific conclusion. The new analysis of the synergic-epistatic interactions of the three bacterial wilt (BW) resistance loci (SmCYP82D47, SmEPS1, SmRoq1) is a significant addition that clarifies a complex genetic architecture. However, the revised manuscript is unfortunately still beset by several critical issues:

- The authors' response to the request for a reference-independent pangenome-GWAS (using PGGB) is contradictory.
- In the light of the new data added, the VIGS results are difficult to interpret
- The manuscript's primary narrative for fruit color focuses on the 12.4 Mb inversion on Chr10 while dismissing other loci as "less significant". This claim is wrong; the authors' own supplementary table shows SVs on Chr05 and Chr06 that are orders of magnitude more statistically significant.
- The methodological novelty of this paper has been superseded by the recent publications of Gaccione et al. (2025) and Yu et al (2025).

Below the point-by-point evaluation of the authors' responses:

Comment R5C1. The authors adopted the term "sub-pangenome" in the title and abstract. They have added a paragraph to the Discussion acknowledging the geographical bias and calling for future studies to include samples from India, the Middle East, Africa, etc. This is a transparent and appropriate response.

Comment R5C2. This is still a significant shortcoming. The authors claim to have performed the requested PGGB-GWAS, then they dismiss the method, stating it "reported a substantially lower number of SVs" and that for BW resistance, "no signals were detected at the two stable peaks on Chr04 and Chr05." They use this "failure" to justify retaining their original, biased SVGAP results. This justification does not stand upon closer scrutiny: For BWR, Supplementary Table 13 explicitly lists 7 "PGGB derived SV", albeit on different chromosomes and with different p-values than with SVGAP (Supplementary Table 11). Fig. 6C and Suppl Fig. 22 are not comparable, since one refers to SNP-GWAS and the other to SV-GWAS and, for fruit color, it is not clear if the Manhattan plot in Fig. 4c refers to SNPs, SVs or both, so the reader is left wondering about the comparison between SVGAP-GWAS and PGGB-GWAS. Since PGGB is a robust, reference free method, in order to substantiate their claims the authors must report the complete sets of associations (SNPs, short InDels and SVs) obtained for the two traits with the two methods. Their current response is evasive.

Comment R5C3. The authors had already addressed it in the previous revision.

Comment R5C4. The authors provided a new Supplementary Figure 24 and a new Discussion paragraph detailing the "synergic-epistatic interactions." This is a valuable addition that significantly strengthens the manuscript.

Comment R5C5. The authors have revised Fig. 6e by adding color labels that correspond to the SmEPS1 gene clusters shown in Supplementary Fig. 20, thus addressing my comment.

Comment R5C6. The authors have performed VIGS for SmRoq1 and updated the Results and Methods sections (Lines 497–504, 533–540, and 913–916), as well as revised Fig. 6g and 6h accordingly. The figure provided in the rebuttal letter but not in the supplementary materials suggests that, for each VIGS fragment, there are multiple transcripts with high enough homology to be silenced by the SmEPS1 and SmRoq1 VIGS fragments (see eg Fernandez-Pozo et al Molecular plant, 8(3), 486-488. Also, the reader has no clue on which of these transcripts are recognized by the primers used for the RT-PCR. Finally, the exact genotypes of the 'S065', 'S092', 'S050' accessions used for VIGS whose genotypes are not specified. All these shortcomings make the results still difficult to interpret. The authors should, for each of the genotypes above: i) in a new table indicate the genes homologous to the VIGS fragment and their levels of homology; ii) in table S14 indicate which of the genes are recognized by the RT-PCR oligonucleotides; iii) discuss the results of the VIGS experiment in view of the above information.

Comments R5C7-R5C10. The comments were satisfactorily addressed.

While the authors perform these revisions, they should also address the inconsistencies in Loci returned by GWAS. The manuscript's narrative is built around the 12.4 Mb inversion on Chr10 as the primary example of an SV-trait association. The text explicitly states other loci are "less significant", and indeed Fig 4a seems to support this claim. However, in Table S8, at least one SV-GWAS hit (Chr05_81641732) has a comparable, if not higher significance. Why isn't it shown in Fig 4a? This is very confusing. The authors should revise the Manhattan plots in the main figures to match the data given in supplementary tables both for SNPs and SVs, and present in supplementary figures the comparable results for PGGB-GWAS (see comment R5C2).

Version 4:

Reviewer comments:

Reviewer #5

(Remarks to the Author)

The authors have made substantial efforts to fully address all my comments, except for data availability, which is presently partially addressed.

R5C1 – Scope of the “sub-pangenome” - Addressed. The term “sub-pangenome” is now used consistently, and the Discussion explicitly acknowledges the strong geographical bias of the panel and the need for broader sampling (India, Middle East, Africa). This is an appropriate clarification.

R5C2 – Reference-free PGGB-GWAS - Addressed. The authors now present a reference-free PGGB-based GWAS alongside the reference-based (GATK/SVGAP) analyses. Complete association results for SNPs, InDels and SVs are reported for fruit color and BW resistance (Suppl. Tables 8, 10; Suppl. Fig. 14). High concordance between methods is demonstrated, and all key candidate loci (SmMYB1, SmCYP82D47, SmEPS1, SmRoq1) are supported by both approaches. The previous ambiguity regarding the nature and comparability of the Manhattan plots has been resolved.

R5C3 - Already addressed in the previous revision.

R5C4 – Epistatic architecture of BW resistance - Addressed. The new analysis of synergic/epistatic interactions among SmCYP82D47, SmEPS1 and SmRoq1, supported by a new supplementary figure and expanded Discussion, significantly strengthens the manuscript.

R5C5 – SmEPS1 gene clusters in Fig. 6e - Addressed. Fig. 6e is now correctly annotated and consistent with Supplementary Fig. 20.

R5C6 – VIGS experiments - Addressed. Additional VIGS experiments for SmRoq1 were performed. The authors now clarify (i) homology and potential off-target effects of VIGS fragments, (ii) RT-PCR primer specificity, and (iii) the genotypes and rationale for the accessions used (S065, S092, S050). While VIGS targets multiple homologs, this is now explicit and properly discussed, making the results interpretable.

R5C7–R5C10 - Addressed satisfactorily in the previous version.

Methodological novelty vs. recent literature - Addressed. The manuscript now properly cites and contextualizes Gaccione et al. (2025) and Yu et al. (2025), reframing its contribution as complementary and trait-focused rather than methodologically pioneering.

GWAS narrative consistency (Chr10 inversion vs other loci) - Addressed. Inconsistencies between Manhattan plots and supplementary tables have been corrected. The misleading claim that other loci are “less significant” has been removed, and the Chr10 inversion is now correctly interpreted as a hitchhiking signal linked to SmMYB1, not as the sole major locus.

Data availability. Partially addressed. An accession number for raw sequencing data, assemblies and annotations is

provided (CNGB Project CNP0006177). However, the project is meant to go public on 2026-08-27. The data must be made public by the time the manuscript is published. Also, the full variant callsets and pangenome graph files (VG/PGGB) should be made public, preferably on a dedicated website like SGN.

To all reviewers:

We respond point-by-point to all reviewer comments below. All original comments from the reviewers appear in black text and our responses are below, in blue text. For each reviewer, we have numbered comments in their reviews with tags (e.g. reviewer 1's first comment is tagged [R1C1]). These tags are included in the manuscript and highlighted in green where appropriate to direct the reviewer to the location of the response. Some tagged comments aren't actually tagged in the manuscript, especially when there is no clear textual anchor (like simple deletions of text or some edits to figures). Corrections of typographical errors and insertions of references aren't tagged.

REVIEWER COMMENTS

Reviewer #1 (Remarks to the Author):

You and colleagues reported the populational and pan-genome analysis of eggplant, an important horticulture crop with both scientific and economic importance. They newly assembled 9 genomes, constructed a graph-genome, investigated the effect of SVs on gene expression, and discovered candidate loci involved in eggplant fruit color and disease resistance. Overall, this work offers important genetic resources to understand the historical evolution and to improve eggplant breeding. However, I have some major concerns, e.g., the quality of SV datasets and the association results of fruit colors.

[R1C0]: *We thank the referee for the careful and insightful review of our manuscript. We respond to your comments point-by-point below.*

Major:

1. The authors used a customized SV calling pipeline to identify SVs between newly assembled genomes and the reference genome. In this pipeline, three commonly used tools including SvABA, Delly and Manta were used, and those SVs supported by two callers were retained. Although this pipeline seemed reasonable, please offer more details regarding the SV statistics among three callers. In fact, I am more concerned about the merging of SVs from different accessions using SURVIVOR. The strategy of this tool was to collapse SVs with mutual overlaps into a non-redundant SV set, which was not sensitive enough when dealing with complex SV breakpoints. Meanwhile, the integrated populational SVs were not genotyped accurately for each accession. The authors can try additional SV mergers and cross compare with SURVIVOR results to increase accuracy of population SV, for example, PanPop (Nat Commun, 2024), a recently developed population SV merger which had been optimized for plant species with relatively high repeat contents.

[R1C1]: We would like to thank the reviewer for the valuable comments, particularly regarding the identification and quality of our structural variant (SV) dataset. We recognize that the quality of the SV dataset is essential for downstream analyses such as population genomic analyses, pangenome graph construction and GWAS. To ensure that we obtained a comprehensive and accurate SV dataset, we employed two complementary strategies for SV identification in eggplant.

First, we used a custom assembly-based pipeline called SVGAP (see **Hu et al., 2025**, GitHub: <https://github.com/yiliao1022/SVGAP>) to call SVs from 15 phylogenetically diverse eggplant accessions in relation to the reference genome (S076). This pipeline allowed us to identify 34,938 deletions and 41,064 insertions (ranging from 50 bp to 100 kb), as well as other complex variants, such as 315 inversions and 164 translocations (> 10 kb) (see **Fig. 3a**). The accuracy of this pipeline has been extensively validated, with experimental verification (see **Hu et al., 2025** and our PCR experiments **Supplementary Fig. 21**) confirming up to 87.5% accuracy and a low false positive rate. Given the broad representation of eggplant germplasm in our selected accessions (from a pool of 3,673 accessions, **Supplementary Fig. 5**), we believe this sample set captures a substantial proportion of the major structural variants in the eggplant population, providing a solid foundation for further studies.

Second, we applied three short-read-based methods—SvABA, Delly, and Manta—to call SVs from 201 resequencing datasets, which have an average depth of 20x, ensuring high-quality SV calls. These methods were chosen based on their superior performance in previous benchmark studies (see Kosugi et al., 2019; Cameron et al., 2019). We apologize for not clearly presenting these methods and data in the original manuscript. In the revised version, we have introduced the details of running these methods (see Line 772-807). We appreciate the reviewer's suggestion to compare different methods for merging results, which has significantly improved our understanding of the SV dataset.

As the reviewer correctly pointed out, the SURVIVOR + Jammin pipeline tends to collapse overlapping SVs into a non-redundant SV set, which is less sensitive to complex SV breakpoints. Consequently, it detected fewer SVs compared to the SURVIVOR + PanPop pipeline (97,485 vs. 140,500). Importantly, the calls from the former are highly covered by the latter (up to 99%, see **Supplementary Fig. 22**). We have reported both SV datasets in the revised manuscript, with the final analysis based on the results from the SURVIVOR + PanPop pipeline.

We once again thank the reviewer for the valuable feedback. We hope our revised analyses and explanations address your concerns and provide a clearer understanding of the methods used to construct our SV dataset.

Reference

Cameron, D.L. et al. 2019. *Comprehensive evaluation and characterisation of short read general-purpose structural variant calling software. Nature Communications, 10(1), p.3240.*

Kosugi, S. et al. 2019. *Comprehensive evaluation of structural variation detection algorithms for whole genome sequencing. Genome Biology, 20, pp.1-18.*

Hu, M. et al., 2025. *Benchmarking, detection, and genotyping of structural variants in a population of whole-genome assemblies using the SVGAP pipeline. doi:*

<https://doi.org/10.1101/2025.02.07.637096>

Zheng, Z. et al. 2024. *A sequence-aware merger of genomic structural variations at population scale. Nature Communications, 15(1), p.960.*

2. The authors applied association analysis to investigate the genetic basis of eggplant fruit color. They concluded that the large inversion, 6-bp deletion in coding regions, as well as the gene expression variation of MYB1 gene were all strongly associated with fruit color. Chromosome inversions may have impact on gene expression, especially for those genes that near breakpoints. However, inversions have less impact on gene coding regions only if the coding regions overlapped with breakpoints, while it seemed that the coding variation had strongest evidence associated with fruit color (Fig. 4g)? I suggest the authors to re-organize these results to make more clarification regarding the inference of potential causal variant (coding or regulatory) of this trait.

[R1C2]: *We thank the reviewers for raising this important concern, which has encouraged us to reconsider the relationship between the inversion and fruit color, as well as other genomic variants previously identified by us and others (You et al., 2023; He et al., 2019; Zhou et al., 2022), such as the 6-bp deletion in the coding region of SmMYB1 and its association with fruit color. Indeed, there is no direct evidence to suggest that the inversion is the causal variant underlying the formation of purple fruit color in eggplant, as it does not affect any coding regions of known genes related to fruit color.*

While we observed that eggplant accessions carrying this inversion have significantly higher expression levels of SmMYB1 compared to accessions without it based on 111 RNA-seq data,

suggesting that the inversion may influence *SmMYB1* expression (see Fig. 4e). We further divided the accessions without the inversion into two groups: one group with purple fruit and the other with non-purple fruit (e.g., green or white). We found no significant difference in *SmMYB1* gene expression between the accessions with and without the inversion when the fruit color was purple (see Fig. 4f). However, a significant difference in *SmMYB1* expression was observed between the green or white fruit group and any of the purple groups, regardless of whether they carried the inversion (see Fig. 4f). This pattern was further confirmed by qRT-PCR analysis (see Supplementary Figure 15). Therefore, these results suggest that the inversion may have no effect on *SmMYB1* expression.

Moreover, our analysis of gene expression across all the inversion regions did not reveal any other genes associated with the inversion (see Fig. 4e), further indicating that this inversion may not affect genes related to purple fruit color. This conclusion was also supported by the eGWAS analysis, but we didn't detect any confident signals. Therefore, the eGWAS result was not added in the manuscript.

Given that we were unable to find direct evidence linking the inversion to fruit color, we reconsidered the population history of this inversion. The association between this 12.4 Mb inversion and fruit color is likely due to a founder effect (which is also suggested by reviewer 2 and reviewer 3), where the inversion may have originated from a few founder lines and subsequently spread to higher frequencies in specific populations or regions, such as South China, where purple fruit is preferred. This hypothesis is supported by the narrow genetic diversity of the eggplant accessions carrying the inversion (see Fig. 5b) and the long homologous haplotype blocks across the inversion region (see Fig. 5e). Based on this hypothesis and the available evidence, we propose an evolutionary model for the inversion and the evolution of purple fruit color (see Supplementary Figure 20). We also acknowledge that including more eggplant germplasm from other geographic regions could further validate this hypothesis and provide a clearer understanding of the evolution of purple fruit color in eggplant. We also believe this inversion could serve as an example for studying and interpreting the relationship between inversions and phenotypic traits. As suggested by the reviewer, we reorganized our statement of the relationship between the inversion and fruit color, see changing in the Result section at (Line 408-416) and Discussion section at (Line 602-607). We thank the reviewer again for highlighting this important issue and for agreeing with our interpretation of the inversion and its potential relation to fruit color.

Reference

You, Q. et al. 2023. Mapping and validation of the epistatic and genes controlling anthocyanin biosynthesis in the peel of eggplant (Solanum melongena L.) fruit. Horticulture Research, 10:uhac268

Zhou, X. et al. 2022. Integrated metabolome and transcriptome analysis reveals a regulatory network of fruit peel pigmentation in eggplant (Solanum melongena L.). International Journal of Molecular Sciences, 23(21), p.13475.

He, Y. et al. 2019. Comparative transcription analysis of photosensitive and non-photosensitive eggplants to identify genes involved in dark regulated anthocyanin synthesis. BMC Genomics, 20, 678.

3. The association analysis of SVs and gene expression was performed by RNA-seq data of only 20 accessions (Line 227-247). However, the sample size was a little bit small to have high-confidence conclusions regarding SV effects on gene expression. To meet the standard of a high-quality publication in Nat Commun, I suggest the authors to further incorporate more populational RNA-seq data (>100 accessions) which can better resolve the causality between SV and gene expression, as well as to help the annotation of GWAS results through eQTL mapping.

[R1C3]: *We would like to thank the reviewer for this valuable suggestion. As recommended, we have collected additional RNA-seq data from the fruit, bringing the total to 111 eggplant accessions (Line 272). These new RNA-seq data have been submitted to a publicly available dataset and are accessible for community use (see CNGB database under Project accession number CNP0006177). With this expanded dataset, we were able to confirm our previous findings, providing greater confidence in the results that a substantial number of structural variants (SVs) have a significant effect on gene expression in eggplant, especially the SVs that are very close to genes (see Supplementary Table 7, Fig. 3h and 3i). Additionally, we used the new RNA-seq data (for fruits) to conduct eGWAS analysis for fruit color. However, we did not obtain a confident signal. Therefore, the eGWAS result was not added in the manuscript.*

Minor:

1. The pan-genes in this work was constructed by comparing 9 newly assembled and 5 previous published genomes. The authors should be aware of the potential biases of pan-gene construction originated from different gene annotation methods.

[R1C4]: *Thank you for this insightful point! We have added two more newly assembled and one more recently published genomes, adding up to 11 newly assembled and six published genomes. It is*

a good point that different annotation pipelines may lead to different gene models, therefore causing bias in the gene-based pangenome construction. To ameliorate this question, we carried out two versions of the gene-based pangenome construction. Version 1 used gene annotations from all 17 genomes (11 de novo + 6 published) with 6 published annotations (**Figure 2c-h**), while version 2 only used gene annotations from the 11 de novo genomes with the same annotation pipeline (**Supplementary Fig. 8**). We noticed that while the numbers of different gene family classes were slightly different between the two versions, the general conclusion and trend were quite consistent. For example, the order of Core, Softcore, Dispensable, and Private gene families when comparing gene expressions and Ka/Ks values, and the enriched GO terms associated with “defense” remained similar between the two versions. However, for the Ka/Ks analysis, version 2 was able to separate Dispensable and Private gene families well (significantly different), while this comparison was not significantly different in version 1. This may imply that utilizing gene annotations from the same pipeline may be more accurate/comparable, alternatively, this could be due to the different number of genomes being used.

We have presented both versions in the manuscript, with version 1 presented in **Figure 2c-h** and version 2 presented in **Supplementary Fig. 8**. Considering that the gene-based pangenome was not utilized in the following gene mining regarding fruit color and bacterial wilt disease, plus when it comes to comparison of specific candidate genomic regions associated with bacterial wilt resistance among different genomes (11 de novo), we did manually correct all gene models within these regions using GSaman and transcriptomes from five different organs. We believe this bias should have minimal impact on the major conclusion and biology of the current study. We have updated the manuscript (Line 182-185).

2. Line 123-124, the genetic diversity (π) was unconventionally high in eggplant (0.181, 0.187). It was more common to see an estimation of genetic diversity between 1.0×10^{-3} to 6.0×10^{-3} among major crops including rice, maize and millet (e.g., Nat Genet 54, 1736–1745 (2022)).

[R1C5]: Thank you for this point. The genetic diversity (π values) reported in the current study (0.181-0.323) did look relatively higher than that reported previously in several major crops, including rice (0.024-0.030) (Huang et al., 2012), maize (8.47×10^{-3} to 1.04×10^{-2}) (Chen et al., 2022), soybean (0.0012-0.0029) (Zhou et al., 2015), and millet (0.00042-0.00067) (Chen et al., 2023). Due to a limited number of studies mentioning π in eggplant, we could only find two previous studies that reported π values ranging between 0.03~0.10 in African wild eggplants (Omondi et al., 2024), and between

0.20~0.36 for an eggplant worldwide diversity panel from different geographical areas (Barchi et al., 2023). Therefore, we believe the relatively higher π values were most likely due to species difference. We have updated this information in the manuscript (Line 131-133).

References

Barchi, L. et al. 2023. Analysis of > 3400 worldwide eggplant accessions reveals two independent domestication events and multiple migration-diversification routes. *The Plant Journal*, 116(6), pp.1667-1680.

Chen, L. et al. 2022. Genome sequencing reveals evidence of adaptive variation in the genus *Zea*. *Nature Genetics*, 54(11), pp.1736-1745

Chen, J. et al. 2023. Pangenome analysis reveals genomic variations associated with domestication traits in broomcorn millet. *Nature Genetics*, 55(12), pp.2243-2254

Huang, X. et al. 2012. A map of rice genome variation reveals the origin of cultivated rice. *Nature* 490, 497–501.

Omondi, E.O. et al. 2024. Landscape genomics reveals genetic signals of environmental adaptation of African wild eggplants. *Ecology and Evolution*, 14(7), p.e11662.

Zhou, Z. et al. 2015. Resequencing 302 wild and cultivated accessions identifies genes related to domestication and improvement in soybean. *Nature Biotechnology*, 33, 408–414.

3. Line 187, it should be 'By comparing the other 13 genome assemblies against the reference genome of S076'.

[R1C6]: Thank you for this suggestion. We revised this sentence to "By comparing the other genome assemblies against the reference genome of 'S076'" in the manuscript (Line 215-216).

4. The impact of large inversion on recombination and linkage disequilibrium should be further discussed.

[R1C7]: We thank the reviewer for raising this point. Generally, inversions suppress recombination, leading to long regions of linkage disequilibrium (LD). In our previous analysis of LD across the inversion, we made a mistake in the calculation by separating all accessions into two groups: one with the inversion and the other without. While we found that the inversion group showed a higher decay in LD than the non-inversion group, this was surprising given that the inversion group displays a narrower genetic diversity than the non-inversion group.

Upon closer examination, we found that the subpopulation of eggplant accessions carrying the inversion exhibited a long, homogeneous haplotype with very few SNPs (around 10% of the original

SNPs detected across all accessions, 565 vs. 654,432). We mistakenly used these SNPs for the LD calculation, which resulted in an unexpectedly higher LD, and this was incorrect. We have now corrected this by examining the haplotype structure of the region. Our updated analysis revealed that the inversion forms nearly a single haplotype across the inversion region, suggesting that very few recombination events have occurred since the inversion originated. Consequently, the long homologous haplotype leads to a long LD region. We have updated these results in Fig 5e and Fig 5f, as well as the Discussion section for more discussion at Line 596-607.

5. Line 344, the authors found a 15-fold increase of genetic diversity near breakpoints of inversions. If this inversion encountered consistent artificial selection during breeding, the genetic diversity of this inversion should exhibit significant reduction due to bottleneck effects. I mean, the authors should be careful to these types of observations which might be originated from the mis-mapping of paralogous reads resulted from SVs.

[R1C8]: *We thank the reviewer for raising this concern. Regarding the inversion, we now believe that it likely originated from a small number of purple-fruited eggplant accessions (founder effect), and its increasing frequency in the eggplant population can be attributed to consistent artificial selection during breeding, which aligns with the reviewer's suggestion. We completely agree with the reviewer's point that mis-mapping can lead to overestimating genetic diversity near the breakpoints of SVs. In our case, although we cannot entirely exclude this possibility, we believe it does not affect our conclusions. First, the increased genetic diversity is not confined to the breakpoint region but extends across the entire chromosome and the whole inversion segment. Second, the heightened diversity around the breakpoints of the inversion extends over several megabases. Given that the inversion is relatively recent, the sequences at the breakpoints have not changed significantly. To better convey this, we revised the sentence to: "Our analysis of nucleotide diversity (π) on chromosome 10 revealed that the ancestral group (i.e., without the inversion) exhibited higher genetic diversity across the entire chromosome—approximately twice that of the inversion group. Notably, this difference was even more pronounced within the inversion region and its surrounding breakpoints (**Fig. 5b**)."* (See changes in Line 390-394). Additionally, to strengthen our results, we performed a haplotype analysis and found that accessions carrying the inversion almost form a single haplotype with very few variants (Fig. 5e). This finding further explains the significantly higher genetic diversity in the non-inversion group compared to the inversion group. We hope the reviewer is satisfied with our changes.

Reviewer #2 (Remarks to the Author):

I found the manuscript beautifully detailed and compelling that there is tremendous scientific value in pangenome studies. The use of fruit color and disease resistance are excellent examples. My recommendation for major revision is because of the misportrayal of the work as encompassing global or at least dominantly Asian biodiversity of eggplant but really collections are mostly from China (a major place of great interest!) and the ways eggplant accessions are lumped into larger groupings like countries for some and continents for others makes the study feel inherently biased and hard to follow or trust. The authors should consider how they are categorizing the germplasm and arriving at their conclusions. To me, the value is in the pangenome and GWAS work, not in the domestication confirmation part, which is where the bias and lack of clarity on what the results mean really shows through.

[R2C0]: *We are grateful to reviewer #2 for the positive assessment of our work and critical remarks. We carefully consider and respond to the reviewer's concerns and comments below.*

My specific comments are below. Thank you for the opportunity to review this manuscript.

1. line 44: change 'or' after SE Asia to 'and'

[R2C1]: *Thank you for your comment! We have changed the 'or' to 'and' after SE Asia (Line 44).*

2. line 47: about *S macrocarpon* and *S aethiopicum*, they are now cultivated around the world as well. I've seen plenty of *S. macrocarpon* growing in China (e.g. Guizhou) and both species are getting more popular in the Americas as they are consumed by people with African heritage such as Brazil's massive African diaspora. Please revise to explain they are not just locally cultivated. You may also wish to mention *S. macrocarpon* is consumed for its leaves as well as its fruits, and *S. aethiopicum* is a popular ornamental crop looking like pumpkins in floral bouquets.

[R2C2]: *Thank you for pointing this out and a great suggestion! We have revised accordingly by adding "S. aethiopicum and S. macrocarpon are cultivated around the world as well". And we also added the information about these two cultivated eggplant species as suggested: "S. aethiopicum is a popular ornamental crop looking like pumpkins in floral bouquets, and S. macrocarpon is consumed for its leaves as well as for its fruits." (Line 47-49)*

3. line 63: Break up the paragraph into two starting with 'the concept'

[R2C3]: *Thank you for this great suggestion! We have splitted the paragraph into two (Line 66-67) to make our manuscript more logical.*

4. line 64: 'the full' is impossible. 'much more' should replace 'the full'. Let's not date ourselves....

[R2C4]: *Thank you for pointing this out! We have replaced 'the full' with 'much more' (Line 68).*

5. line 65: obviating  reducing

[R2C5]: *Thank you! We have replaced 'obviating' with 'reducing' (Line 68).*

6. line 75: I don't think the goal of 'to enhance the power of genetic discovery' is crisp enough. Simply saying 'to enable domestication and artificial selection studies to be enhanced with less bias' or something similar is more clear and direct.

[R2C6]: *Thank you for pointing this out! To make the sentence more clear and direct, we revised the sentence as "to enable domestication and artificial selection studies to be enhanced with less bias in eggplant" (Line 79-80).*

7. line 104,105: This sentence is in direct conflict with the sentence on lines 88-89. You mainly have China and SE Asia representation of cultivar diversity, not worldwide phenotypic and genetic diversity.

[R2C7]: *We would like to thank the reviewer for this important comment! It is true that the representativeness of our accessions is limited. To investigate to what extent our assemblies and sequencing data could represent the genetic diversity of eggplants, as suggested by the comment **[R3C2]** from Reviewer#3, we have constructed a new phylogenetic tree (**Supplementary Fig. 5**) by incorporating the sequencing data for a worldwide panel with over 3400 SPET-genotyped accessions (Barchi et al., 2023) and the re-seq data for 25 accessions (Barchi et al., 2021). The result showed that the accessions from our study basically covered most of the major branches of the tree. However, the limitation is the few number of accessions for some branches. We have deleted "worldwide" in this sentence (Line 108).*

Reference:

Barchi, L. et al. 2023. Analysis of >3400 worldwide eggplant accessions reveals two independent domestication events and multiple migration-diversification routes. *The Plant Journal*, 116, 1667–1680.

Barchi, L. et al. 2021. Improved genome assembly and pan-genome provide key insights into eggplant domestication and breeding. *The Plant Journal*, 107, 579–596.

8. line 125: decreased at 69kb in which group?

[R2C8]: Thank you for this point. The LD among SNPs rapidly decreased at 69 Kb ($r^2=0.5$) in the population of 201 accessions. We have added this information (Line 134).

9. line 128: South Asia is included in C1 too. If South America is pointed out, SE Asia should be pointed out. I also don't understand how the C2 and C3 could be parsed out of PCA. There is no explanation of a geographic pattern or region that is consistent with the map. The clusters are also not separated. It is not ideal to group them. Seek a less subjective grouping system.

[R2C9]: Thank you for this insightful point! We apologize for this confusion. It was hard to visualize the separation of C1, C2, and C3 in the PCA figure, probably due to the inclusion of outgroup samples, which made the samples stacked together. We have provided another Supplementary Figure (**Supplementary Fig. 2**) without outgroup samples, which should make the three groups more separable. This grouping was actually summarized based on overall consideration of results from the phylogenetic tree, population structure, geographical information, as well as the PCA (**Figure 1, Supplementary Fig. 1-3, Table S1**). More details regarding the geographical information of the samples have been updated in the manuscript (Line 123-128) as follows. Almost all the samples from Southeast Asia (32/34) and South America (11/13) were assigned to the C1 group. The C2 group mainly consisted (74.19%, 46/62) of members from South China (especially Guangdong Province). The majority (80.82%, 59/73) of the C3 group contained accessions from a wide range of 17 Provinces in China. We hope our revisions could address your concerns.

10. Fig 1: I don't understand how admixture could split the populations. I would support these three or four populations based on the phylogenetic tree but there are plenty of accessions in the admixture plot that could be rearranged into different groups based on pop assignment color, like some in the purple-blue group that have more than 50% red. Maybe explaining the decision making process

would be helpful here. I am also curious if you go with $k=4$ if the light blue are a clade of their own in the phylogeny. If so, it would be nice to highlight on f.

[R2C10]: *Thank you for this great point! It is interesting and true that if we go with higher K values, such as $K=4$ or $K=5$, more sub-groups may show up as their own clusters in the phylogenetic tree. Please kindly refer to our response to **[R2C9]**, this grouping was actually summarized based on overall consideration of results from the phylogenetic tree, population structure, geographical information, as well as the PCA (**Figure 1, Supplementary Fig. 1-3, Table S1**). It seems that the grouping with three groups could be reasonably explained by the geographic information. (Line 123-128)*

11. line 159: a matter of style, but it would help readers to say A total of 16,512 (44.81%) core, 3570 (9.69%) softcore, 16,...dispensable, and 391 (1.06%) private gene families were obtained ().

[R2C11]: *Thank you for this comment! We have revised this sentence accordingly as suggested (Line 171-173).*

12. Fig 2a: these colors are similar to the C1-C3 purple and green. I suggest you change the colors so they don't look similar. The colors may also not be good for colorblind people. What about a dark red-orange for new genomes and black for published genomes?

[R2C12]: *We apologize for this unfriendly color pattern. As suggested by the reviewer, we have now changed the color of **Figure 2a** to dark red-orange and black, and also updated the colors of **Figure 2** in the revised manuscript.*

13. Fig 2: I don't understand the calculations of GO counts and p values with fold change. What is being compared to what for examining representation as fold enrichment? The legend could be clearer if a little longer for this section.

[R2C13]: *Thank you for this point! We have made more clarifications in the figure legend to make it clearer. The GO enrichment analysis was carried out by comparing the GO terms associated with dispensable genes to the background of all GO terms associated with all genes in either the 17 genomes (version 1, **Figure 2h**) or the 11 genomes (version 2, **Supplementary Fig. 8f**). GO counts indicate the number of genes associated with a certain GO in dispensable genes. Fold enrichment*

score was calculated as $(\text{No. of dispensable genes with a certain GO} / \text{No. of dispensable genes}) / (\text{No. of all genes with a certain GO} / \text{No. of all genes})$. The Figure legend has been updated (Line 205-208).

14. Fig 3: Here too I cannot easily see the differences in shade. Please choose colors and shades that are more distinguishable from each other.

[R2C14]: *Thank you for pointing this out. We have revised the colors in Figure 3.*

15. line 315: 'Known color-related genes are depicted.' These could be more specifically described as the genes in the Hydroxycinnamic acid synthesis pathway that take from anthocyanin biosynthesis precursors. C4H is the known key enzyme shunting molecules to HCAAs or pigments. I would just advise the authors to make sure they highlight the literature on these pathways and not to oversimplify as color-related genes.

[R2C15]: *Thank you for this comment! We apologize for this inappropriate statement. We have rephrased this sentence as "Known anthocyanin biosynthesis pathway-related genes are depicted" (Line 357). We also provided more details of these genes including their literatures in **Table S8**.*

16. Fig 5a: I cannot easily see the differences between uncertain and other countries but I think this section needs some reconsideration. You have purple arising in another country very early in the tree, both with and without the inversion. The authors don't acknowledge where that came from. It was favored in China but also most accessions chosen were purple eggplants in China -- look how few non-purple eggplants there are from China. I am not convinced the inversion arose in China. It looks like it arose in multiple places. It does seem to be favored in China, so I am in agreement with the authors on that. What would make this more interesting is to get more fine-scale, where the purple eggplants without the inversion from China (the bottom cluster) could be associated with a geolocation (province level? Latitude? Ecoregion?) and so could the varieties on the top from China that are green. Then regarding the purple inversion accession from other countries, how thoroughly were those countries sampled here? Without naming the countries we cannot know. I bet one is from India and there is only one representative. This study unfortunately suffers from being biased geographically toward China while doing a broader landscape analysis and thus requires more precaution with caveats to not jump to conclusions about what happened in China versus the rest of Asia. Then it suffers from underutilizing the high resolution of Chinese germplasm in the study, not doing a proper landscape-level analysis.

[R2C16]: *We thank the reviewer for the thoughtful comments and valuable suggestions. We acknowledge that our sample is geographically biased towards China. However, we have made an effort to demonstrate how our sample represents the diversity of eggplant by including it in a phylogenetic analysis with over 3,600 accessions, based on SPET genotyping data. While this analysis confirmed that our sample is indeed biased towards Chinese-origin accessions, it also shows that our samples encompass major branches of the global phylogenetic tree.*

Regarding the representation of our sample, we can confidently state that it primarily reflects the Southeast Asia domestication center (Barchi et al., 2023), with China representing one subpopulation of this center. Additionally, we have included resequencing data from 25 more eggplant accessions, mainly from European countries such as Italy and Spain, in the revised manuscript. This new data reveals that the inversion is present in four Italian accessions, which supports the reviewer's insight that the inversion likely arose in multiple regions, making it difficult to attribute its origin solely to China. The high frequency of the inversion in China may be associated with the selection for purple color traits in the region. A model illustrating the evolution of the inversion is provided in Supplementary Figure 20. However, several questions remain open, particularly regarding the origin of the inversion.

As suggested by the reviewer, we also explored whether the inversion correlates with provincial level, latitude, or ecoregion. We did not find evidence to suggest a relationship, as the inversion is randomly distributed across provinces and latitudes. Further studies with more geographically diverse samples and related wild accessions may help clarify the evolutionary dynamics of the inversion.

17. line 477: This stat of 65% of eggplant being grown by South America is incorrect. It seemed high so I calculated on my own, and total yield for 2021 (the most recent FAO data), defined by yield in 100g/ha, was 2322313. The proportion of that for South America is 11.2%. If you use another metric, like tons, South America is not even a percent of eggplant in tons. Asia is 94% of eggplant tons production.

[R2C17]: *We apologize for this grammar error leading to incorrect information. We were trying to emphasize that our samples are mostly from Asian countries, which contributes a large proportion of global eggplant production. We have double-checked the FAO website (<https://www.fao.org/faostat/en/#data/QCL/visualize>), and obtained the global production share of eggplant for Asia (93.5%, 2021-2023, sum). We have revised this sentence accordingly (Line 537-538).*

18. line 486: 'confined to one center' - make sure it's clear if China is expected to be derived from the SE Asia center, a South Asian center, or a China center. This will be confusing to readers and it sounds like the authors are unsure themselves about how Chinese germplasm relates to the domestication narratives.

[R2C18]: *We apologize for this misportrayal of information. In order to not make confusions, we have updated this sentence as “Considering that our samples are primarily from China and Southeast Asia” (Line 547-548).*

19. line 519: 'we ensured a comprehensive'  you cannot ensure a comprehensive catalog of SVs from a species when you are not sampling everything and you are limited by technology. Avoid hyperbole and say 'we attempted to build a comprehensive catalog' or 'we aimed for a comprehensive catalog'.

[R2C19]: *We apologize for this misportrayal of information. Thank you for the suggestion! We totally agree that the limitations in sampling and technology will not be able to ensure a comprehensive catalog of SVs. So we revised the description as “we attempted to build a comprehensive SV catalog” as suggested (Line 581).*

20. line 532: it would be beneficial to relate this to the experiences of breeders. has breeding for purple color ever been a challenge because this inversion was not known? Probably impossible to find such information but I am curious what reports are out there on breeding for various gradations of purple in eggplant.

[R2C20]: *We thank the reviewer for bringing up this point. Selecting for purple color in a breeding program should not pose significant challenges. In our lab, we have developed several genetic markers associated with important genes, such as SmMYB1, SmANS, SmDFR, and SmAPRR2-like, which have been linked to fruit colors, including purple, white, and green. We still anticipate that developing a marker for this inversion could be beneficial for purple color breeding.*

As is well-known, the formation of anthocyanins in eggplant peel color is complex, as it is influenced by light, temperature, and other environmental factors. To our knowledge, there are a few reports

addressing the different gradations of purple in eggplant. However, we previously developed two markers related to dark purple and purplish-red hues over a decade ago (Liao et al., 2009).

Reference

*Yi, L., SUN, B. J., SUN, G. W., LIU, H. C., LI, Z. L., LI, Z. X., & CHEN, R. Y. (2009). AFLP and SCAR markers associated with peel color in eggplant (*Solanum melongena*). *Agricultural Sciences in China*, 8(12), 1466-1474.*

21. line 535: not global, YOUR samples. Make sure to describe accurately. The big flaw in this paper is the misportrayal of the diversity examined herein. Be careful.

[R2C21]: *We apologize for this misportrayal of information. We have deleted the word “global” and checked the manuscript to avoid this kind of description in the whole manuscript (Line 597).*

22. line 544: studied in detail in China. Not especially in China. You didn't look enough elsewhere to say 'especially'

[R2C22]: *Sorry for this misportrayal of information. Since the writing has been re-organized, we have deleted this sentence.*

23. line 658: citation marked twice, no closed parentheses.

[R2C23]: *We apologize for this mistake. We have removed one duplicate citation and closed the parentheses (Line 714).*

24. Data Availability: I see August 2026 as the date the data are made public. Will that change once the manuscript is in press to align with publication?

[R2C24]: *Thank you for this question! Yes! Of course. We will contact the CNGBdb manager to release our data immediately to make our data publicly available.*

Reviewer #3 (Remarks to the Author):

The present study re-sequenced 198 eggplant accessions and de novo assembled nine genomes from them to represent eggplants' genetic diversity. They also identified a large inversion associated with purple fruit color and two genes associated with bacterial wilt resistance. This study provides a valuable resource for future studies, however, I have some concerns.

[R3C0]: *We thank the referee for the careful and insightful review of our manuscript. We respond to your comments point-by-point below.*

Major ones:

(1) As claimed in the title, pangenome analysis was applied in this study, but the benefits of constructing a pangenome are unclear.

[R3C1]: *Thank you for this insightful point! Pangenomes have been playing an increasingly important role in plant genetic and genomic research due to their privilege in representing more genetic diversity and minimizing the bias caused by using a single reference genome (He et al., 2024). Apart from the gene-based pangenome construction, several strategies are available for pangenome construction at the DNA sequence level, such as map-to-pan, whole genome alignment, and graph-based (reference-based or reference-free) strategies. In our study, we used the VG method for pangenome graph construction and subsequent population genotyping using next-generation sequencing short reads. The full utilization of pangenomes could also be challenged by technical limitations, such as genotyping a population using short reads.*

*It is admittedly that the pangenome analysis in the current study did not play a major or direct role in some major analyses such as SV characterization and association mapping for fruit color and bacterial wilt resistance. However, we believe the pangenome was important and did facilitate several findings as below. (1) The original identification of gene number variations of EPS1 benefited from the gene number comparisons in the gene-based pangenome analysis. (2) The analysis of population characteristics of SVs (**Figure 3d**) was only possible because of the availability of the pangenome graph and subsequent populational genotyping. (3) The SVs genotyped based on the pangenome graph were used for SV-GWAS (**Supplementary Fig. 14 d, e and Supplementary Fig. 18**), which facilitated gene mapping and candidate gene identification, especially for bacterial wilt resistance GWAS. The two SVs (SVe and SVr) differentiating gene copy numbers of EPS1 and Roq1 were also identified based on their populational genotypes. (4) The graph-based pangenome and associated*

populational variation data are also a great resource for future mapping numerous other important traits of eggplants, which shall benefit the research community of eggplants.

Reference

He, W. et al. 2024. The developments and prospects of plant super pangenomes: demands, approaches and applications. *Plant Communications*, 6:101230.

(2) To what extent the assembled genomes can represent the genetic diversity of eggplants? Inclusion of the 25 accessions re-sequenced by Barchi et al. (2021) and the over 3400 SPET-genotyped accessions sequenced later (Barchi et al., 2023) may help clarify the representativeness of this genome collection. Especially, it can help estimate the power of the pangenome in SV genotyping with short reads.

[R3C2]: Thank you for this great point! We have downloaded the re-seq data for 25 accessions (Barchi et al., 2021) and sequencing data for over 3400 SPET-genotyped accessions (Barchi et al., 2023), and performed variant calling using S076 as the reference with the GATK pipeline. By using common SNPs, we finally obtained a vcf file containing genotypes of 3,673 accessions at 710 SNP loci, which was further used to construct a phylogenetic tree with IQ-TREE (**Supplementary Fig. 5**). The tree was rooted using *S. incanum* as the outgroup. By highlighting the 11 de novo assembled and 6 published genomes, we observed that this set of 17 genomes could well represent the genetic diversity of eggplants. Although a few branches were not covered by an assembled genome, the re-sequenced accessions within them could also capture a certain proportion of their genetic variations. Overall, we believe the 17 genomes together with re-sequencing data can decently represent a large proportion of genetic diversity of eggplants. This has been updated in the manuscript (Line 156-160).

Reference:

Barchi, L. et al. 2023. Analysis of >3400 worldwide eggplant accessions reveals two independent domestication events and multiple migration-diversification routes. *The Plant Journal*, 116, 1667–1680.

Barchi, L. et al. 2021. Improved genome assembly and pan-genome provide key insights into eggplant domestication and breeding. *The Plant Journal*, 107, 579–596.

(3) The association between the large inversion and the fruit color may not 'underlie' the formation of purple eggplant fruit. I have several questions related to this part of the analysis.

[R3C3]: *We thank the reviewer for raising this important concern. After further consideration and additional analyses, we agree that the large inversion is more likely not the causal variant underlying the formation of purple eggplant fruit. Instead, our new analyses suggest that the increased frequency of the inversion may be a result of genetic hitchhiking associated with the selection for purple color during domestication or breeding processes, potentially driven by a founder effect (as knowledgeably suggested by the reviewer) or artificial selection. We now respond to the other questions one by one below.*

(3.1) Line 276-311: Is the strongest signal for related to SmPAL and SmMYB1 genes on the inversion? Given that nearly half of accessions without this inversion are purple, the association between this inversion and the purple color is likely hitchhiked by the suppression of recombination, because the 6-bp deletion on SmMYB1 can better explain the expression change. There might be more surrounding SNPs and SVs with higher association. How far are the two genes, SmHIPP and SmMYB1, distant from the inversion? To further dissect the function of the large inversion and 6-bp deletion, the two variations can be combined to contrast the haplotype versus phenotype in figure 4h.

[R3C3.1]: *We thank the reviewer for their helpful explanation of the observation and valuable suggestions! GWAS based on SNPs, InDels, and SVs consistently identified a large number of significantly associated loci with the purple color across the inversion region (see **Fig. 4a**, **Supplementary Fig. S14**). At least four genes previously identified as being related to color formation in eggplant, including SmCHI (P -value = $7.78e-09$), SmPAL (P -value = $6.87e-08$), SmMYB1 (P -value = $5.25e-25$), and Sm3GT (P -value = $1.98e-09$), were found to be near or overlap with the significantly associated loci. The strongest GWAS signals we detected were mostly in SmMYB1 (P -value = $5.25e-25$), which is located very close to the breakpoint of the inversion (~5 Kb apart). SmHIPP is located near the other breakpoint (for further discussion on this gene, please refer to our response to **[R3C3.3]** below). We did not detect any loci with stronger associations than those linked to SmMYB1.*

*With additional analyses, we agree with the reviewer that the association between the inversion and purple peel color formation can be better explained by genetic hitchhiking (for further explanation, see our response to your comment **[R3C3.2]** below). We hypothesize that genetic hitchhiking involves not just a single genetic locus, but rather a combination of multiple genes related to fruit color formation, potentially including genes within the inversion, such as SmMYB1, SmCHI, SmPAL, and even genes from other chromosomal regions.*

Accessions carrying the 6-bp deletion are non-purple and are exclusively excluded from the inversion, indicating that the inversion likely arose in accessions without this 6-bp deletion. The combination of these two variants helps to better explain our proposed model of the evolution of the inversion and fruit color in eggplant (see **Supplementary Fig. S20**).

(3.2) Line 335-337: It is more likely that this inversion was inherited from the accessions at the base of the tree, including accessions from both China and other regions, if that tree is rooted. No group information is available in this tree.

[R3C3.2]: *We appreciate this insightful comment. Tracing the origin of this inversion without complete germplasm resources is indeed a challenging task. Although most of the eggplant accessions carrying this inversion are from China, some accessions from other regions (like Italy, see **Supplementary Table 9**), as well as those at the base of the tree, also carry the inversion (see **Fig. 5a**). Therefore, it is reasonable, as the reviewer commented, that this inversion may have originated from accessions at the base and existing for a long time. However, we propose that this inversion is just a new mutation, and originally occur in only a smaller subset of accessions with purple fruit, then it likely underwent expansion in China, though it may have also occurred in other regions.*

*This hypothesis is supported by three lines of evidence: First, the breakpoints of this inversion in the genome are very clear (see **Fig. 4c**), indicating that this inversion is a relatively recent mutation, and the breakpoints remain novel. Second, accessions carrying the inversion exhibit relatively low genetic diversity compared to those without the inversion, likely due to a founder effect or bottleneck. Third, and most importantly, when we used only the SNPs from the inversion region to construct the tree, we found that no accessions at the base of the tree carried the inversion (see **Fig. 5e**). All accessions harboring the inversion clustered together at the bottom cluster of the tree. This suggests a monophyletic group of inversion-harboring accessions. The accessions at the base of the tree, which were constructed using genome-wide variants, may not accurately represent the evolutionary history of local regions. These accessions could be explained by introgression or recombination during breeding and exchange.*

*We apologize for the lack of group information in **Fig. 5a**. In the revised manuscript, we have added this information to the tree, highlighted with different colors.*

(3.3) Line 339-352: No evidence was shown that the allele of SmHIPP linked with this inversion is beneficial to the stress response. The nearly fixed genotype within the inversion-harboring

monophyletic group could be explained by narrow genetic diversity caused by founder effect rather than selection against an inversion, especially when this inversion may not be causal.

[R3C3.3]: *We thank the reviewer for pointing this out and helping to explain the narrow genetic diversity of the inversion group. We apologize for overstating the function of SmHIPP. While we initially proposed that SmHIPP might be associated with stress response based on the functional annotation of this gene and the reported functions of its homologs in Arabidopsis, we now believe this may not be the case. We highlighted this gene because we hypothesized that the higher frequency of the inversion gained during domestication could be beneficial due to its tight linkage with the preferred purple peel color during eggplant breeding, as well as potentially other traits like stress response. However, upon further analysis, we have demonstrated that the inversion has no effect on gene expression across the inversion region (see **Fig. 4h** and **Supplementary Figure 15**). Therefore, we have removed the description of this gene in the revised manuscript.*

*Motivated by the reviewer's comment, we performed additional analysis to trace the origin of the inversion and its relationship with fruit color. 1. We compared the expression patterns between different groups: purple with the inversion vs. purple without the inversion, purple with the inversion vs. non-purple, and purple without the inversion vs. non-purple. These pairwise comparisons confirmed that the inversion has no significant effect on gene expression. 2. We constructed a phylogenetic tree using only the SNPs from the inversion region and found that accessions carrying this inversion almost exclusively form a single group (See **Fig. 5e**), suggesting they are closely related and likely share a common genetic origin. 3. Accessions carrying the inversion show a long haplotype across the inversion region. Given that these accessions exhibit narrow genetic diversity compared to the non-inversion group, we are now more confident in believing that the inversion is not the causal variant, and we propose that the founder effect better explains the fixed genotype of the inversion-harboring monophyletic group. In the revised manuscript, we provide a model to illustrate the evolutionary history of color evolution and the inversion (see **Supplementary Fig.20**).*

(3.4) Line 354-362 & figure 3e & 3f: It is surprising that the LD decays much faster in the INV group than in the non-INV group, given that the nucleotide diversity is much higher in the non-INV group. I would suspect it of being an artificial error caused by using the wrong reference for physical distance calculation. The high LD could disappear if a 'non-INV' genome was used as the reference genome. Besides, since 'S076' was used as reference to talk about variants, the 'INV' group should be those harboring the 12.4-Mb inversion which are different from 'S076'.

[R3C3.4]: *We sincerely thank the reviewer for carefully reading our manuscript and pointing out the surprising result regarding the LD pattern. Motivated by these comments, we reanalyzed the LD pattern and identified a mistake in our previous calculation. We had used VCFtools to calculate the LD, which can only include heterozygous sites in the analysis. In the entire region across the inversion, there are 67,822 SNPs, but for the subgroup of accessions carrying the inversion, only 547 SNPs (less than 1%) were retained, most of which are homozygous sites. This strongly suggests a long single haplotype for the inversion group in this region. Indeed, haplotype analysis across this region confirmed that the inversion forms a long single haplotype, which explains why LD is expected to be very low. When we calculated LD across the inversion using all accessions, we found that the inverted region had much lower LD than both the chromosome-wide level and other regions.*

*In the revised manuscript, we have removed the LD analysis for the subgroup, as we believe there were not enough accurate SNPs in the inversion group to perform a reliable analysis. Instead, we have included an LD pattern for all accessions (see **Fig. 5f**) and a haplotype analysis (see **Fig. 5e**) of the inversion region, which further supports the hypothesis that the inversion is more likely to have arisen from a founder effect.*

*The reference 'S076' we used is the derived state of the inversion. As suggested by the reviewer, in the revised manuscript, we changed the reference to a non-inversion accession 'S126' and performed the analysis in Figure 5 (see **Figure 5**). The results are highly consistent with results when using 'S076' as the reference.*

Thank you again for this insightful comment, we hope you are satisfied with our responses.

I also have minor concerns related to the illustration and interpretation of results and details of Methods:

We address each minor concern one by one as follows:

(4) Need more information to illustrate the association between geographic origin and C1, C2, and C3 grouping, for example summarizing location versus population in table S1.

[R3C4]: *Thank you for this great suggestion! We have further summarized the locations of our samples corresponding to the populations in **Table S1**. This part has been revised as follows (Line 125-128). Almost all the samples from Southeast Asia (32/34) and South America (11/13) were*

assigned to the C1 group. The C2 group mainly consisted (74.19%, 46/62) of members from South China (especially Guangdong Province). The majority (80.82%, 59/73) of the C3 group contained accessions from a wide range of 17 Provinces in China.

(5) Line 130-131: Need more evidence to support the conclusion. Higher genetic diversity doesn't necessarily mean that genetic variants are more important.

[R3C5]: *Thank you for this comment! We agree that higher genetic diversity doesn't necessarily mean genetic variants are more important. Therefore, we have deleted this sentence.*

(6) Line 160-162 & 729-731 & figure 2f & 2g: How were the expression profiles obtained? Are they representing the 14 genomes used in this study? Need further information on how Ka/Ks values were calculated. What was the purpose of using an outgroup in TBtools? What type of gene pairs was compared? How to pair genes if there are paralogs?

[R3C6]: *Thank you for this comment! We have clarified this information in the manuscript. We used the transcriptomes of roots, stems, leaves, flowers, and fruits from the 11 samples (2 newly added) with de novo genomes for calculation of expression profiles, since these data were available and more comparable (Line 197-198). They should be able to represent the 17 genomes (11 de novo + 6 published), since the goal was to compare gene expressions among different gene family classes.*

*The calculation of Ka/Ks was also clarified in the methods part (Line 821-825). We have revised the way of Ka/Ks calculation as follows. First, we obtained ortholog gene pairs for all possible pairwise comparisons among the 17 genomes using JCVI. The paralog pairs were also obtained by comparing each genome with itself using JCVI. Then, we carried out two versions of subsequent Ka/Ks calculations. Version 1 considered only ortholog pairs (**Figure 2g, Supplementary Fig. 8**); version 2 considered both ortholog and paralog pairs (**Supplementary Fig. 7**). In this way, no outgroup was used. It seems that the general trend of the results of the two versions were consistent.*

(7) Figure 3a & 3b & line 252: The figure legend for 3b is unclear. Additionally, if the value 19,824 represents the number of SVs present in PI180, it doesn't align with the length of the bar in 3a. The bars in 3a don't match numbers in supplementary table S4 either. The 'growth rate' is sensitive to the order to include samples, so the decline is artificial.

[R3C7]: We apologize for not making the figure legends and the numbers in each figure and table clearer. The inconsistent numbering is due to our use of slightly different SV sets (i.e. filtered with size or not) for different analyses (e.g., Table and Figure). For example, we used one SV set (with size >50 bp) and another set (with size between 50 bp and 100 kb). To ensure consistency, in the revised manuscript, we have used the size-filtered SV dataset (size between 50 bp and 100 kb PAVs) for all analyses, and make the numbers consistent in the Figures and Tables.

The number in **Figure 3b** is slightly different from **Figures 3a** and **Table S4** because the SVs in **Figure 3b** were obtained using further genotyping methods. To make the presentation of the SV sets more clearer, we now change to report the Shared and Specific Number of SVs in Figure 3b by adding samples with the order based on divergence levels. The shared number of SVs refer to SVs detected in at least two samples, while the specific SVs refer to SV only being detected in one sample. We agree with the reviewer that the "growth rate" is sensitive to the order in which the accessions are included. In the revised manuscript, we have adjusted the order to reflect increasing diversity (see our changes in **Fig. 3b**).

(8) Line 197-199: 'S076' is the reference genome use for comparison, then why is the 12.4-Mb inversion described to happen in 'S076' instead of other accessions in supplementary figure 7? is there evidence supporting the ancestry state?

[R3C8]: We apologize for this confusing presentation. The inversion has been traced to occur in purple fruit eggplant accessions, including the reference 'S076'. The ancestry state can be determined by the outgroup *S. violaceum* (S098). In the revised manuscript, we revised the description in the figure legend of **Supplementary Figure 11** (which is the original **Supplementary Figure 7**). We hope this has addressed your concerns.

(9) Line 199-201 & figure 3c: More details are required for this analysis. Different types of TEs can be harbored in the same insertion or deletion. 'Tandem duplication' is not a type of TE, so it is weird to include it here. Please also define 'the most common SVs'.

[R3C9]: We apologize for not making the results of **Figure 3c** clearer and more detailed. Our aim was to describe the proportion of each type of SV relative to the total SVs. Based on our observations, we categorized the SVs into five classes: LTR-TE-related SVs, DNA-TE-related SVs, TATA satellite expansion and contraction SVs, tandem duplications, and complex SVs that overlap with both LTR- or DNA-based TEs (as recommended by the reviewer), as well as gene-related SVs,

with all other SVs categorized as 'others.' To clarify this classification, we have added a paragraph in the Methods section to describe it in detail (Line 781-785). The number of SVs in each class is also included in the source data.

The term 'most common SVs' was intended to refer to the SVs that appear most frequently in our dataset. In the revised manuscript, we have changed this description to 'most abundant SVs.' We hope that this revision addresses your concerns.

(10) Line 201-207: More details are required in the SV calling and genotyping methods. No information is available on the pipeline (SVGAP, <https://github.com/yilia01022>) used to call SVs based on genome comparison. The parameters used for each software should be provided. Additionally, how many SVs were included in the final graph? Were there any size filtering on these SVs and in the downstream Analyses?

[R3C10]: *We apologize for not clearly explaining the method for SV calling and genotyping. The SVGAP pipeline is now available on GitHub at: <https://github.com/yilia01022/SVGAP>, and the manuscript reports this pipeline as a preprint at <https://www.biorxiv.org/content/10.1101/2025.02.07.637096v1.article-metrics>. We have also comprehensively revised the method section to describe SV calling with the SVGAP pipeline, as well as the parameters for each short-read-based method (i.e., Menta, Delly, and Abavs), which can be found in the **Methods** section at line 792-802.*

For SVGAP SVs callset, we used a total of 41064 insertions and 34938 deletions used for the pangenome construction, with size between 50 bp and 100,000 bp. For the short-reads based SV callsets, we used a total of 140, 500 deletions and insertions (with size > 50 bp and < 100,1000bp) for the pangenome construction. These two pangenomes were all used to GWAS analysis, but only the SVGAP-based pangenome was used for population genomic analysis as it is more accurate and comprehensive.

We thank you again for pointing this out and helping to make our manuscript more detailed. We hope our revisions have addressed your concerns.

(11) Line 208-226: The number of sampled genomes is too small to represent the real allele distribution in the population. A reasonable way is including the 201 re-sequenced accessions.

[R3C11]: *We thank the reviewer for this helpful suggestion. In the revised manuscript, we have added three new genomes to describe the minor allele frequency in the population. These assemblies can represent the major diversity within the eggplant gene pool (see **Supplementary Fig. 5**).*

*Additionally, we performed allele frequency analysis using the 201 re-sequenced accessions for SVs, as suggested by the reviewer. We report the results of the 16 assemblies in **Fig. 3d** and the analysis with the 201 re-sequenced accessions in **Supplementary Fig. 12c**.*

(12) Line 229-230: Like figure 3c, definition of the SV type classification needs to be clarified. And there is no evidence supporting that LTR-TE insertions have higher 'deleterious impacts' on genes.

[R3C12]: *Thank you for raising this concern. To clarify the method for classifying SV types, we have included the defining criteria in the **Methods** section (Lines 781-785). We agree with the reviewer that we did not provide direct evidence to show that LTR-TE insertions have higher 'deleterious impacts' on genes. In the revision, we have modified the description to: 'likely owing to insertional biases or a larger effect on gene expression' (see our changes on Lines 262-263). We hope this revision addresses your concern.*

(13) Line 231-239: Supplementary figure 6 and 7 mentioned are missing. The large sample sizes for comparisons in figure 3g increase the power, which may increase type I error. Please add a bootstrap-based method to provide a confidence interval.

[R3C13]: *We apologize for the incomplete and missing information. In the revised manuscript, we have included the complete information in **Supplementary Figure 13 a,b** (corresponding to **Supplementary Fig 6 and 7**).*

*We agree with the reviewer's comment that our analysis in **Figure 3g** may be susceptible to Type I error. To address this, we have added a bootstrap-based method to provide a confidence interval, as suggested by the reviewer (see our changes in **Fig. 3g** and **Supplementary Fig. 13a**). Thank you again for this helpful suggestion!*

(14) Line 266-270: Signal for SmTT8 is absent in figure 4a.

[R3C14]: *We apologize for this misportrayal of information in Fig 4a. When we double checked the GWAS candidate genes, we found that the SmTT8 was not in the candidate list. We have removed SmTT8 in the manuscript. We apologize for this mistake.*

(15) Figure 6c: GWAS for different phenotypic data should be plotted separately to improve clarity. These plots can be aligned to compare the consistency of signals.

[R3C15]: *Thank you for this point! The GWAS Manhattan plots for different phenotypic data were also plotted separately as **Supplementary Fig. 16-18**. These plots were aligned and the consistency of signals could be well observed.*

(16) Line 414-434: Are these two selected SVs signaled in SV-based GWAS as described in line 401?

What are the relative locations of selected SVs related to the target gene *SmEPS1*?

[R3C16]: *A great point! We apologize for this confusion. The two SV markers (SVe and SVr, **Figure 6e**) used to differentiate gene numbers for *EPS1* and *Roq1* homologs were different with the two SVs (Chr05:84617337, Chr05:84619808) signaled in SV-GWAS (**Supplementary Fig. 18**).*

*Originally, the top signals of SNP-GWAS were directly located within a gene model (evm.model.Chr05.2618, Chr05: 84613990-84619320) annotated as an *EPS1* homolog in the S076 reference genome (**Figure 6c**). Consistently, SV-GWAS also revealed two significant SV signals within this region (SV_Chr05:84617337, p-value 4.56-e6; SV_Chr05:84619808, p-value 2.89-e8) (**Supplementary Fig. 18**). However, after comparing the haplotypes of these variants between resistant and susceptible accessions, we found that none of them could explain the phenotype well individually, even by fixing the genotype of *SmCYP82D47* on Chr04. Strikingly, when checking the results of gene-based pangenome analysis, we noticed a great variation in the numbers of *EPS1* homologs (0~4) of different genomes. This led us to speculate that a single bi-allelic (or multiple-allelic) variation (SNP, Indel, or SV) may not be able to explain the phenotype very well, and the situation could be more complicated. Therefore, to obtain a clear picture of this region, we manually corrected/annotated all gene models within this region (R1 in **Figure 6e**) for the 11 de novo assembled genomes. Interestingly in S076, instead of harboring a single *EPS1* gene, this region (Chr05:84613990-84619320) actually contains two genes right adjacent to each other or intertwined with each other, including one truncated/pseudogene *EPS1* and a homolog of *Roq1* (previously reported to convey bacterial wilt resistance in tomato). Considering the complexity of this area, it is*

highly likely that *Roq1* homologs may also differ in gene numbers among different genomes. Therefore, we also manually corrected gene annotations for additional four regions (R2-R5 in **Figure 6e**) potentially harboring *Roq1* homologs. To further investigate the potential correlation of *EPS1* and *Roq1* homologs with this trait, we identified two SV markers (*SVe* for *EPS1*, Chr05:84599196; *SVr* for *Roq1*, Chr05:84828646) to differentiate the scenario of abundant/expanded homologs with relatively less homologs for the two genes. These two markers were not among the significant signals in SV-GWAS, likely due to the complexity of structural changes in these regions as discussed above. However, these two SV markers, when considered together, were highly associated with this trait (**Figure 6f**).

In terms of their locations, *SVe* was located at ~14.7 Kb to the left of the original GWAS signaled region (Chr05: 84613990-84619320), while *SVr* was located in another region harboring *Roq1* homolog clusters, ~209.3 Kb to the right of the original region. We have also revised **Figure 6e** so that the locations of *SVe* and *SVr* (colored triangles) relative to *EPS1*/*Roq1* genes can be visualized. All locations of SVs and their associated genes were provided in **Table S11**.

Reviewer #4 (Remarks to the Author):

In this study, the role of structural variation in skin color and bacterial wilt resistance of eggplant was discussed by pan-genomic analysis. In addition, the genetic information of 201 eggplant germplasm was analyzed to provide guidance for genetic domestication of eggplant. The results of this study are detailed and have certain value for basic research and molecular breeding of eggplant in the future. In response to this study, the reviewers have the following questions:

[R4C0]: We thank the referee for the careful and insightful review of our manuscript. We respond to your comments point-by-point below.

1. What was the basis for the selection of S076 as a reference genome in the population structure analysis? We know that the authors assembled nine genomes from scratch, and I think the authors need to explain the basis for selecting reference genomes for population structure analysis.

[R4C1]: Thank you for this great point! The reason why we selected 'S076' as a reference genome in our population structure analysis was because 'S076' is a representative breeding line with purple fruits typical in South China. Moreover, 'S076' showed high resistance to bacterial wilt disease of eggplant (Gong et al., 2024), which may be beneficial for identifying disease resistance genes by serving as a reference genome. We have clarified these in Lines 117-118.

Reference

Gong, C. et al. 2024. A QTL of eggplant shapes the rhizosphere bacterial community, co-responsible for resistance to bacterial wilt. Horticulture Research, 11(2), p.uhad272.

2. Please refine the genome assembly strategy and the different technical approaches to obtain a specific description of the amount of data available.

[R4C2]: Thank you for this important comment! We apologize for the misportrayal of information. We have refined this section accordingly (Line 708-710).

3. It is suggested that each attached figure in Figure 2 should choose a color match with more vivid color contrast, which will help readers to distinguish better.

[R4C3]: We apologize for this inappropriate color pattern. We have revised the colors in Figure 2.

4. For the obtained structural variation, the author needs to consider the molecular verification of some representative important structural variation to prove the reliability of the data.

[R4C4]: *We thank you for this valuable suggestion! As recommended by the reviewer, we applied two methods to verify the reliability of our SV dataset. First, the SVGAP pipeline has been extensively tested and shown to provide high accuracy and low false positive rates in SV detection (see Hu et al., 2025, GitHub: <https://github.com/yiliao1022/SVGAP>). Second, we randomly selected 24 SVs for PCR verification and achieved 87.5% accuracy (**Supplementary Fig. 21**).*

5. Based on numerous previous reports, fruit shape of eggplant is also an important selected trait. Has the author tried to analyze structural variation and fruit shape selection of eggplant?

[R4C5]: *Thank you for this suggestion! We agree that fruit shape is also an important trait for eggplant. However, it was not included in the association mapping part, since we did not phenotype this trait. We will work on it in this coming growing season, and perform more analyses in the future.*

6. Regarding the classification of fruit color, we believe that the distinction between purple and non-purple is too simplified. Currently, there are abundant reports on the color of eggplant fruits, and the results obtained by the author are not enough in terms of novelty compared with previous reports. This paper systematically expounds the formation and domestication of eggplant peel color.

[R4C6]: *Thank you for this insightful point! We agree that the distinction between the purple and non-purple color of eggplant fruit might be simple, but it is actually a stable and reliable way to score the color. As we know, anthocyanins formation of the peel color is complex in eggplant, because anthocyanins could be affected by light, temperature, and other environmental factors. It would be less stable if we phenotype using “degrees” (high to low). To contrast, we measured anthocyanin pigmentation in a way as a qualitative trait, and scored the anthocyanin pigmentation as “1” and non-anthocyanin pigmentation as “0”. In this way, the presence or absence of anthocyanins was actually not affected by environmental factors in different planting seasons.*

It is admittedly that while we identified several novel loci for bacterial wilt resistance, the major candidate gene (SmMYB1) for fruit color on Chr10 is a previously known gene. Despite this, we were able to identify a 12.4 Mb inversion at a high frequency (50.7%) within the population, which is associated with fruit color, likely due to genetic hitchhiking. We believe this is of great biological

importance, since it could serve as an example of large SVs associated with biological traits. Moreover, the association signals on other chromosomes also provide candidate genomic regions/variants for future investigations.

7. The description of bacterial wilt inoculation methods and evaluation criteria as well as biological and technical duplication is not clear enough, please add a detailed description.

[R4C7]: *Thank you for this great comment! We apologize that some of these descriptions were shown in the Results part instead of in the Methods section. To be more specific, a total of four batches (three batches each with two completely randomized blocks in 2022, and one batch with three completely randomized blocks in 2023) of plants were conducted for bacterial wilt inoculation. The bacterial wilt inoculation followed the method reported previously (Gong et al., 2024). For bacterial wilt disease phenotyping, each seedling was scored as resistant (score 1) or susceptible (either dead or wilt, score 0) consecutively for 5 weeks after infection with *Ralstonia solanacearum* at the 4-5 true leaf seedling stage. We have updated these details in the Methods section (Line 680-687).*

Reference

Gong, C. et al. 2024. A QTL of eggplant shapes the rhizosphere bacterial community, co-responsible for resistance to bacterial wilt. Horticulture Research, 11(2), p.uhad272.

8. This study provides a comprehensive analysis of structural variants (SVs) in eggplant; however, the SV data are not publicly available. The authors should make these data accessible to promote transparency and facilitate further research. Additionally, the description of the methods used for SV detection is insufficient. The authors should provide more detailed information about the methodologies employed for identifying the SVs.

[R4C8]: *We thank the reviewer for raising this point. In the revised manuscript, we have uploaded our SV datasets to <https://github.com/yilia01022/eggplantpangenome>, which include SVs identified from 16 full genome assemblies and up to 200 re-sequenced datasets. These datasets are now freely available for download and use to the public. Additionally, we have carefully revised the Methods section to clarify how the SVs were identified (lines 772-802). We hope our revision addresses your concerns.*

9. One striking finding in this study is the observation that a large inversion is strongly associated with

peel color in eggplant, a phenomenon also observed in other organisms. Although the authors present compelling evidence to identify the causal variants and provide some population genetic analyses to demonstrate the potential artificial selection of this inversion, the current explanation could be further strengthened. I encourage the authors to incorporate additional population analyses or discussion to better elucidate the domestication history and evolutionary significance of this inversion.

[R4C9]: *We thank the reviewer and all the other reviewers for raising this insightful point. Motivated by the reviewer's comment, and with additional analyses, we now have a clearer understanding of the origin and its relationship with fruit color. Our findings regarding this inversion are as follows: (1) the inversion originated from a small number of purple-fruited accessions; (2) these purple-fruited accessions (carrying the inversion) were selected as key breeding materials (founder effect); (3) the inversion has been strongly selected due to genetic hitchhiking with favorable traits (e.g., purple fruit) in certain regions (e.g., China), leading to an increase in its frequency within the population. For further details, please refer to our responses to [R1C2], [R2C16], and [R3C3]. We hope you find our response satisfactory.*

10. To clarify how *SmEPS1* is associated with bacterial wilt resistance, I encourage the authors to provide more analyses, such as comparing the expression and the protein sequences among the different copies of *SmEPS1*.

[R4C10]: *This is a great point! We have performed additional analyses on comparing EPS1 homologs among different genomes (**Supplementary Fig. 19**). By rooting a phylogenetic tree with S098 as an outgroup, the 20 EPS1 homologs were classified to four clusters, corresponding to the four copies of EPS1. Since EPS1 homologs are absent in this region for S027, S054, and S076, no expressions were detected. The only EPS1 homolog in S001 also showed low expression levels. In contrast, the EPS1 homologs for genomes with multiple copies showed much higher expression levels. It seems that EPS1 homologs in Cluster 2 and 4 may play an essential role since their expressions were quite high. However, it requires future in-depth study to find out which copy or copies play a role in disease resistance, since not only the gene numbers, but also the protein lengths vary among different homologs. We have updated the manuscript (Line 658-661).*

To all reviewers:

*We sincerely thank all reviewers for their thoughtful and constructive comments. We are grateful for the time and effort invested in reviewing our manuscript. The suggestions and feedback have been very helpful in improving the quality of our work. In this revision, we have carefully addressed all comments to the best of our ability. Below, we provide a point-by-point response. Reviewer comments are shown in **black**, and our replies are in **blue**. Each comment is tagged (e.g., [R1C1] for Reviewer 1, Comment 1), and corresponding changes are marked in **green** in the revised manuscript. Some minor changes, such as typo corrections, small text edits, or reference updates, are not explicitly tagged in the manuscript but have been implemented as suggested.*

Reviewer #1 (Remarks to the Author):

The authors have done a nice work in addressing my concerns. The manuscript is now much improved. Thank you! For additional suggestion, I will not put the result of two pan-gene versions into the main text, which will reduce the readability of this section. I agreed with the authors to retain the pan-genes of "11denovo + 6published" in Results, while to re-organize the "version2 pan-genes", emphasizing the potential biases of gene annotations into Discussion.

*[R1C1]: We thank the reviewer for the positive feedback and for agreeing with our treatment of the gene-based pangenome analysis. As suggested, we retained the "11 de novo + 6 published" pan-gene set in the Results section of the main text, moved the "version 2 pan-genes" to the Supplementary Results (**Supplementary Fig. 6-8**), and discussed the potential bias introduced by gene annotation in the Discussion (please see changes in **Lines 175, 188-191, 591-594**).*

Reviewer #3 (Remarks to the Author):

Thank you to the authors for the revised manuscript. I appreciate your efforts in addressing my previous comments and enhancing the overall quality of the paper. The manuscript provides valuable genomic resources and presents interesting trait associations. Most of my concerns have been addressed. However, after reviewing this revised version, I still have a few remaining concerns. The authors have made notable improvements in clarifying the representativeness of the assembled genomes, tracing the origin of the large inversion, refining the methodology, and removing inconclusive inferences. Despite these improvements, several concerns remain unaddressed or insufficiently justified.

We sincerely thank the reviewer for the positive evaluation of our previous revision, as well as for the continued interest and thoughtful suggestions to improve our work. The reviewer's constructive and insightful feedback have helped us further enhance the quality of the manuscript. In response, we have done our best to carefully address each comment. We hope the revised version meets your expectations. Below, we provide our point-by-point responses.

I use numbered tags provided by the authors to trace the comments.

[R3C1] This response does not adequately address my concern.

(1) I infer that the authors' statement regarding "benefiting from the gene number comparisons" refers to the results described in lines 447–461. However, this section does not provide substantive evidence from gene-based pangenome gene family data to support the association between gene copy number variation (CNV) and phenotype.

We appreciate the reviewer's comment and apologize for not clearly presenting the logic behind our analysis. The discovery of the association between the CNVs and the bacterial wilt resistance phenotype was not solely achieved through the gene-based or SV-based graph pangenome; however, both datasets played a critical supporting role in facilitating this finding.

*For GWAS analysis, we initially included all three types of variants—SNPs, small InDels, and SVs (≥ 50 bp). All three consistently produced significant association signals in the region on Chr05 that harbors the two target genes, *EPS1* and *Roq1*. Previously, we generated two SV datasets: one from 17 fully assembled genomes using SVGAP, and another from short-read resequencing of a large population. We constructed separate graph-based pangenomes and genotyped SVs across 201 samples, followed by GWAS. Only the SVs from the assembly-based dataset yielded significant signals (see the original **Supplementary Fig. 18** or updated **Supplementary Fig. 18**). This highlights the advantage of using high-quality, reference-grade genome assemblies, as short-read methods often miss a large fraction of SVs (up to 50%) (Cleal and Baird, 2022). As suggested by the reviewer and ensuring consistency, we now combined these two SV datasets for GWAS analysis and received the expected signals (see the revised **Supplementary Fig. 14 and 18**).*

*Although GWAS signals from SNPs, InDels, and SVs all converge on this region, the specific causal variants or genes underlying the phenotypic differences remain unclear. Notably, SNPs and InDels within *EPS1* and *Roq1* did not show strong effects. In contrast, we identified two SVs overlapping these genes and observed copy number differences across assemblies using the gene family-based pangenome, which prompted a detailed comparative genomic analysis. The result suggests that copy number variation (CNV) in *EPS1* and *Roq1* is likely among the causal variations associated with bacterial wilt resistance. This finding highlights the important supporting role of gene-based, graph-based SV representation and multiple high-quality genome assemblies in refining association signals and identifying candidate causal variants. Please see **Lines 464-466, 473-493**.*

*We also recognize that the full potential of the pangenome resource may extend beyond the scope of this study. To address this, we now explicitly discuss its future applications in broader population genomics and trait association studies in the **Discussion** section (**Lines 551-553**).*

Reference:

Cleal, K. and Baird, D.M., 2022. Dysgu: efficient structural variant calling using short or long reads. Nucleic Acids Research, 50(9), pp.e53-e53.

Moreover, the results presented raise concerns about the quality of gene annotation. Specifically, the structure of the gene *evm.model.Chr05.2618* may have resulted from an erroneous fusion of two neighboring genes, *evm.model.Chr05.2786* and *evm.model.Chr05.2787*. This potential annotation error is partially supported by subsequent manual curation (lines 462–465).

*We agree with the reviewer that high-quality gene annotation is critically important for all downstream analyses. However, accurate gene annotation remains a persistent challenge, especially in large and repetitive plant genomes. Acknowledging this, we manually curated gene models within the target regions. This manual correction helped ensure the reliability of our results and underscores the importance of accurate gene annotation in interpreting functional variation and identifying potential causal variants. Changes have been made at **Lines 460-461**.*

Figure 6e illustrates CNV, but the distinction between "R+/R-" needs clarification.

*We apologize for not clearly presenting the definition of R+/R- in **Fig. 6e**. R+ denotes relatively high copy numbers of the *Roq1* gene (i.e., ≥ 13 copies), as seen in accessions such as S076 ($n = 13$) and S027 ($n = 17$). R- represents relatively low copy numbers (i.e., ≤ 11 copies), ranging from 5 to 11 in the remaining accessions. We have revised the legend of **Fig. 6e** and the text to clarify this information (**Lines 485-486**).*

Additionally, based on the gene distribution patterns, genotypes carrying "E+" seem to originate from the same ancestral sequence, and similarly, those with "R+" appear to share a common ancestral sequence. This suggests that resistance in accessions carrying these genotypes may be driven by functional gene variations rather than solely by a potential dosage effect resulting from gene expansion.

*We appreciate the reviewer's insightful interpretation! We agree that genotypes labeled as 'E+' (i.e., those with relatively high copy numbers of *EPS1*) likely originate from a common recent ancestor, as do those classified as 'R+'. However, due to the complex structural nature of this genomic region, reconstructing the precise evolutionary trajectory remains challenging. We thank the reviewer's propose that the observed resistance in accessions carrying these genotypes may be driven by functional variation within the genes themselves, rather than solely by gene dosage effects due to copy number expansion. We have revised the manuscript accordingly to note that the resistance may result from gene dosage effects, unidentified functional variants within *EPS1* or *Roq1*, or a combination of both mechanisms (**Lines 675-678**).*

Furthermore, if the pangenome analysis only contributes to the identification of the EPS1 gene, then the mention of structural variants (SVs) related to fruit color in the title could be misleading.

We thank the reviewer for the helpful comment. The identification of the large inversion linked to fruit color could only be confidently achieved through comparison of multiple high-quality reference-grade genome assemblies. Therefore, the identification of this SV also benefitted from our pangenome resource (i.e. multiple reference genomes). We understand the reviewer's concern, and to better reflect the main findings and address the reviewer's suggestion, we have revised the title to "A sub-pangenome resource reveals large hidden structural variants associated with fruit color and bacterial wilt resistance in eggplant."

(2) The genetic diversity of SVs within the studied populations is one of the key contributions of this study. However, the analysis of this diversity does not substantiate the claim made in the title that the study "uncovers large hidden structural variants associated with fruit color and bacterial wilt resistance."

We appreciate the reviewer's valuable feedback. To avoid potential overstatement and better reflect the actual contribution of our work, we have revised the title to "A sub-pangenome resource reveals large hidden structural variants associated with fruit color and bacterial wilt resistance in eggplant."

(3) There is a substantial discrepancy between the revised Suppl. Fig. 14d and the original Suppl. Fig. 9B. Notably, the authors used two different SV datasets, "SV_panpop" and "SV_SVGAP," in Suppl. Fig. 14d and 14e, respectively. This inconsistency is confusing and does not align with the uniform description of SV detection and genotyping in the main text. Similar inconsistencies are also present in Suppl. Fig. 18. These contradictions undermine the reliability of the analysis.

We appreciate the reviewer's careful examination and valuable feedback. We apologize for the inconsistency and any confusion it may have caused. Some of these figures were vestiges from early versions of the manuscript. In the original submission, we used two different SV datasets—assembly-based and short-read-based—to explore how SV discovery approaches might affect GWAS results.

*In response to the reviewer's concern, we have reanalyzed all relevant figures using the finalized, consistent SV genotyping dataset, which combines both assembly-based and short-read-based SV callsets across all samples. **Supplementary Figs. 14 and 18** have been updated accordingly to ensure consistency with the main text.*

(4) The graph-based pangenome and SV population diversity are highly valuable resources. However, they do not sufficiently address the core concern—how these resources facilitate the identification of phenotype-associated SVs in this study.

We thank the reviewer once again for highlighting this concern. We agree that the graph-based pangenome and SV dataset are valuable resources. Accordingly, we have carefully curated and made these resources publicly available (<https://doi.org/10.5281/zenodo.15877227>) for the broader plant genomics and eggplant research communities, with the hope that they will support future studies in eggplant genetics.

*Although these resources may not appear to act as a major contribution for identifying phenotype-associated SVs in this study, we did initially incorporate them into our GWAS analyses of both fruit color and bacterial wilt resistance (**Lines 313, 445, 461-463**). Their contribution may seem limited because other types of genetic variation, such as SNPs and small InDels, were already sufficient to reveal strong associations with these traits. Nevertheless, the SV dataset played an important complementary role by enabling detailed comparative genomic analyses, which ultimately led us to identify key structural variants. We believe that these resources will be highly valuable for exploring additional traits in future studies. We have revised the Discussion section (**Line 551-553**).*

[R3C3] Fig. 5e provides strong evidence. Please add 'Origin' information as in 5a to the figure to support the inferred origin of the Chinese-type inversion.

*We appreciate the reviewer's helpful suggestion. The 'Origin' information has been added to **Fig. 5e**, and the results are consistent with those in **Fig. 5a**, indicating that this inversion is predominantly of higher frequency in China samples.*

[R3C7] The authors should first carefully review their analyses, finalize the dataset they intend to use, and ensure consistency before re-computing SV distributions and conducting subsequent analyses, including figure generation. This will help maintain the credibility of their results by ensuring that the data used is consistent throughout.

*Thank you for the detailed examination and helpful suggestions. In response, we have finalized the use of two SV datasets in our analysis. The first SV dataset ($n = 75,683$) was generated from 17 high-quality genome assemblies using our pipeline, SVGAP. The second dataset ($n = 187,412$) is a merged set combining the assembly-based SVs ($n = 75,683$) with those identified from short-read resequencing data ($n = 140,500$). We used the first callset for population genomic analysis, as reported in **Fig. 3**. This is because short-read-based SV calling in population samples has several limitations: 1) it tends to detect far more deletions than insertions; 2) it often misses a large proportion of SVs (up to ~50%), especially in repetitive regions, leading to biased chromosomal distributions; 3) it has reduced sensitivity to large SVs and is biased toward shorter ones; 4) its overall accuracy is lower compared to assembly-based approaches. These limitations can significantly affect the interpretation and reliability of population genetic analyses. Therefore, we chose to use the more accurate assembly-based SV dataset for that*

*part of the study. However, to further address the reviewer's concern, we also performed allele frequency analyses using the combined SV dataset for comparison, as shown in **Supplementary Figure 12c**.*

The combined SV dataset from short-read and assembly-based callsets was used to construct the SV-based graph pangenome and applied in downstream GWAS analyses, as these analyses can benefit from a more comprehensive representation of SV diversity. We hope the reviewer finds our resolution and explanation regarding the use of the SV datasets in the manuscript satisfactory.

The revised Figure 3b still does not align with the reported results. According to the results section and Table S4, there are a total of 76K SVs, yet Figure 3b displays fewer than 71K SVs. Additionally, the authors have released two SV datasets containing 75,683 and 140,500 SVs, neither of which matches the reported results. Similar inconsistencies are present in other parts of the study.

*We apologize again for this oversight. The inconsistency arose due to the use of intermediate versions of the SV dataset during different stages of analysis and figure preparation (i.e., pairwise SV calling and merging across samples). In the revised manuscript, we have consistently used the assembly-based SV callset ($n = 75,683$) for the population genomic analyses reported in **Fig. 3**. **Fig. 3b** has been revised accordingly.*

Unless there is a specific reason, which should be clearly explained, SV-GWAS should be conducted using the final SV genotyping dataset rather than one of these intermediate versions.

*As suggested by the reviewer, we have re-conducted the SV-GWAS analyses using the finalized combined SV dataset, replacing the previous results that were based on two separate datasets (i.e., assembly-based and short-read-based SV datasets). The relevant figures, including **Supplementary Fig. 14, 18** have been revised accordingly. We sincerely thank the reviewer again for highlighting this important issue.*

[R3C10] One of the key contributions of this study is the construction of a graph and the precise SV genotyping using vg toolkit, enabling more accurate and comprehensive graph-based SV analysis. However, according to this response, the actual SV-aware population genomic analyses were conducted using SV genotyping data derived from SVGAP, which is based on a single reference genome. This approach does not fully demonstrate the advantages of assembling multiple genomes and constructing a graph, which is a central strength of this study.

We apologize for not clearly explaining the approach we used in the previous revision response. We utilized the SV genotyping data from the SVGAP for our population genomic analyses in order to ensure high-quality and unbiased SV callset (i.e., without bias in size, types, and chromosome distribution) when inferring population genetic characteristics. Additionally, we also

employed SV genotyping data from VG by combining assembly-based and short-read based SV callsets, and corresponding results are reported in **Supplementary Fig. 12c**.

Regarding the issue of using a single reference genome, we kindly refer the reviewer to our response for **R5C3**. We did attempt a reference-free approach, such as using all-to-all assemblies to construct the pangenome graph (e.g., PGGB). However, these methods detected significantly fewer SVs compared to the SVGAP pipeline. Our benchmark survey of several SV discovery and pangenome construction methods revealed that current techniques do not perform well with large and repetitive plant genomes (Hu et al., 2025; [Benchmarking, detection, and genotyping of structural variants in a population of whole-genome assemblies using the SVGAP pipeline | bioRxiv] which has been accepted in *Molecular Biology and Evolution* and will appear soon). Consequently, we ultimately retained the SV results from SVGAP for subsequent population genotyping using VG.

Reference:

Hu, M., Wan, P., Chen, C., Tang, S., Chen, J., et al. (2025). Benchmarking, detection, and genotyping of structural variants in a population of whole-genome assemblies using the SVGAP pipeline. *bioRxiv*, 2025-02.

[R3C16] Please incorporate the genotype data (E/R) from Figure 6f into the dataset presented in Figure 6b. Additionally, the authors should release the SV genotyping results that were consistently used throughout the study, ensuring that they include all relevant accessions.

We appreciate the helpful suggestion from the reviewer. As suggested by the reviewer, we have now incorporated the genotype data (E+/E-, R+/R-) from **Fig. 6f** into **Fig. 6b**. These genotype classifications are presented as pie charts and correspond to the averaged disease incidence rates. Also, we have made the SV genotyping results publicly available as a VCF file, which includes all relevant accessions (<https://doi.org/10.5281/zenodo.15877227>).

Overall, the manuscript has improved, but the emphasis on gene-based and graph-based pangenome analysis has not yet been sufficiently supported.

We sincerely appreciate the reviewer's constructive criticism and comments, which have been instrumental in improving our manuscript. We hope that the reviewer is satisfied with our revisions.

Reviewer #4 (Remarks to the Author):

The revised manuscript has been improved a lot and my concerns have been addressed. Only minor suggestions are given as below:

Thank you for this positive feedback!

1. Authors used two accession names, 'GQIE' and 'HQIE', in Figure 2b, e, and in Figure 4c, d, but they were not defined in the text. Although 'GQIE' most likely refers to 'GUIQIE-1' and 'HQIE' should refer to 'HQ-1315', consistent names shall be used in order to avoid misunderstanding.

Thank you for this comment! We have changed 'GQIE' to 'GUIQIE-1' and 'HQIE' to 'HQ-1315' in Fig. 2 and Fig. 4, and double-checked the whole text to confirm that the names are consistent.

2. I suggest authors immediately release the genomes and sequencing data once the paper is in press.

Thank you for this comment! We will of course immediately release the genomes and sequencing data once the paper is in press. We believe these data resources will benefit the eggplant and Solanaceae research communities.

Reviewer #5 (Remarks to the Author):

Review: You et al performed a pangenome analysis of eggplant, based on 6 published and 11 novel chromosome-level genomes, and short read resequencing of 201 accessions. They then proceeded to compute PAVs, SNPs and SVs using a reference-based approach, construct a graph pangenome, and finally focus on two phenotypes: anthocyanin biosynthesis, where they find a large inversion associated (most likely through a founder effect), but not causally related to a difference in fruit color, and bacterial wilt resistance, where they conduct an eggplant and thorough GWAS analysis of the genes involved, demonstrating the power of using multiple chromosome-scale assemblies. While most of the conclusions are expected, the bacterial wilt work is highly novel. However, the experimental approach presents some serious limitations:

We thank the reviewer for the insightful review of our manuscript. We respond to your comments point-by-point below.

1. The main one, as pointed out also by reviewer 2 (R2C0), is the sampling: the accessions used originate almost exclusively from one of the two recognized domestication centers (SE Asia) and from China, with just one accession from the second domestication center (Indian subcontinent)

2 from Central/Northern Europe, and 2 from the Americas (Fig 1a). Several important domestication/diversity centers (India, Middle East, Southern and Eastern Europe, Africa, Japan and Korea) are not represented, or represented just by one accession in India. This limitation is further exacerbated by the bias in the resequenced accessions, that cluster in a narrow region of the diversity tree (Supplementary figure 5). Thus, I don't think this concern by reviewer 2 has been appropriately addressed. A pangenome should give a more or less complete representation of genes within a species or clade, so this double geographic and genetic bias in the population is a serious limitation. The definition "regional pangenome" or "sub-pangenome" seems more appropriate for the present dataset. This bias in the accessions has a bearing also on other comments by reviewer 2, such as R2C16 (did the inversion arise in China? Its abundance is due to a founder effect?) or R2C19 (the catalog of eggplant SV is not comprehensive). Without a geographically and genetically balanced sample, these comments cannot be addressed.

[R5C1]: We appreciate both Reviewer 2 and Reviewer 5 for raising this concern regarding sampling bias. We agree that this is a potential limitation of our current study. To partially address this, we incorporated resequencing data from 25 additional eggplant accessions previously published by Barchi et al. (2019, 2021) and Gramazio et al. (2019). These samples include accessions from Italy (10), India (4), Spain (3), China (3), Turkey (1), Thailand (1), Sri Lanka (1), Israel (1), and France (1).

After including these samples, the original C1 group was subdivided into two distinct clusters: one primarily composed of samples from Southeast Asia, and the other mainly composed of samples from Europe and South America (Fig.1c, d, f and g). This result confirms the reviewers' concern and further supports findings from previous and current studies that eggplant genetic diversity is strongly correlated with geographic origin (Barchi et al., 2023).

Therefore, we accept the reviewer's suggestion that the current dataset is better described as a regional or sub-pangenome resource. In response, we have:

- 1. Added a new section describing the graph-based sub-pangenome resource (Lines 293-307);*
- 2. Revised the title to: "A sub-pangenome resource reveals large hidden structural variants associated with fruit color and bacterial wilt resistance in eggplant";*
- 3. Clarified how this sub-pangenome helps explain the origin of the inversion related to fruit color and the representation of SVs (Line 325-331);*
- 4. Expanded the discussion on how this resource could contribute to future eggplant genomic research (Lines 551-553).*

We hope these revisions more accurately reflect the scope of our study and satisfactorily address the reviewer's concerns.

References:

Barchi, L., Pietrella, M., Venturini, L., Minio, A., Toppino, L., et al. (2019) A chromosome-anchored eggplant genome sequence reveals key events in Solanaceae evolution. *Scientific Reports*, 9, 11769.

Barchi, L., Rabanus-Wallace, M.T., Prohens, J., Toppino, L., Padmarasu, S., et al. (2021) Improved genome assembly and pan-genome provide key insights into eggplant domestication and breeding. *The Plant Journal*, 107, 579–596.

Barchi, L., Aprea, G., Rabanus-Wallace, M.T., Toppino, L., Alonso, D., et al. (2023) Analysis of > 3400 worldwide eggplant accessions reveals two independent domestication events and multiple migration-diversification routes. *The Plant Journal*, 116, 1667-1680.

Gramazio, P., Yan, H., Hasing, T., Vilanova, S., Prohens, J. et al. (2019) Whole-genome resequencing of seven eggplant (*Solanum melongena*) and one wild relative (*S. incanum*) accessions provides new insights and breeding tools for eggplant enhancement. *Frontiers in Plant Science*, 10, 1220.

2. A second limitation is the outgroup used for the phylogenetic analyses, composed by 4 *S. aethiopicum* and 1 *S. violaceum* resequenced accessions, of which the latter was also assembled to chromosome scale. Both species belong to the Anguivi grade, and are thus phylogenetically distant from *S. melongena*, whose direct progenitor and sister wild species are, respectively, *S. insanum* and *S. incanum*. This limitation questions the validity of many of the phylogenetic analyses made, such as inferring the ancestral state of SV (Fig. 3d) or the ancestral component analysis (Fig 1e). The analyses should be repeated using the direct eggplant progenitor *S. insanum* and/or the sister species *S. incanum*.

[R5C2]: We appreciate the reviewer's constructive suggestion. Initially, we did not use *S. insanum* or *S. incanum* as outgroups because high-quality genome assemblies for these species were not available to us at the time of our original analysis.

In the revised version, we have incorporated *S. insanum* as the outgroup (**Lines 221, 249**), thanks to the recent availability of a chromosome-scale genome assembly (N50 = 44.6 Mb) from the *Solanum* pangenome project (Benoit et al. 2025). With this addition, we have repeated the relevant phylogenetic and ancestral state analyses. Using *S. insanum* as the outgroup has significantly improved our results. For instance, the proportion of variant sites for which we could confidently infer the ancestral state increased from 28.1% to 74% (**Supplementary Table 6**). Although the primary findings have not changed substantively, the inclusion of *S. insanum* has substantially strengthened the resolution and robustness of our evolutionary inferences. We

sincerely thank the reviewer again for this valuable suggestion, which has greatly enhanced the quality of our study.

Reference:

Benoit, M., Jenike, K.M., Satterlee, J.W., Ramakrishnan, S., Gentile, I., Hendelman, A., Passalacqua, M.J., Suresh, H., Shohat, H., Robitaille, G.M. and Fitzgerald, B., 2025. Solanum pan-genetics reveals paralogues as contingencies in crop engineering. Nature, pp.1-11.

3. A third limitation is the approach used for pangenome graph construction. The main advantage of graph-based pangenomes is an unbiased all-vs-all comparison, thus avoiding the so-called reference bias (Secomandi, et al (2025). Nature Genetics, 1-14). Two main pipelines exist for this purpose: Minigraph-Cactus (Hickey, et al. (2024). Nature biotechnology, 42(4), 663-673) and PanGenome Graph Builder (Garrison, et al. (2024). Nature Methods, 1-5.). I therefore wonder why authors decided to first detect SVs with a reference-based approach and then build the graph. The resulting reference bias, summed to the geographical and genetic bias mentioned above, is likely to limit the value of this pangenome. Reference bias can be removed using an all-vs-all comparison, eg based on PGGB.

We thank the reviewer for raising this important point. We agree that one of the key advantages of graph-based pangenomes lies in their ability to reduce reference bias through unbiased all-vs-all genome comparisons. We also greatly appreciate the continuing efforts of the pangenome research community to address these challenges. As the reviewer noted, several recent pipelines—such as Minigraph-Cactus and PGGB—have been developed to construct more comprehensive and reference-free pangenomes (Secomandi et al., 2025; Hickey et al., 2024; Garrison et al., 2024). However, we note that these tools have primarily been developed and benchmarked on animal genomes, and their application to large, repetitive, and structurally complex plant genomes remains limited. Such genomes pose significant technical challenges for genome assembly, alignment, and SV detection, often requiring high computational resources and more sensitive alignment algorithms (Song et al., 2024).

*To address the reviewer's concern, we applied both Minigraph-Cactus and PGGB to our eggplant genome assemblies. However, both pipelines identified substantially fewer SVs compared to SVGAP. For instance, PGGB detected only about half of SVs across the 17 assemblies, compared to SVGAP. We believe this discrepancy reflects current limitations of all-vs-all methods when applied to large, repetitive plant genomes, particularly in the alignment step. In our independent benchmarking study (The paper is accepted in **Molecular Biology and Evolution** and will soon appear; Hu et al., 2025), we also observed that PGGB and Minigraph-Cactus showed reduced sensitivity on plant genomes such as rice, relative to SVGAP (please see below Figure). Given these observations, we chose to adopt the current approach in the study: first detecting SVs using a reference-based method (SVGAP), followed by graph*

construction using the identified variants. But, we are happy to report the PGGB result if necessary.

We expect that future improvements in graph-based pangenome construction—especially approaches optimized for large and repetitive plant genomes—will greatly enhance the accuracy and completeness of SV discovery. We are actively following developments in this area and look forward to integrating more scalable and robust methods like PGGB and Minigraph-Cactus as they continue to evolve.

Reference:

Secomandi, S., Gallo, G.R., Rossi, R., Rodríguez Fernandes, C., Jarvis, E.D., Bonisoli-Alquati, A., Gianfranceschi, L. and Formenti, G., 2025. Pangenome graphs and their applications in biodiversity genomics. *Nature Genetics*, 57(1), pp.13-26.

Hickey, G., Monlong, J., Ebler, J., Novak, A.M., Eizenga, J.M., Gao, Y., Marschall, T., Li, H. and Paten, B., 2024. Pangenome graph construction from genome alignments with Minigraph-Cactus. *Nature biotechnology*, 42(4), pp.663-673.

Garrison, E., Guarracino, A., Heumos, S., Villani, F., Bao, Z., Tattini, L., Hagmann, J., Vorbrugg, S., Marco-Sola, S., Kubica, C. and Ashbrook, D.G., 2024. Building pangenome graphs. *Nature Methods*, 21(11), pp.2008-2012.

Song, B., Buckler, E.S. and Stitzer, M.C., 2024. New whole-genome alignment tools are needed for tapping into plant diversity. *Trends in Plant Science*, 29(3):355-69.

Hu, M., Wan, P., Chen, C., Tang, S., Chen, J., et al. (2025). Benchmarking, detection, and genotyping of structural variants in a population of whole-genome assemblies using the SVGAP pipeline. *bioRxiv*, 2025-02.

Other suggestions for improvement:

4. The fact that gene families in the pangenome “nearly reached a plateau after adding 17 genomes” (line 171) is largely expected. What about using all genes identified in the 17 genomes?

*[R5C4]: Thank you for raising this point. To address the reviewer’s concern, we conducted an additional analysis that included all genes, including those not assigned to orthogroups. The results showed that the number of pan genes continued to increase with the addition of each genome, suggesting that private genes substantially contribute to the observed non-plateauing pattern. We have reported this result in **Supplementary Fig. 8** and highlighted it in the Discussion section (**Lines 584-585**).*

5. In the admixture analysis, authors tested 2-5 Ks. In the CV plot however, k from 2 to 8 are analyzed. Why did they select 3 as best K, since the CV plot shows a plateau at 8 (Fig. 1A)? This point was also raised by reviewer 2 on different grounds (R2C10) and needs to be better addressed.

*[R5C5]: We appreciate the reviewer for raising this point. In our admixture analysis, the cross-validation (CV) error continued to decrease without reaching a clear plateau even at K = 10 (**Supplementary Fig. 1**), including after we added 25 additional samples from previously published studies (please see our response to **R5C1**). However, the rate of decline began to noticeably slow at K = 4, suggesting that these values may represent reasonable estimates of population structure, as also recommended by Reviewer 2. Furthermore, the results from PCA,*

phylogenetic analysis, and the geographical origins of the samples consistently support the presence of four major groups (especially Fig. 1f). Therefore, we selected K = 4 to present the admixture results shown in Fig. 1c, d, f, and g. We have updated the figure, its legend, and the corresponding text in the revised manuscript (Line 124-127).

6. Previous work by other groups has identified several candidate genes for anthocyanin biosynthesis in eggplant, as well as, in some cases, the causal mutations (see eg Florio, et al (2021). International Journal of Molecular Sciences, 22(17), 9174; Mangino, et al (2022). Frontiers in Plant Science, 13, 847789; You, et al (2023). Horticulture Research, 10(2), uhac268; Xiao, et al (2024). International Journal of Molecular Sciences, 25(10), 5241). Of these, the only paper that is cited (lines 305 and 616) is the You et al paper, by some of the authors of the present paper. In particular, the 6-bp deletion in the Chr10 Myb gene that is causing the non-purple phenotype (Fig 4f-g in the present paper) has been described by Mangino et al (2022) (fig 5j in that paper). Appropriate credit should be given to previous work in the field by other groups.

We apologize for the oversight and sincerely thank the reviewers for bringing these valuable references to our attention. We have carefully read and appreciated their contributions to the eggplant genomics community, and have now cited them in the appropriate sections of the revised manuscript.

Reference:

Florio FE, Gattolin S, Toppino L, Bassolino L, Fibiani M, Lo Scalzo R, Rotino GL. A Sm1AAT Acyltransferase Variant Causes a Major Difference in Eggplant (Solanum melongena L.) Peel Anthocyanin Composition. Int J Mol Sci. 2021 Aug 25;22(17):9174.

Mangino G, Arrones A, Plazas M, Pook T, Prohens J, Gramazio P, Vilanova S. Newly Developed MAGIC Population Allows Identification of Strong Associations and Candidate Genes for Anthocyanin Pigmentation in Eggplant. Front Plant Sci. 2022 Mar 7;13:847789.

You Q, Li H, Wu J, Li T, Wang Y, Sun G, Li Z, Sun B. Mapping and validation of the epistatic D and P genes controlling anthocyanin biosynthesis in the peel of eggplant (Solanum melongena L.) fruit. Hortic Res. 2022 Dec 2;10(2):uhac268.

Xiao K, Tan F, Zhang A, Zhou Y, Zhu W, Bao C, Zha D, Wu X. Fine Mapping of Candidate Gene Controlling Anthocyanin Biosynthesis for Purple Peel in Solanum melongena L. Int J Mol Sci. 2024 May 11;25(10):5241.

7. In the text, the authors conclude that the 12.4 Mb inversion on Chr 10 is not associated with the non-purple color, but it reduces genetic diversity in the surrounding chromosomal region. This conclusion is correct, but not thoroughly analyzed and inappropriately presented in the abstract: a) both P_i and Tajima's D are reduced in the whole pericentromeric heterochromatin (Fig. 5b, d; Chr10 should be metacentric, Fang, H., et al (2025). International Journal of

Biological Macromolecules, 284, 138094). How does an inversion in a chromosome arm affect these two parameters in pericentromeric heterochromatin, even on the other side of the centromere? Is it due to “gamete culling” (i.e. gametes that have a recombination event in the inverted region contain unviable chromosomes, thus reducing recombination over the whole chromosome)? If so, what is the predicted fitness on the chromosome carrying the inversion in a free-pollinating population? This point should be discussed. b) The present formulation of the abstract “we identified a 12.4 Mb 27 inversion at a high frequency (50.7%) within the population, which is associated with fruit color, likely due to genetic hitchhiking” is misleading, in that it suggests some sort of causal relationship between the inversion and fruit color. I suggest changing into “...associated with a previously identified mutation for fruit color..”

[R5C7]: We are grateful to the reviewer for the thoughtful comments and valuable suggestions.

a) The reduction in nucleotide diversity (π) and Tajima's D is not confined to the inverted region or the pericentromeric regions of chromosome 10, but extends along nearly the entire chromosome (Fig. 5b and 5d). We believe this pattern is more likely the result of a founder effect or bottleneck event, rather than a direct consequence of the inversion itself. The population distribution of this inversion reveals that it occurs at a higher frequency in Chinese accessions, suggesting that it may have undergone artificial selection during domestication and breeding in China.

The reviewer's suggestion regarding “gamete culling” is highly insightful. If we understand correctly, this refers to the phenomenon where, in heterozygous individuals, recombination within large paracentric inversions can generate acentric or dicentric chromatids, leading to inviable gametes. This process can suppress recombination and reduce genetic diversity, not only within the inversion but potentially across the entire chromosome if selection broadly disfavors recombinant gametes. However, since both the ancestral haplotype and the inversion-carrying haplotype are found at high frequencies in the population, we suspect there may be limited differences in adaptive fitness between them. Nevertheless, it would be very interesting to experimentally test this hypothesis by performing crosses between individuals with and without the inversion in future studies.

b) We appreciate the reviewer's clarification. We have revised the sentence as suggested to: “...associated with a previously identified mutation for fruit color, likely due to genetic hitchhiking.” (Line 29)

Once again, we thank the reviewer for these constructive and thought-provoking suggestions.

Minor points:

8. I don't understand fig 2g: It supposedly depicts Ka/Ks values for orthologous pairs in the core, softcore, dispensable, and private genes. But if a gene is private, how can it be part of an

orthologous pair? Could the authors explain better how they did the analyses in this figure and suppl figs 7 and 8e?

[R5C8]: We apologize for the confusion. In our previous analysis, we used two different tools, which led to inconsistencies. First, we classified genes into core, softcore, dispensable, and private categories based on OrthoFinder results. Second, we employed JCVI (v1.3.5) to identify orthologous gene pairs from all pairwise comparisons across the 17 genomes (as illustrated in Fig. 2g). The syntenic and orthologous gene pairs identified by JCVI were used to calculate Ka/Ks values. These gene pairs were then categorized into core, softcore, dispensable, and private groups based on the OrthoFinder classification.

However, we noticed that in a small number of cases (n = 666, ~0.04% of all gene pairs), genes previously classified as private by OrthoFinder also had syntenic orthologs in JCVI results. This discrepancy caused confusion due to the inconsistency between the outputs of the two software tools.

To resolve this issue and ensure consistency, we revised our Ka/Ks analysis by excluding gene pairs involving private genes. Only gene pairs supported by both JCVI and OrthoFinder (i.e., assigned to the same gene family and found in syntenic regions) were retained for comparison. Fig. 2g, Supplementary Fig. 6, and related method descriptions (Line 852-853) have been updated accordingly.

9. Ref 14, which should refer to a chromosome level eggplant assembly, instead refers to phylogenomic discovery of deleterious mutations in potato. Ref 69, which should refer to fruit pigmentation, instead is a review on bacterial wilt resistance in Solanaceae. Please correct.

Thank you for this clarification! We have confirmed that Reference 14 refers to the phylogenomic discovery of deleterious mutations in potato, which also included over 30 new genome assemblies of Solanum species, including three eggplant accessions used in our analysis. Therefore, this reference should be accurately cited. However, Reference 69 was indeed incorrectly cited, and we have now corrected it accordingly in the revised manuscript.

REVIEWER COMMENTS

Reviewer #3 (Remarks to the Author):

My concerns have been largely addressed in this revision. I have only one remaining comment and one minor suggestion.

We sincerely thank the reviewer for the time and constructive feedback. We are glad that the previous concerns have been largely addressed and have carefully considered the remaining comment and minor suggestion as detailed below.

(1) The authors have effectively highlighted the advantage of graph-based pangenomes in accurately identifying structural variants (SVs).

Thank you!

However, the proposed association between gene dosage effects and bacterial wilt (BW) resistance remains insufficiently supported.

Response [R3C1]:

To address the reviewer's concern, we employed an additional approach to genotype the copy number variations (CNVs) of SmEPS1 and SmRoq1 across approximately 200 accessions to verify their association with bacterial wilt (BW) resistance. Briefly, resequencing data were mapped to a reference sequence, which harbors a single copy of both SmEPS1 (S126_evm.model.Chr05.2786) and SmRoq1 (S126_evm.model.Chr05.2787), and a single-copy gene (S126_evm.model.Chr05.2898) determined by OrthoFinder analysis of 17 chromosome-scale genomes serving as a control. Copy numbers were estimated using Bedtools (v2.31.1; coverage -mean) (Quinlan and Hall, 2010) based on the mean read coverage of two conserved regions within SmEPS1 and SmRoq1, which were normalized to that of the single-copy gene.

*Accessions were classified as E+ or E- if the normalized mean coverage of SmEPS1 was greater or less than 2, respectively, and as R+ or R- if the normalized mean coverage of SmRoq1 was greater or less than 15, respectively. Using these genotypes, we performed a correlation analysis between CNVs and BW resistance and observed an obvious correlation (**Supplementary Fig. 19**). This correlation pattern was highly consistent with the previous association analysis between the two structural variants (SV_e and SV_r) and BW resistance, suggesting that SV_e and SV_r largely correspond to the CNV genotypes of SmEPS1 and SmRoq1, likely due to their strong genetic linkage.*

However, we acknowledge that the current evidence is insufficient to conclude that the CNVs themselves are the direct causal variants underlying BW resistance. Given the structural complexity of the genomic regions harboring these genes, alternative explanations are possible. It remains unclear which specific gene copies are functionally associated with BW resistance, and the observed correlation between higher copy number and resistance may

instead reflect linkage with the true causal copy or with other nearby genomic variants. We have incorporated this interpretation into the revised Discussion (Lines 676-681).

This association is further complicated by the finding that silencing a single gene, *SmEPS1*, results in a substantial reduction in BW resistance (Fig. 6g,h), even though the accession background of the wild-type control is not specified.

*Due to the presence of multiple copies of both *SmEPS1* and *SmRoq1* in the eggplant genome, for each gene type, the VIGS experiment was designed to target a conserved fragment shared across all copies. As a result, the silencing effect likely acts on most, if not all, expressed copies.*

Regarding the reviewer's concern about the accession background of the wild-type control used in the VIGS experiments (Fig. 6g,h), we have now clarified this information in the Results section (Lines 497-501) and in the figure legend of Fig. 6 (Lines 533-536). The revised legend reads as follows:

*“g, Relative expressions of *SmCYP82D47* in control (TRV::00) and *SmCYP82D47*-silenced (TRV::*SmCYP82D47*) plants of ‘S065’ (with TGG_E-_R- genotype); *SmEPS1* in control (TRV::00) and *SmEPS1*-silenced (TRV::*SmEPS1*) plants of ‘S092’ (with TGA_E+_R- genotype); and *SmRoq1* in control (TRV::00) and *SmRoq1*-silenced (TRV::*SmRoq1*) plants of ‘S050’ (with TGA_E-_R+ genotype).”*

Moreover, the connection between the representative SVs (‘SVe’ and ‘SVr’) and the observed copy number variation (CNV) is not yet strongly substantiated, particularly given the limited sampling of only 11 assembled genomes.

Please see our response above, where we extended the genotyping to all ~200 accessions. The correlations between SVs and BW, and between CNVs and BW, were highly consistent, suggesting that they are strongly genetically linked.

The contribution of CNVs may be better interpreted as providing a genetic basis for potential mutational events rather than directly contributing to phenotypic resistance.

Thank you for this suggestion. As noted in our response above, it remains unclear which specific gene copies are functionally associated with BW resistance, and the observed correlation between higher copy number and resistance may instead reflect linkage with the true causal copy or other nearby genomic variants. We have incorporated this interpretation into the revised Discussion (Lines 676-684).

Supplementary Fig. 19. Comparison of bacterial wilt incidence rates for 182 accessions with different genotype combinations defined by two approaches for *SmCYP82D47*, *SmEPS1* and *SmRoq1*. **a**, Comparison based on *SmCYP82D47*, *SVe*, and *SVr*; **b**, Comparison based on *SmCYP82D47* and normalized mean coverages of *SmEPS1* and *SmRoq1*. Normalized average incidence rates are plotted. In boxplots, interquartile shows as lower and upper edges of boxes, respectively, and central lines stand for the median. Different letters (a, b, c, etc.) above the bars indicate significantly different values ($p < 0.05$) calculated using the one-way ANOVA.

(2) Minor suggestion: The criteria used to define and determine **derived alleles** should be clearly described in the Methods section.

Response[R3C2]:

We apologize for the lack of this information. We have updated the Methods section, “Structural variant identification, analysis, and graph-based pangenome construction,” by adding the following text (**Lines 839-849**):

“To calculate minor allele frequency (MAF), we used VCFtools with the `--freq` command. To estimate derived allele frequency (DAF), we used the genotype of *S. insanum* in the VCF file as the ancestral state. Specifically, when the genotype of *S. insanum* was “0/0,” the alternative genotype “1/1” was treated as the derived allele, and vice versa. For the genome assembly-based SV callset generated using SVGAP, the genome of *S. insanum* was obtained from a recent study; approximately 74.3% (56,486 out of 76,002) of presence/absence variants (PAVs) were successfully assigned an ancestral state. We also tested *Solanum violaceum* (S098) as an alternative outgroup reference, but only 28.1% (21,351/76,002) of sites could be assigned ancestral states. Nevertheless, the derived allele frequencies estimated using both references were highly consistent (**Fig. 3d** and **Supplementary Fig. 12c**). For the short-read-based SV callset, the genotype of accession S225 (*S. insanum*) was used as the ancestral reference.”

Reviewer #5 (Remarks to the Author):

(1) The authors have only partially addressed my main concern, which was the scarce representation of accessions from the Indian domestication center. They added few resequenced accessions from Europe and Asia, but the Indian domestication center remains crucially underrepresented, together with other important regions, such as the Middle East, Africa, Korea and Japan (Fig 1a). Therefore, the definition “sub-pangenome” is the most appropriate. Its limitations should be more extensively discussed, indicating explicitly that it lacks a sufficient number of accessions from the Indian domestication center and expansions from that center, and therefore many genomic events that occurred in these regions are likely missing from this resource, limiting its usefulness.

Response[R5C1]:

*Thank you for the comments and suggestions. We agree with the reviewers’ opinion that the current genetic resources only partially represent the genetic diversity of cultivated eggplants, as shown in **Supplementary Fig. 4**. Therefore, we used the term “**sub-pangenome**” to describe our resource in the title and also see the Discussion section and the description in **Lines 574-577** as follows:*

“[R5C1] Considering that our samples are primarily from China and Southeast Asia, future studies incorporating more samples from diverse origins, particularly from India, the Middle East, Africa, Korea and Japan, will be essential to expand our comprehensive understanding of eggplant domestication, migration routes, and genomic diversity.”

We apologize for not providing a more comprehensive discussion of this limitation in the previous version. In the revised manuscript, we have added additional discussion on this topic to better acknowledge the underrepresentation of genetic diversity in our dataset. We hope that the reviewers find this revision satisfactory.

(2) As noted previously, this limitation is likely to be exacerbated by the reference bias introduced by graph construction using SVGAP. The authors, in their reply, show that the reference-independent PGGB software is performing as well as SVGAP in detecting bias. A logical next step is to perform GWAS on the PGGB reference-independent graph. This analysis is missing in the present revision and should be introduced.

Response [R5C2]:

*Thank you for this valuable suggestion. Our previous benchmarking analysis showed that **PGGB** performed comparably to **SVGAP** on simulated SV datasets. Therefore, we applied PGGB to our set of 17 eggplant genome assemblies. However, it reported a substantially lower number of SVs compared with SVGAP (31,793 vs. 76,481), suggesting that its performance on large and complex plant genomes still has room for improvement. As suggested, we also used the PGGB-based SVs for the GWAS analysis. The results showed that for fruit color, only one SV signal was detected on **Chr10**. For bacterial wilt resistance, no signals were detected at the two stable peaks on **Chr04** and **Chr05** that were previously identified by SNP-GWAS, InDel-GWAS, and SVGAP-SV-GWAS. Considering the relatively small number of SVs obtained from the PGGB-derived graph, we have therefore retained the*

SVGAP-based results in the main manuscript and reported the PGGB-based results in **Supplementary Fig. 22 or Table S13 (Lines 508-509)**. We hope that the reviewer finds our revision satisfactory.

Supplementary Figure 22. GWAS analysis for fruit color and bacterial wilt resistance in eggplant using a PGGB-derived SV dataset. a, SV-GWAS for fruit color. b, SV-GWAS for bacterial wilt resistance for four batches, including 22-1, 22-2, 22-3, and 23-1 batches, respectively.

(3) My third comment, i.e. the lack of a proper outgroup to anchor the phylogenetic data, has been addressed by using a single, publicly available *S. insanum* chromosome level assembly that, as the authors note, “substantially strengthened” the resolution and robustness of their evolutionary inferences. It must be noted that *S. insanum* is heterogeneous, with some accessions being feral or admixed forms with *S. melongena*, and therefore the resolution and robustness of the evolutionary inferences would be further strengthened by including multiple assemblies of *S. insanum* and of the closely related *S. incanum*. Given the fact that

generating additional chromosome-level assemblies is time-consuming and expensive, this is not an absolute requirement.

Response [R5C3]:

*We sincerely appreciate the reviewer's insightful comment and understanding. Given the potential limitation of using only one *S. insanum* genome as the outgroup, we have also reported our previous results using *Solanum violaceum* (S098) as an alternative outgroup. Although S098 resulted in a smaller number of ancestral sites compared with *S. insanum*, the derived allele frequency spectra obtained from both outgroups were highly consistent (**Fig. 3d and Supplementary Fig. 12d**). We acknowledge that using a single *S. insanum* genome assembly as the outgroup may introduce some uncertainty in identifying ancestral states; however, the overall trends remain robust. Incorporating additional *S. insanum* or *S. incanum* genome assemblies in future analyses will further improve the accuracy and completeness of ancestral state inference. We thank the reviewers again for their thoughtful feedback and understanding.*

Additional comments:

Ralstonia resistance:

(4) The relative contribution of CYP82, EPS1 and Roq1 genes in Ralstonia resistance remains elusive and should be better studied. In order to better understand it, a supplementary figure should be provided in which pie charts are shown for two of the gene classes in genotypes separately for the two classes of the third gene, together with the % average susceptibility of the two classes. Eg: pie charts for the four classes (E+R+, E-R+, E+R-, and E-R-) in TGG and TGA genotypes. And so on, for the E+/E- and R+/R- genotypes. % susceptibility and number of accessions should be indicated for the genotypes in each class.

Response[R5C4]:

*We thank the reviewer for this insightful suggestion. As recommended, we have included an additional supplementary figure (see **Supplementary Fig. 24**) to illustrate the relative contribution of SmCYP82D47, SmEPS1, and SmRoq1 to Ralstonia resistance.*

*In this new figure, we present pie charts showing the distribution of genotypes for two gene classes separately for the two allelic classes of the third gene (e.g., E+R+, E-R+, E+R-, and E-R- in TGG and TGA genotypes) (**Supplementary Fig. 24a**). For each genotype class, we also indicate both the number of accessions and the average susceptibility (%) to Ralstonia.*

The above results reveal that the presence of either a higher copy number of SmEPS1 or SmRoq1 is sufficient to confer resistance, provided that SmCYP82D47 is in the wild-type state. However, when SmCYP82D47 carries a mutation, elevated copy numbers of both SmEPS1 and SmRoq1 are required to achieve resistance. Overall, SmCYP82D47 appears to make a relatively greater contribution to resistance among the three gene types, whereas

SmEPS1 and *SmRoq1* contribute at comparable levels. Notably, plants exhibit the highest susceptibility when all three genes are off (carrying a mutation or with lower copy numbers).

These data help visualize the interaction among the three resistance-related genes and their contribution patterns to bacterial wilt resistance. The corresponding description has been added to the Discussion section (**Lines 686–694**).

Supplementary Figure 24. Distribution of *SmCYP82D47*, *SmEPS1* (*E*), and *SmRoq1* (*R*) genotypes and their association with bacterial wilt resistance in eggplant. **a**, Pie charts showing the proportions of *E*+*R*+, *E*-*R*+, *E*+*R*-, and *E*-*R*- genotypes in TGG and TGA genetic backgrounds. **b**, Pie charts showing the proportions of TGA_{*R*+}, TGA_{*R*-}, TGG_{*R*+}, and TGG_{*R*-} genotypes in *E*- and *E*+ genetic backgrounds. **c**, Pie charts showing the proportions of TGA_{*E*+}, TGA_{*E*-}, TGG_{*E*+}, and TGG_{*E*-} genotypes in *R*+ and *R*- genetic backgrounds. The number of accessions and the average susceptibility (%) to *Ralstonia solanacearum* are indicated for each genotype class in pie charts.

(5) Supplementary figure 19 is not very informative in clarifying the role of the different genes in *ralstonia* resistance. First, it is unclear how genes in the different clusters correlate with those in fig. 6e. These should be marked with names or, better, with colors corresponding to the different clusters in supplementary fig 19. Importantly, supplementary figures (or if it's too complicated, tables) should be provided, giving the expression of individual genes in populations with different classes of susceptibility to *ralstonia* (eg, low, intermediate and high) and for all the CYP82, EPS and Roq1 genes analyzed. Alternatively, the CYP82, EPS and Roq1 gene composition and expression should be added to Table S10.

Response[R5C5]:

We sincerely thank the reviewer for these constructive suggestions. Following the reviewer's advice, we have revised Fig. 6e by adding color labels that correspond to the SmEPS1 gene clusters shown in Supplementary Fig. 19 (updated as now Supplementary Fig. 20), to make the correspondence between these two figures clearer. In addition, we have supplemented **Table S10** with detailed information on the composition and expressions of all SmCYP82, SmEPS, and SmRoq1 genes across 111 accessions with different susceptibility levels to *Ralstonia*.

With the availability of fruit transcriptomes from over 100 samples, we compared gene expression levels among samples carrying different genotype combinations of the three gene classes and exhibiting varying resistance levels. Since the mutation in SmCYP82 introduces a premature stop codon and is not associated with expression changes, no clear expression patterns were observed for this gene. In contrast, by fixing the genotypes of SmCYP82 and SmRoq1, SmEPS1 showed a general trend, where E+ samples exhibited higher expression levels and lower disease incidence rates, while E- samples displayed lower expression levels and higher incidence rates (**Supplementary Figure 21**).

No obvious trends were detected for SmRoq1 expression, likely reflecting the more complex nature of this gene, which could be caused by several reasons. First, SmRoq1 exists in substantially more copies than SmEPS1, with highly variable copy numbers among samples. Second, individual copies may exhibit differential tissue-specific or pathogen-induced expression patterns. Therefore, for SmRoq1, it remains to be clarified in future studies whether variation in bacterial wilt resistance is primarily driven by overall copy number variation, by specific members, or by other functional variants. The corresponding description has been added to the Discussion section (**Lines 495-497, 676-684**).

Supplementary Figure 21. Comparison of SmEPS1 gene expressions across samples with different genotype combinations of the three gene classes and with different resistance levels. a, The disease incidence rate for different genotype combinations of the three gene classes. b, The TPM values of SmEPS1 using S126 as a reference genome for different genotype combinations of the three gene classes.

(6) VIGS: Although the authors claim that they performed RT-PCR to evaluate the silencing of the different EPS1 and CYP82D47 homologs, I wasn't able to find anywhere the data. These data should be shown, together with a table with the % homology of the different homologs to the silencing fragments used. It would be also appropriate to show VIGS data also for the Roq1 homologs.

Response [R5C6]:

We sincerely thank the reviewer for this valuable comment. SmCYP82D47 is a single-copy gene (as defined by OrthoFinder analysis) across eggplant genomes. A fragment located in its conserved region was selected for the VIGS experiment and confirmed to be unique in the eggplant genome based on BLASTn results. Due to the presence of multiple copies of both SmEPS1 and SmRoq1 in the eggplant genome, for each gene type, the VIGS experiment was designed to target a conserved fragment shared across all copies, rather than targeting each homolog individually. In response to the reviewer's suggestion, we have provided a new figure below showing the sequence alignments of the silencing fragments with their corresponding homologs. The red region indicates the conserved region among homologs.

Following the reviewer's suggestion, we have additionally performed VIGS for SmRoq1 and updated the Results and Methods sections (Lines 497–504, 533–540, and 913–916), as well as revised Fig. 6g and 6h accordingly.

Figure. Comparison of VIGS fragments with homologs of SmEPS1 and SmRoq1. a, Sequence alignments with SmEPS1 homologs, and the conserved region was highlighted in red. **b,** Sequence alignments with SmRoq1 homologs, and the conserved region was highlighted in red.

Fig. 6g, Relative expressions of SmCYP82D47 in control (TRV::00) and SmCYP82D47-silenced (TRV::SmCYP82D47) plants ‘S065’ (with TGG_E-R- genotype), SmEPS1 in control (TRV::00) and SmEPS1-silenced (TRV::SmEPS1) plants ‘S092’ (with TGA_E+R- genotype), and SmRoq1 in control (TRV::00) and SmRoq1-silenced (TRV::SmRoq1) plants ‘S050’ (with TGA_E-R+ genotype). Data are presented as mean values \pm SD of five independent biological replicates. Significance levels are computed from Student’s t-test. **h**, The phenotype of control, SmCYP82D47-silenced, SmEPS1-silenced, and SmRoq1-silenced plants at 10 days after *R. solanacearum* infection. Each treatment had five biological replicates.

(7) Minor: The pie chart data on CYP, EPS and Roq1-like genotypes in various classes of ralstonia susceptibility are shown only for four susceptibility classes (0 to 40-60). They should be shown also for the remaining two classes (60-80 and 80-100).

Response[R5C7]:

We thank the reviewer for this helpful suggestion. As shown in Fig. 6b, the pie charts were generated based on the average incidence rates calculated from four batches of our population. The two higher susceptibility classes (60–80 and 80–100) are not displayed because very few or no accessions fell into these categories, leading to the absence of the corresponding green bars in Fig. 6b. Therefore, these two classes were not shown in the pie charts.

(8) Once these data are provided, a discussion of the synergic-epistatic relations of the three classes of genes in determining Ralstonia resistance should be provided.

Response [R5C8]:

We appreciate the reviewer’s valuable suggestion to discuss the potential synergistic and epistatic relationships among SmCYP82D47, SmEPS1, and SmRoq1 genes in determining Ralstonia resistance. Our results indicate that SmCYP82D47 plays a central role in bacterial wilt resistance. When SmCYP82D47 carries a premature stop codon, accessions with high copy numbers of SmEPS1 and SmRoq1 still fail to exhibit strong resistance (**Supplementary Fig. 24**). In contrast, when SmCYP82D47 is functional, higher copy numbers of either

SmEPS1, SmRoq1, or both confer strong resistance, whereas accessions with lower copy numbers for both remain susceptible. These observations suggest that SmCYP82D47 exerts the most pronounced effect among the three loci, while SmEPS1 and SmRoq1 contribute comparably to resistance. The corresponding discussion has been added to the Discussion section (Lines 686–694).

Other:

(9) Supplementary Table 2 provides only the statistics for the 11 de novo assembled genomes. Statistics (except HiFi and HiC, if not available) should be provided for all 17 chromosome level genomes used in the analyses, including the newly added *S. insanum*.

Response [R5C9]:

*We thank the reviewer for this suggestion. We have now updated **Table S2** to include statistics for all 17 chromosome-level genomes used in our analyses including the newly added *S. insanum*.*

(10) Data availability: A search of the Chinese National Genebank with the Project or individual genome accession numbers found no results. Raw sequencing data, genome assemblies, annotations, must be released on a public database and the graph on a public pangenome browser such as PpanG (<https://doi.org/10.1186/s12864-024-10302-5>) upon paper acceptance.

Response [R5C10]:

*We thank the reviewer for this important comment. All raw sequencing data, genome assemblies, and genome annotations have been pre-deposited at the China National Genebank (CNGB) under the project accession number **CNP0006177**, which will be immediately released upon paper acceptance. Reviewers can also access and download our data via FTP using the following credentials:*

host: control.cngb.org

FTP account: ngb_03856

FTP password: loyM#SAW0w

For the pangenome graph deposition, we have carefully reviewed the PpanG browser (<https://cgm.sjtu.edu.cn/PPanG/>) and found that it currently lacks a data submission interface, and its visualization page is temporarily unavailable. Therefore, we sincerely apologize that we are unable to upload our graph data to this platform. As an alternative, we have deposited the SV datasets on Zenodo (<https://doi.org/10.5281/zenodo.15877227>), which allows the graph files to be readily reconstructed using vgtools.

REVIEWER COMMENTS

Reviewer #3 (Remarks to the Author):

All my previous concerns have been fully addressed.

We thank the reviewer for the positive evaluation of our previous revision.

Reviewer #5 (Remarks to the Author):

Overall Assessment

The authors have made a substantial effort to address my comments, and the manuscript is improved as a result. The clarification of the 12.4 Mb inversion on chromosome 10 as a case of genetic hitchhiking rather than a causal variant is a valuable scientific conclusion. The new analysis of the synergic-epistatic interactions of the three bacterial wilt (BW) resistance loci (*SmCYP82D47*, *SmEPS1*, *SmRoq1*) is a significant addition that clarifies a complex genetic architecture. However, the revised manuscript is unfortunately still beset by several critical issues:

We thank the reviewer for the positive evaluation and encouragement regarding our previous revision. We also greatly appreciate the valuable new points raised and detailed suggestions to improve our manuscript, which have helped us to further improve the manuscript.

(1) The authors' response to the request for a reference-independent pangenome-GWAS (using PGGB) is contradictory.

We apologize that our previous revision did not fully address the reviewer's concerns. Here, we provide a detailed PGGB-based analysis; please refer to our response to R5C2 below.

(2) In the light of the new data added, the VIGS results are difficult to interpret

Please kindly refer to our response to R5C6 below.

(3) The manuscript's primary narrative for fruit color focuses on the 12.4 Mb inversion on Chr10 while dismissing other loci as "less significant". This claim is wrong; the authors' own supplementary table shows SVs on Chr05 and Chr06 that are orders of magnitude more statistically significant.

*Response [R5C12]:
Please kindly refer to our response to R5C12 below.*

(4) The methodological novelty of this paper has been superseded by the recent publications of Gaccione et al. (2025) and Yu et al (2025).

[R5C11]:

We are pleased to see these excellent studies on the eggplant pangenome and thank the authors for their contributions to the field. These works have been highly informative and have helped us refine the current study. We also emphasize that our work provides complementary contributions by integrating pangenome resources with trait-focused analyses, particularly for bacterial wilt (BW) resistance and peel color. This integration improves our understanding of how pangenome variation underlies key agronomic traits and its implications for breeding. We are committed to working in concert with these efforts to advance eggplant breeding and to disseminate genomic and population resources to the broader eggplant community. To acknowledge these important contributions, we have cited these studies in the revised manuscript (lines: 72-79, 86-102).

Below the point-by-point evaluation of the authors' responses:

Comment R5C1. The authors adopted the term "sub-pangenome" in the title and abstract. They have added a paragraph to the Discussion acknowledging the geographical bias and calling for future studies to include samples from India, the Middle East, Africa, etc. This is a transparent and appropriate response.

We thank the reviewer for the positive evaluation of our previous revision.

Comment R5C2. This is still a significant shortcoming. The authors claim to have performed the requested PGGB-GWAS, then they dismiss the method, stating it "reported a substantially lower number of SVs" and that for BW resistance, "no signals were detected at the two stable peaks on Chr04 and Chr05." They use this "failure" to justify retaining their original, biased SVGAP results. This justification does not stand upon closer scrutiny: For BWR, Supplementary Table 13 explicitly lists 7 "PGGB derived SV", albeit on different chromosomes and with different p-values than with SVGAP (Supplementary Table 11). Fig. 6C and Suppl Fig. 22 are not comparable, since one refers to SNP-GWAS and the other to SV-GWAS and, for fruit color, it is not clear if the Manhattan plot in Fig. 4c refers to SNPs, SVs or both, so the reader is left wondering about the comparison between SVGAP-GWAS and PGGB-GWAS. Since PGGB is a robust, reference free method, in order to substantiate their claims the authors must report the complete sets of associations (SNPs, short InDels and SVs) obtained for the two traits with the two methods. Their current response is evasive.

Response [R5C2]:

We thank the reviewer for this suggestion. In the revised manuscript, we report two sub pangenome graphs. The first is a reference based graph constructed with the vg pipeline by integrating SVs identified by SVGAP from the assembled genomes with variants derived from short read resequencing data from 226 eggplant accessions. From this resource, we generated three variant sets. SNPs and InDels were called by mapping the short read resequencing data to the S076 reference genome using the standard GATK pipeline, whereas SVs were genotyped across samples by mapping short reads to the vg derived pangenome graph. The second is a reference free pangenome graph built using the PGGB pipeline from genome assemblies of 16

cultivated eggplant accessions, including 10 generated in this study and six previously published assemblies. We deconstructed this graph using S076 as the reference backbone, yielding 31,793 structural variants (SVs; >50 bp), 2,045,873 InDels, and 6,939,066 SNPs.

Prior to GWAS, we further filtered variants based on minor allele frequency and missingness and compared callsets between the two strategies. Overall, the GATK and SVGAP vg pipelines produced more variants than PGGB, as expected because the former also incorporates the 226 short read resequencing dataset. Summary statistics are provided in Supplementary Table 8. For SNPs, 97.61% of PGGB derived SNPs were also detected in the GATK callset or within a 2 kb flanking window. For InDels, 88.79% of PGGB derived calls overlapped the GATK callset or fell within a 5 kb flanking window. Together, these results indicate strong concordance between the two approaches (Supplementary Table 8).

Supplementary Table 8. Comparison of the variation total numbers used ref-based (GATK, SVGAP) and ref-free (PGGB) methods.

Variation types	Total variation no. (%)	Ref-based (GATK, SVGAP)	Ref-free (PGGB)
SNP	Total SNP no.	3,698,811	1,842,596
	overlapped SNP region (extended ± 2 Kb)	3,071,623	1,798,572
	overlapped SNP region (extended ± 2 Kb) / total SNP no. (%)	83.04%	97.61%
InDel	Total InDel no.	349,227	359,952
	overlapped InDel region (extended ± 5 Kb)	305,627	319,608
	overlapped InDel region (extended ± 5 Kb) / total InDel no. (%)	87.52%	88.79%
SV	Total SV no.	31,672	20,854
	overlapped SV region (extended ± 10 Kb)	22,184	15,173
	overlapped SV region (extended ± 10 Kb) / total SV no. (%)	70.04%	72.76%

We then performed GWAS for fruit color and bacterial wilt resistance using variants from both strategies and compared the associated loci identified by each. As expected, the reference based callsets generally yielded more significant associations than the reference free dataset (Supplementary Table 10). For fruit color, GATK SNPs identified 3,055 significant loci, compared with 373 loci detected using PGGB SNPs. Of the loci detected by PGGB SNPs, 97.59% were also detected in the GATK SNP results or within a 2 kb flanking window, and a similarly high overlap was observed for InDels (98.04%). GWAS based on either SV callset detected relatively few significant loci. Overall, these results indicate that the two approaches identify largely consistent association signals (Supplementary Fig. 14).

Supplementary Table 10. Comparison of the variation numbers associated with fruit color and bacterial wilt resistance used GATK, SVGAP and PGGB methods.

Variation types	GWAS associated variation no. (%)	Fruit color trait		Bacterial wilt resistance trait	
		ref-based (GATK, SVGAP)	PGGB	ref-based (GATK, SVGAP)	PGGB
SNP	associated SNP	3,055	373	3,262	1,039
	overlapped associated SNP region (extended ± 2 Kb)	1,244	364	1,675	755
	overlapped associated SNP region (extended ± 2 Kb) / associated SNP no. (%)	40.72%	97.59%	51.35%	72.67%
InDel	associated InDel	167	51	488	194
	overlapped associated InDel region (extended ± 5 Kb)	72	50	172	100
	overlapped associated InDel region (extended ± 5 Kb) / associated InDel no. (%)	43.11%	98.04%	35.25%	51.55%
SV	associated SV	8	1	22	10
	overlapped SV	0	0	0	0
	overlapped SV no. / associated SV no. (%)	0%	0%	0%	0%

For bacterial wilt resistance, overlap rates were lower than for fruit color but remained substantial, reaching 72.67% for SNPs and 51.55% for InDels, likely reflecting a more complex genetic architecture. Specifically, 72.67% of loci detected by PGGB SNP GWAS overlapped the GATK SNP results or fell within a 2 kb flanking window, and the corresponding overlap for InDels was 51.55% (Supplementary Table 10). Importantly, the key candidate genes for fruit color and bacterial wilt resistance were supported by both variant callsets, reinforcing the robustness of these candidates (Supplementary Table9, 13, and 15).

Supplementary Fig. 14 GWAS analysis for fruit color and bacterial wilt resistance in eggplant with reference based (GATK, SVGAP and Panpop) and non-reference based (PGGB) methods. **a, b, c**, Genome-wide association analyses based on SNPs, InDels, and SVs for fruit color using reference-based (GATK, SVGAP, and Panpop) and reference-free (PGGB) methods. **d, e, f**, Genome-wide association analyses based on SNPs, InDels, and SVs for bacterial wilt resistance using reference-based (GATK, SVGAP, and Panpop) and reference-free (PGGB) methods, respectively.

Comment R5C3. The authors had already addressed it in the previous revision.

We thank the reviewer for the positive evaluation of our previous revision.

Comment R5C4. The authors provided a new Supplementary Figure 24 and a new Discussion paragraph detailing the "synergic-epistatic interactions." This is a valuable addition that significantly strengthens the manuscript.

We thank the reviewer for the positive evaluation of our previous revision.

Comment R5C5. The authors have revised Fig. 6e by adding color labels that correspond to the SmEPS1 gene clusters shown in Supplementary Fig. 20, thus addressing my comment.

We thank the reviewer for the positive evaluation of our previous revision.

Comment R5C6. The authors have performed VIGS for SmRoq1 and updated the Results and Methods sections (Lines 497–504, 533–540, and 913–916), as well as revised Fig. 6g and 6h accordingly. The figure provided in the rebuttal letter but not in the supplementary materials suggests that, for each VIGS fragment, there are multiple transcripts with high enough homology to be silenced by the SmEPS1 and SmRoq1 VIGS fragments (see eg Fernandez-Pozo et al Molecular plant, 8(3), 486-488. Also, the reader has no clue on which of these transcripts are recognized by the primers used for the RT-PCR.

We thank the reviewer for raising this concern. To clarify the VIGS results and assess potential off-target effects, we have added Supplementary Fig. 24 and Supplementary Table S17, which summarize the sequence similarity between each VIGS target fragment (for SmEPS1 and SmRoq1) and all annotated genes. These analyses indicate that the SmEPS1 and SmRoq1 VIGS fragments are highly specific, showing high similarity only to the corresponding SmEPS1 (n = 3) or SmRoq1 homolog(s) (n = 10) and not to other genes (Supplementary Table S17). A similar specificity pattern was observed for the qRT-PCR primer sets, which also matched only homologs of their respective target genes.

Supplementary Table 17. Homology levels between the VIGS target fragment and its homologous genes.

Reference genome	genotypes used for VIGS	Homologous genes	Refer	homology percentage between VIGS target fragment and genes	qRT-PCR primer detected
S092	TGA_E+R-	evm.model.Chr05.2742	EPS1 homolog	96.93%	detected
S092	TGA_E+R-	evm.model.Chr05.2746	EPS1 homolog	100.00%	detected
S092	TGA_E+R-	evm.model.Chr05.2747	EPS1 homolog	NA	
S092	TGA_E+R-	GSaman00003	EPS1 homolog	96.93%	detected
S076	TGA_E-R+	evm.model.Chr05.2618	Roq1-like homolog	NA	
S076	TGA_E-R+	GSaman00009	Roq1-like homolog	91.70%	
S076	TGA_E-R+	GSaman00011	Roq1-like homolog	NA	
S076	TGA_E-R+	GSaman00012	Roq1-like homolog	91.30%	
S076	TGA_E-R+	GSaman00013	Roq1-like homolog	92.16%	
S076	TGA_E-R+	GSaman00014	Roq1-like homolog	NA	
S076	TGA_E-R+	GSaman00015	Roq1-like homolog	89.29%	
S076	TGA_E-R+	GSaman00031	Roq1-like homolog	100.00%	detected
S076	TGA_E-R+	GSaman00033	Roq1-like homolog	94.83%	
S076	TGA_E-R+	GSaman00035	Roq1-like homolog	100.00%	detected
S076	TGA_E-R+	GSaman00039	Roq1-like homolog	100.00%	detected
S076	TGA_E-R+	GSaman00040	Roq1-like homolog	94.83%	
S076	TGA_E-R+	GSaman00041	Roq1-like homolog	93.65%	

Finally, the exact genotypes of the 'S065', 'S092', 'S050' accessions used for VIGS whose genotypes are not specified.

We apologize that this information was not presented more clearly to readers. The genotypes of the three accessions used for VIGS were determined using the three markers described in Fig. 6d, e --TGG/TGA, E+/E-, and R+/R-. Using a simple on/off scheme to denote marker states, we selected accessions in which the gene of interest was "on" while the other two genes were "off". Specifically, accession 'S065' (TGG_E-R-) was used for VIGS-SmCYP82D47, accession 'S092' (TGA_E+R-) was used for VIGS-SmEPS1, and accession 'S050' (TGA_E-R+) was used for VIGS-

SmRoq1.

We have now clarified this information in Results section (Lines: 515-519, 554-557) and updated the text accordingly.

All these shortcomings make the results still difficult to interpret. The authors should, for each of the genotypes above:

i) in a new table indicate the genes homologous to the VIGS fragment and their levels of homology;

We appreciate this insightful suggestion from the reviewer, which helped improve the clarity of our VIGS experiments. To evaluate potential off-target effects, we assessed sequence homology between each VIGS fragment and annotated genes. For SmEPS1, we used the 'S092' genome assembly as the reference and aligned the SmEPS1 VIGS fragment against all annotated coding sequences (CDSs) using BLASTn. For SmRoq1, because a genome assembly for 'S050' is not available, we used the 'S076' assembly as the reference, as it carries the same TGA_E-_R+ genotype as 'S050', and similarly aligned the SmRoq1 VIGS fragment against all CDSs using BLASTn (Lines: 953-955).

We have added Supplementary Table S17 and Supplementary Fig. 24 to summarize the genes showing homology to each VIGS fragment and their corresponding similarity levels. The SmEPS1 VIGS fragment showed high sequence similarity (>96%) to three of the four SmEPS1 homologs. The SmRoq1 VIGS fragment showed varying degrees of homology to 10 of the 13 SmRoq1 homologs, with sequence identities ranging from 89.29% to 100%, and no significant hits were detected for the remaining four homologs. Importantly, these significant matches were confined to SmEPS1 or SmRoq1 homologs, and no other annotated genes showed detectable homology to either VIGS fragment.

ii) in table S14 indicate which of the genes are recognized by the RT-PCR oligonucleotides;

Similarly to point i), we aligned the qRT-PCR primer pairs against all coding sequences (CDSs) using Bowtie with the parameters -S -f -v 2 -l 100 -X 5000 -y. Genes showing primer hits are listed in Supplementary Table 17. Each primer pair is aligned exclusively to SmEPS1 or SmRoq1 homologs, respectively. Notably, none of the previously reported genes associated with BW resistance (listed in former Table S14, now Table S15) showed any hits with either primer pair (Lines: 955-957).

For SmEPS1, the primer pair matched the same three EPS1 homologs that were also targeted by the VIGS fragment, indicating that this primer pair collectively reflects the expression levels of these three homologs.

For SmRoq1, the primer pair matched three Roq1 homologs that share 100% sequence identity with the VIGS fragment, indicating that this primer pair most likely reflects the expression levels of these three homologs.

iii) discuss the results of the VIGS experiment in view of the above information.

Thank you for this suggestion! Based our additional analysis we provide a short Discussion as follows:

Our VIGS experiments showed that silencing SmEPS1 or SmRoq1 altered BW resistance, supporting a functional involvement of these genes in disease resistance. Because both genes occur as multi-copy families, VIGS may co-silence closely related homologs due to sequence similarity (Fernandez-Pozo et al., 2015); nevertheless, sequence comparisons indicate that both the VIGS fragments and the qRT-PCR primers match only SmEPS1 or SmRoq1 homologs and not other genes (Supplementary Table 17; Supplementary Fig. 24). Thus, the VIGS phenotypes likely reflect the combined effects of silencing one or a subset of homologs within each gene family, rather than unintended impacts on unrelated loci.

We have added discussion points in the manuscript (Lines: 700-707).

Fernandez-Pozo, N., Rosli, H. G., Martin, G. B. & Mueller, L. A. The SGN VIGS tool: user-friendly software to design virus-induced gene silencing (VIGS) constructs for functional genomics. Mol Plant 8, 486-488 (2015).

Comments R5C7-R5C10. The comments were satisfactorily addressed.

We thank the reviewer for the positive evaluation of our previous revision.

While the authors perform these revisions, they should also address the inconsistencies in Loci returned by GWAS. The manuscript's narrative is built around the 12.4 Mb inversion on Chr10 as the primary example of an SV-trait association. The text explicitly states other loci are "less significant", and indeed Fig 4a seems to support this claim. However, in Table S8, at least one SV-GWAS hit (Chr05_81641732) has a comparable, if not higher significance. Why isn't it shown in Fig 4a? This is very confusing. The authors should revise the Manhattan plots in the main figures to match the data given in supplementary tables both for SNPs and SVs, and present in supplementary figures the comparable results for PGGB-GWAS (see comment R5C2).

Response [R5C12]:

We sincerely apologize for this inconsistency and the confusion it caused. After carefully re-examining the materials, we found that Supplementary Table 9 (previous Table S8) was inadvertently not updated after the initial submission and therefore remained outdated through subsequent rounds of revision. In contrast, the Manhattan plots were generated from the updated GWAS results, leading to the apparent mismatch noted by the reviewer. Notably, several loci that appeared significant in earlier versions are no longer significant or are not retained in the updated analyses, primarily due to the updated variant set and filtering procedure.

For example, for Chr05_81641732, we revisited this site in the original VCF and found a genotype distribution of 195 individuals with 0/0, 2 with 0/1, and 4 with 1/1, indicating an extremely low minor allele frequency. In addition, manual inspection did

not reveal an obvious genotype–phenotype association, and consistent with this, Chr05_81641732 does not appear as a significant signal in the updated GWAS results. Taken together, some of these loci likely represent false-positive signals in our earlier GWAS analysis.

To fully address the reviewer’s concern, we revised the manuscript as follow:

- 1. Updated former Table S8 (now Table S9) to the finalized GWAS output so that the reported loci are consistent with the Manhattan plots.*
- 2. Systematically verified consistency between all Manhattan plots (SNP-, InDel-, and SV-based) and the loci reported in the tables.*
- 3. Revised the main-text wording to remove potentially misleading language such as “less important/less significant”.*
- 4. Updated the main figures by consolidating the Manhattan plots from SNP-, InDel-, and SV-based GWAS into a single integrated presentation, and revised Fig. 4a (fruit color GWAS) and Fig. 6c (bacterial wilt resistance GWAS) accordingly.*
- 5. Following the reviewer’s suggestion, we additionally present the comparable reference-free PGGB-GWAS results in the Supplementary Information (now Supplementary Fig. 14).*

We hope that our revisions satisfactorily address the reviewers’ concerns, and we thank you again for your detailed comments and suggestions.

REVIEWER COMMENTS

Reviewer #5 (Remarks to the Author):

The authors have made substantial efforts to fully address all my comments, except for data availability, which is presently partially addressed.

Data availability. Partially addressed. An accession number for raw sequencing data, assemblies and annotations is provided (CNGB Project CNP0006177). However, the project is meant to go public on 2026-08-27. The data must be made public by the time the manuscript is published. Also, the full variant callsets and pangenome graph files (VG/PGGB) should be made public, preferably on a dedicated website like SGN.

We thank the reviewer for the positive evaluation of our previous revision.

*Regarding data availability, we would like to clarify that all datasets associated with this study have now been fully released and are publicly accessible. The raw sequencing data, genome assemblies, and annotations under CNGB Project **CNP0006177** have been made public and are currently available for viewing and download.*

*In addition, the complete variant callsets (VCF files) and the pangenome graph files have been deposited in the **Zenodo** repository and are publicly accessible via the following **DOI: 10.5281/zenodo.18425195**.*

We have updated the Data Availability section accordingly to reflect the public release of all relevant data.